# Asymptotically Free Sketched Ridge Ensembles: Risks, Cross-Validation, and Tuning

**Pratik Patil**
Department of Statistics
University of California, Berkeley
California, CA 94720, USA
`pratikpatil@berkeley.edu`

**Daniel LeJeune**
Department of Statistics
Stanford University
California, CA 94305, USA
`daniel@dlej.net`

## ABSTRACT

We employ random matrix theory to establish consistency of generalized cross validation (GCV) for estimating prediction risks of sketched ridge regression ensembles, enabling efficient and consistent tuning of regularization and sketching parameters. Our results hold for a broad class of asymptotically free sketches under very mild data assumptions. For squared prediction risk, we provide a decomposition into an unsketched equivalent implicit ridge bias and a sketching-based variance, and prove that the risk can be globally optimized by only tuning sketch size in infinite ensembles. For general subquadratic prediction risk functionals, we extend GCV to construct consistent risk estimators, and thereby obtain distributional convergence of the GCV-corrected predictions in Wasserstein-2 metric. This in particular allows construction of prediction intervals with asymptotically correct coverage conditional on the training data. We also propose an "ensemble trick" whereby the risk for unsketched ridge regression can be efficiently estimated via GCV using small sketched ridge ensembles. We empirically validate our theoretical results using both synthetic and real large-scale datasets with practical sketches including CountSketch and subsampled randomized discrete cosine transforms.

## 1 INTRODUCTION

*Random sketching* is a powerful tool for reducing the computational complexity associated with large-scale datasets by projecting them to a lower-dimensional space for efficient computations. Sketching has been a remarkable success both in practical applications and from a theoretical standpoint: it has enabled application of statistical techniques to problem scales that were formerly unimaginable (Aghazadeh et al., 2018; Murray et al., 2023), while enjoying rigorous technical guarantees that ensure the underlying learning problem essentially remains unchanged provided the sketch dimension is not too small (e.g., above the rank of the full data matrix) (Tropp, 2011; Woodruff, 2014).

However, real-world data scenarios often deviate from these ideal conditions for which the problem remains unchanged. For one, real data often has a tail of non-vanishing eigenvalues and is not truly low rank. For another, our available resources may impose constraints on sketch sizes, forcing them to fall below the critical threshold. When the sketch size is critically low, the learning problem can change significantly. In particular, when reducing the dimensionality below the threshold to solve the original problem, the problem becomes *implicitly regularized* (Mahoney, 2011; Thanei et al., 2017). Recent work has precisely characterized this problem change in linear regression (LeJeune et al., 2022), being exactly equal to ridge regression in an infinite ensemble of sketched predictors (LeJeune et al., 2020), with the size of the sketch acting as an additional hyperparameter that affects the implicit regularization.

If the underlying problem changes with sketching, a key question arises: *can we reliably and efficiently tune hyperparameters of sketched prediction models, such as the sketch size?* While cross-validation (CV) is the classical way to tune hyperparameters, standard $k$-fold CV (with small or moderate $k$ values, such as 5 or 10) is not statistically consistent for high-dimensional data (Xu et al., 2019), and leave-one-out CV (LOOCV) is often computationally infeasible. Generalized cross-validation (GCV), on the other hand, is an extremely efficient method for estimating generalization error using

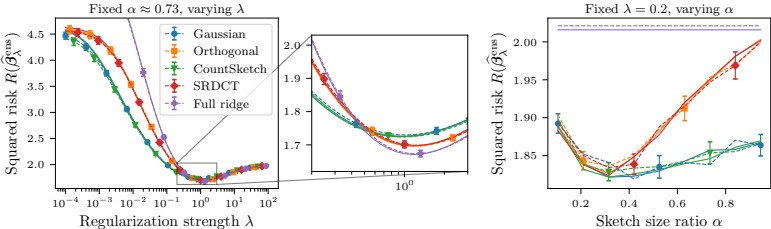

Figure 1: **GCV provides consistent risk estimation for sketched ridge regression.** We show squared risk (solid) and GCV estimates (dashed) for sketched regression ensembles of $K = 5$ predictors on synthetic data with $n = 500$ observations and $p = 600$ features. **Left:** Each sketch induces its own risk curve in regularization strength $\lambda$, but across all sketches GCV is consistent. **Middle:** Minimizers and minimum values can vary by sketching type. **Right:** Each sketch also induces a risk curve in sketch size $\alpha = q/p$, so sketch size can be tuned to optimize risk. Error bars denote standard error of the mean over 100 trials. Here, SRDCT refers to a subsampled randomized discrete cosine transform (see Appendix G for further details).

only training data (Craven & Wahba, 1979; Hastie et al., 2009), providing asymptotically exact error estimators in high dimensions with similar computational cost to fitting the model (Patil et al., 2021; Wei et al., 2022). However, since the consistency of GCV is due to certain concentration of measure phenomena of data, it is unclear whether GCV should also provide a consistent error estimator for predictors with sketched data, in particular when combining several sketched predictors in an ensemble, such as in distributed optimization settings.

In this work, we prove that efficient consistent tuning of hyperparameters of sketched ridge regression ensembles is achievable with GCV (see Figure 1 for an illustration). Furthermore, we state our results for a very broad class of *asymptotically free* sketching matrices, a notion from free probability theory (Voiculescu, 1997; Mingo & Speicher, 2017) generalizing rotational invariance. Below we present a summary of our main results in this paper and provide an outline of the paper.

1. **Squared risk asymptotics.** We provide precise asymptotics of squared risk and its GCV estimator for sketched ridge ensembles in Theorem 2 for the class of asymptotically free sketches applied to features. We give this result in terms of an exact bias–variance decomposition into an equivalent implicit unsketched ridge regression risk and an inflation term due to randomness of the sketch that is controlled by ensemble size.

2. **Distributional and functional consistencies.** We prove consistency of GCV risk estimators for a broad class of subquadratic risk functionals in Theorems 3 and 4. To the best of our knowledge, this is the first extension of GCV beyond residual-based risk functionals in any setting. In doing so, we also prove the consistency of estimating the joint response–prediction distribution using GCV in Wasserstein $W_2$ metric in Corollary 5, enabling the use of GCV for also evaluating classification error and constructing prediction intervals with valid asymptotic conditional coverage.

3. **Tuning applications.** Exploiting the special form of the risk decomposition, we propose a method in the form of an "ensemble trick" to tune unsketched ridge regression using only sketched ensembles. We also prove that large unregularized sketched ensembles with tuned sketch size can achieve the optimal unsketched ridge regression risk in Proposition 6.

Throughout all of our results, we impose very weak assumptions: we require no model on the relationship between response variables and features; we allow for arbitrary feature covariance with random matrix structure; we allow any sketch that satisfies asymptotic freeness, which we empirically verify for CountSketch (Charikar et al., 2004) and subsampled randomized discrete cosine transforms (SRDCT); and we allow for the consideration of zero or even negative regularization. All proofs and details of experiments and additional numerical illustrations are deferred to the appendices.

**Related work.** For context, we briefly discuss related work on sketching, ridge regression, and CV.

*Sketching and implicit regularization.* The implicit regularization effect of sketching has been known for some time (Mahoney, 2011; Thanei et al., 2017). This effect is strongly related to *inversion bias*, and has been precisely characterized in a number of settings in recent years (Mutny et al., 2020; Dereziński et al., 2021a;b). Most recently, LeJeune et al. (2022) showed that sketched matrix inversions are asymptotically equivalent to unsketched implicitly regularized inversions, and that this holds not only for i.i.d. random sketches but also for asymptotically free sketches. This result is a crucial component of our bias–variance decomposition of GCV risk. By allowing free sketches, we can apply our results to many sketches used in practice with limited prior theoretical understanding.

*High-dimensional ridge and sketching.* Ridge regression, particularly "ridgeless" regression where the regularization level approaches zero, has recently attracted great attention. In the overparameterized regime, where the number of features exceeds the number of observations, the ridgeless estimator interpolates the training data and exhibits a peculiar generalization behaviour (Belkin et al., 2020; Bartlett et al., 2020; Hastie et al., 2022). Different sketching variants and their risks for a single sketched ridge estimator under positive regularization are analyzed in Liu & Dobriban (2020). Very recently, Bach (2024) considers the effect random sketching that includes ridgeless regression. Our work broadens the scope of these works by considering all asymptotically free sketched ensembles and accommodating zero (and negative) regularization. Complementary to feature sketching, there is an emerging interest in subsampling and observation sketching. The statistical properties of various subsampled predictors are analyzed by Patil et al. (2023); Du et al. (2023); Patil & Du (2024); Chen et al. (2023); Ando & Komaki (2023). At a high level, this work acts as "dual" to this literature.

*Cross-validation and tuning.* CV is a prevalent method for model assessment and selection (Hastie et al., 2009). For surveys on CV variants, we refer readers to Arlot & Celisse (2010); Zhang & Yang (2015). Initially proposed for linear smoothers in the fixed-X design settings, GCV provides an extremely efficient alternative to traditional CV methods like LOOCV (Golub et al., 1979; Craven & Wahba, 1979). It approximates the so-called "shortcut" LOOCV formula (Hastie et al., 2009). More recently, there has been growing interest in GCV in the random-X design settings. Consistency properties of GCV have been investigated: for ridge regression under various scenarios (Adlam & Pennington, 2020; Patil et al., 2021; 2022; Wei et al., 2022; Han & Xu, 2023), for the lasso (Bayati & Montanari, 2011; Celentano et al., 2023), and for general regularized $M$-estimators (Bellec, 2023; Bellec & Shen, 2022), among others. Our work adds to this literature by analyzing GCV for freely sketched ridge ensembles and establishing its consistency across a broad class of risk functionals.

## 2 SKETCHED ENSEMBLES

Let $((\mathbf{x}_i, y_i))_{i=1}^n$ be $n$ i.i.d. observations in $\mathbb{R}^p \times \mathbb{R}$. We denote by $\mathbf{X} \in \mathbb{R}^{n \times p}$ the data matrix whose $i$-th row contains $\mathbf{x}_i^\top$ and by $\mathbf{y} \in \mathbb{R}^n$ the associated response vector whose $i$-th entry contains $y_i$.

**Sketched ensembles and risk functionals.** Consider a collection of $K$ independent sketching matrices $\mathbf{S}_k \in \mathbb{R}^{p \times q}$ for $k \in [K]$. We consider sketched ridge regression where we apply the sketching matrix $\mathbf{S}_k$ to the features (columns) of the data $\mathbf{X}$ only. We denote the sketching solution as

$$\widehat{\boldsymbol{\beta}}_\lambda^k = \mathbf{S}_k \widehat{\boldsymbol{\beta}}_\lambda^{\mathbf{S}_k} \quad \text{for} \quad \widehat{\boldsymbol{\beta}}_\lambda^{\mathbf{S}_k} = \underset{\boldsymbol{\beta} \in \mathbb{R}^q}{\arg\min} \, \tfrac{1}{n} \|\mathbf{y} - \mathbf{X}\mathbf{S}_k\boldsymbol{\beta}\|_2^2 + \lambda\|\boldsymbol{\beta}\|_2^2, \tag{1}$$

where $\lambda$ is the ridge regularization level. We obtain the final ensemble estimator as a simple un-weighted average of $K$ independently sketched predictors, each of which admits a simple expression:

$$\widehat{\boldsymbol{\beta}}_\lambda^{\text{ens}} = \frac{1}{K} \sum_{k=1}^K \widehat{\boldsymbol{\beta}}_\lambda^k, \quad \text{where} \quad \widehat{\boldsymbol{\beta}}_\lambda^k = \tfrac{1}{n}\mathbf{S}_k\big(\tfrac{1}{n}\mathbf{S}_k^\top\mathbf{X}^\top\mathbf{X}\mathbf{S}_k + \lambda\mathbf{I}_q\big)^{-1}\mathbf{S}_k^\top\mathbf{X}^\top\mathbf{y}. \tag{2}$$

It is worth mentioning that, in practice, it is not necessary to "broadcast" $\widehat{\boldsymbol{\beta}}_\lambda^{\mathbf{S}_k}$ back to $p$-dimensional space to realize $\widehat{\boldsymbol{\beta}}_\lambda^k$, and all computation can (and should) be done in the sketched domain. Note also that we allow for $\lambda$ to be possibly negative in when writing (2) (see Theorem 1 for details). Let $(\mathbf{x}_0, y_0)$ be a test point drawn independently from the same distribution as the training data. Risk functionals of the ensemble estimator are properties of the joint distribution of $(y_0, \mathbf{x}_0^\top \widehat{\boldsymbol{\beta}}_\lambda^{\text{ens}})$. Letting $P_\lambda^{\text{ens}}$ denote this distribution, we are interested in estimating linear functionals of $P_\lambda^{\text{ens}}$. That is, let $t : \mathbb{R}^2 \to \mathbb{R}$ be an error function. Define the corresponding conditional prediction risk functional as

$$T(\widehat{\boldsymbol{\beta}}_\lambda^{\text{ens}}) = \int t(y, z) \, \mathrm{d}P_\lambda^{\text{ens}}(y, z) = \mathbb{E}_{\mathbf{x}_0, y_0} \left[ t(y_0, \mathbf{x}_0^\top \widehat{\boldsymbol{\beta}}_\lambda^{\text{ens}}) \, \Big| \, \mathbf{X}, \mathbf{y}, (\mathbf{S}_k)_{k=1}^K \right]. \tag{3}$$

A special case is the squared risk when $t(y, z) = (y - z)^2$, denoted by $R(\widehat{\boldsymbol{\beta}}_\lambda^{\text{ens}})$ in the sequel.

**Proposed GCV plug-in estimators.** Note that each individual estimator $\widehat{\boldsymbol{\beta}}_\lambda^k$ of the ensemble is a linear smoother with smoothing matrix $\mathbf{L}_\lambda^k = \tfrac{1}{n}\mathbf{X}\mathbf{S}_k(\tfrac{1}{n}\mathbf{S}_k^\top\mathbf{X}^\top\mathbf{X}\mathbf{S}_k + \lambda\mathbf{I}_q)^{-1}\mathbf{S}_k^\top\mathbf{X}^\top$, in the sense that the training data predictions are given by $\mathbf{X}\widehat{\boldsymbol{\beta}}_\lambda^k = \mathbf{L}_\lambda^k \mathbf{y}$. This motivates our consideration of estimators based on generalized cross-validation (GCV) (Hastie et al., 2009, Chapter 7). Given any linear smoother of the responses with smoothing matrix $\mathbf{L}$, the GCV estimator of the squared prediction risk is $\tfrac{1}{n}\|\mathbf{y} - \mathbf{L}\mathbf{y}\|_2^2/(1 - \tfrac{1}{n}\text{tr}(\mathbf{L}))^2$. GCV enjoys certain consistency properties in the fixed-X setting (Li, 1985; 1986) and has recently been shown to also be consistent under various random-X settings for ridge regression (Patil et al., 2021; Wei et al., 2022; Han & Xu, 2023).

We extend the GCV estimator to general functionals by considering GCV as a plug-in estimator of squared risk of the form $\frac{1}{n}\sum_{i=1}^{n}(y_i - z_i)^2$. Determining the $z_i$ that correspond to GCV, we obtain the empirical distribution of GCV-corrected predictions as follows:

$$\widehat{P}_\lambda^{\mathrm{ens}} = \frac{1}{n}\sum_{i=1}^{n}\delta\left\{\left(y_i, \frac{x_i^\top \widehat{\beta}_\lambda^{\mathrm{ens}} - \frac{1}{n}\mathrm{tr}[\mathbf{L}_\lambda^{\mathrm{ens}}]y_i}{1 - \frac{1}{n}\mathrm{tr}[\mathbf{L}_\lambda^{\mathrm{ens}}]}\right)\right\}, \quad \text{where} \quad \mathbf{L}_\lambda^{\mathrm{ens}} = \frac{1}{K}\sum_{k=1}^{K}\mathbf{L}_\lambda^k. \tag{4}$$

Here $\delta\{\mathbf{a}\}$ denotes a Dirac measure located at an atom $\mathbf{a} \in \mathbb{R}^2$. To give some intuition as to why this is a reasonable choice, consider that when fitting a model, the predictions on training points will be excessively correlated with the training responses. In order to match the test distribution, we need to cancel this increased correlation, which we accomplish by subtracting an appropriately scaled $y_i$.

Using this empirical distribution, we form the plug-in GCV risk functional estimators

$$\widehat{T}(\widehat{\beta}_\lambda^{\mathrm{ens}}) = \frac{1}{n}\sum_{i=1}^{n} t\left(y_i, \frac{x_i^\top \widehat{\beta}_\lambda^{\mathrm{ens}} - \frac{1}{n}\mathrm{tr}[\mathbf{L}_\lambda^{\mathrm{ens}}]y_i}{1 - \frac{1}{n}\mathrm{tr}[\mathbf{L}_\lambda^{\mathrm{ens}}]}\right) \quad \text{and} \quad \widehat{R}(\widehat{\beta}_\lambda^{\mathrm{ens}}) = \frac{\frac{1}{n}\|\mathbf{y} - \mathbf{X}\widehat{\beta}_\lambda^{\mathrm{ens}}\|_2^2}{(1 - \frac{1}{n}\mathrm{tr}[\mathbf{L}_\lambda^{\mathrm{ens}}])^2}. \tag{5}$$

In the case where $\lambda \to 0^+$ but ridgeless regression is well-defined, the denominator may tend to zero. However, the numerator will also tend to zero, and therefore one should interpret this quantity as its analytic continuation, which is also well-defined. In practice, if so desired, one can choose very small (positive and negative) $\lambda$ near zero and interpolate for a first-order approximation.

We emphasize that the GCV-corrected predictions are "free lunch" in most circumstances. For example, when tuning over $\lambda$, it is common to precompute a decomposition of $\mathbf{X}\mathbf{S}_k$ such that subsequent matrix inversions for each $\lambda$ are very inexpensive, and the same decomposition can be used to evaluate $\frac{1}{n}\mathrm{tr}[\mathbf{L}_\lambda^{\mathrm{ens}}]$ exactly. Otherwise, Monte-Carlo trace estimation is a common strategy for GCV (Girard, 1989; Hutchinson, 1989) that yields consistent estimators using very few (even single) samples, such that the additional computational cost is essentially the same as fitting the model. See Appendix H for computational complexity comparisons of various cross-validation methods.

## 3 SQUARED RISK ASYMPTOTICS AND CONSISTENCY

We now derive the asymptotics of squared risk and its GCV estimator for the finite ensemble sketched estimator. The special structure of the squared risk allows us to obtain explicit forms of the asymptotics that shed light on the dependence of both the ensemble risk and GCV on $K$, the size of the ensemble. We then show consistency of GCV for squared risk using these asymptotics.

We express our asymptotic results using the asymptotic equivalence notation $\mathbf{A}_n \simeq \mathbf{B}_n$, which means that for any sequence of $\mathbf{\Theta}_n$ having $\|\mathbf{\Theta}_n\|_{\mathrm{tr}} = \mathrm{tr}\left[(\mathbf{\Theta}_n\mathbf{\Theta}_n^\top)^{1/2}\right]$ uniformly bounded in $n$, $\lim_{n\to\infty}\mathrm{tr}\left[\mathbf{\Theta}_n(\mathbf{A}_n - \mathbf{B}_n)\right] = 0$ almost surely. In the case that $\mathbf{A}_n$ and $\mathbf{B}_n$ are scalars $a_n$ and $b_n$ such as risk estimators, this reduces to $\lim_{n\to\infty}(a_n - b_n) = 0$. Our forthcoming results apply to a sequence of problems of increasing dimensionality proportional to $n$, and we omit the explicit dependence on $n$ in our statements.

For our theoretical analysis, we need our sketching matrix $\mathbf{S}$ to have favorable properties. The sketch should preserve much of the essential structure of the data, even through (regularized) matrix inversion. A sufficient yet quite general condition for this is *freeness* (Voiculescu, 1997; Mingo & Speicher, 2017).

**Assumption A** (Sketch structure). Let $\mathbf{S}\mathbf{S}^\top$ and $\frac{1}{n}\mathbf{X}^\top\mathbf{X}$ converge almost surely to bounded operators infinitesimally free with respect to $(\frac{1}{p}\mathrm{tr}[\cdot], \mathrm{tr}[\mathbf{\Theta}(\cdot)])$ for any $\mathbf{\Theta}$ independent of $\mathbf{S}$ with $\|\mathbf{\Theta}\|_{\mathrm{tr}}$ uniformly bounded, and let $\mathbf{S}\mathbf{S}^\top$ have limiting S-transform $\mathscr{S}_{\mathbf{S}\mathbf{S}^\top}$ analytic on $\mathbb{C}^-$.

We give a background on freeness including infinitesimal freeness (Shlyakhtenko, 2018) in Appendix A. Intuitively, freeness of a pair of operators $\mathbf{A}$ and $\mathbf{B}$ means that the eigenvectors of one are completely unaligned or incoherent with the eigenvectors of the other. For example, if $\mathbf{A} = \mathbf{U}\mathbf{D}\mathbf{U}^\top$ for a uniformly random unitary matrix $\mathbf{U}$ drawn independently of positive semidefinite $\mathbf{B}$ and $\mathbf{D}$, then $\mathbf{A}$ and $\mathbf{B}$ are almost surely asymptotically infinitesimally free (Cébron et al., 2022).[1] For this reason, we expect any sketch that is *rotationally invariant*, a desired property of sketches in practice as we do not wish the sketch to prefer any particular dimensions of our data, to satisfy Assumption A.

---

[1] Note that this includes the two sketches most commonly studied theoretically: those with i.i.d. Gaussian entries, and random orthogonal projections, which both have analytic S-transforms (see Appendix A.4).

The property that the sketch preserves the structure of the data is captured in the notion of sub-ordination and conditional expectation in free probability (Biane, 1998), closely related to the *deterministic equivalents* (Dobriban & Sheng, 2020; 2021) used in random matrix theory. LeJeune et al. (2022) recently extended such results to infinitesimally free operators in the context of sketching, which will form the basis of our analysis.[2] For the statements to follow, define $\widehat{\boldsymbol{\Sigma}} = \frac{1}{n}\mathbf{X}^\top\mathbf{X}$ and $\lambda_0 = -\liminf_{p\to\infty}\lambda^+_{\min}(\mathbf{S}^\top\widehat{\boldsymbol{\Sigma}}\mathbf{S})$. Here $\lambda^+_{\min}(\mathbf{A})$ denotes the minimum nonzero eigenvalue of a symmetric matrix $\mathbf{A}$. In addition, define the population covariance matrix $\boldsymbol{\Sigma} = \mathbb{E}[\mathbf{x}_0\mathbf{x}_0^\top]$.

**Theorem 1** (Free sketching equivalence; LeJeune et al. (2022), Theorem 7.2). *Under Assumption A, for all $\lambda > \lambda_0$,*

$$\mathbf{S}(\mathbf{S}^\top\widehat{\boldsymbol{\Sigma}}\mathbf{S} + \lambda\mathbf{I}_q)^{-1}\mathbf{S}^\top \simeq (\widehat{\boldsymbol{\Sigma}} + \mu\mathbf{I}_p)^{-1}, \tag{6}$$

*where $\mu > -\lambda^+_{\min}(\widehat{\boldsymbol{\Sigma}})$ is increasing in $\lambda > \lambda_0$ and satisfies*

$$\mu \simeq \lambda\mathscr{S}_{\mathbf{S}\mathbf{S}^\top}\left(-\frac{1}{p}\mathrm{tr}\left[\mathbf{S}^\top\widehat{\boldsymbol{\Sigma}}\mathbf{S}(\mathbf{S}^\top\widehat{\boldsymbol{\Sigma}}\mathbf{S} + \lambda\mathbf{I}_q)^{-1}\right]\right) \simeq \lambda\mathscr{S}_{\mathbf{S}\mathbf{S}^\top}\left(-\frac{1}{p}\mathrm{tr}\left[\widehat{\boldsymbol{\Sigma}}(\widehat{\boldsymbol{\Sigma}} + \mu\mathbf{I}_p)^{-1}\right]\right). \tag{7}$$

Put another way, when we sketch $\widehat{\boldsymbol{\Sigma}}$ and compute a regularized inverse, it is (in a first-order sense) as if we had computed an unsketched regularized inverse of $\widehat{\boldsymbol{\Sigma}}$, potentially with a different "implicit" regularization strength $\mu$ instead of $\lambda$. Since the result holds for free sketching matrices, we expect this to include fast practical sketches such as CountSketch (Charikar et al., 2004) and subsampled randomized Fourier and Hadamard transforms (SRFT/SRHT) (Tropp, 2011; Lacotte et al., 2020), which were demonstrated empirically to satisfy the same relationship by LeJeune et al. (2022), and for which we also provide further empirical support in this work in Appendices A.2 and A.3.

While the form of the relationship between the original and implicit regularization parameters $\lambda$ and $\mu$ in Theorem 1 may seem complicated, the remarkable fact is that our GCV consistency results in the next section are agnostic to the specific form of any of the quantities involved (such as $\mathscr{S}_{\mathbf{S}\mathbf{S}^\top}$ and $\mu$). That is, GCV is able to make the appropriate correction in a way that adapts to the specific choice of sketch, such that the statistician need not worry. Nevertheless, for the interested reader we provide a listing of known examples of sketches satisfying Assumption A and their corresponding S-transforms in Table 4 in Appendix A.4, parameterized by $\alpha = q/p$.

We first state a result on the decomposition of squared risk and the GCV estimator. Here we let $\widehat{\boldsymbol{\beta}}^{\mathrm{ridge}}_\mu$ denote the ridge estimator fit on unsketched data at the implicit regularization parameter $\mu$.

**Theorem 2** (Risk and GCV asymptotics). *Suppose Assumption A holds, and that the operator norm of $\boldsymbol{\Sigma}$ and second moment of $y_0$ are uniformly bounded in $p$. Then, for $\lambda > \lambda_0$ and all $K$,*

$$R(\widehat{\boldsymbol{\beta}}^{\mathrm{ens}}_\lambda) \simeq R(\widehat{\boldsymbol{\beta}}^{\mathrm{ridge}}_\mu) + \frac{\mu'\Delta}{K} \quad and \quad \widehat{R}(\widehat{\boldsymbol{\beta}}^{\mathrm{ens}}_\lambda) \simeq \widehat{R}(\widehat{\boldsymbol{\beta}}^{\mathrm{ridge}}_\mu) + \frac{\mu''\Delta}{K}, \tag{8}$$

*where $\mu$ is as given in Theorem 1, $\Delta = \frac{1}{n}\mathbf{y}^\top(\frac{1}{n}\mathbf{X}\mathbf{X}^\top + \mu\mathbf{I}_n)^{-2}\mathbf{y} \geqslant 0$, and $\mu' \geqslant 0$ is a certain non-negative inflation factor in the risk that only depends on $\mathscr{S}_{\mathbf{S}\mathbf{S}^\top}$, $\widehat{\boldsymbol{\Sigma}}$, and $\boldsymbol{\Sigma}$, while $\mu'' \geqslant 0$ is a certain non-negative inflation factor in the risk estimator that only depends on $\mathscr{S}_{\mathbf{S}\mathbf{S}^\top}$ and $\widehat{\boldsymbol{\Sigma}}$.*

In other words, this result gives *bias–variance* decompositions for both squared risk and its GCV estimator for sketched ensembles. The result says that the risk of the sketched predictor is equal to the risk of the unsketched equivalent implicit ridge regressor (bias) plus a term due to the randomness of the sketching that depends on the inflation factor $\mu'$ or $\mu''$ (variance), which is controlled by the ensemble size at a rate of $1/K$ (see Figure G.1 for a numerical verification of this rate).

We refer the reader to Theorem 16 in Appendix C for precise expressions for $\mu'$ and $\mu''$, and to LeJeune et al. (2022) for illustrations of their relationship of these parameters with $\alpha$ and $\lambda$ in the case of i.i.d. sketching. For expressions of limiting non-sketched risk and GCV for ridge regression, we also refer to Patil et al. (2021), which could be combined with (8) to obtain exact formulas for asymptotic risk and GCV for sketched ridge regression, or to Bach (2024) for exact squared risk expressions in the i.i.d. sketching case for $K = 1$.

For our consistency result, we impose certain mild random matrix assumptions on the feature vectors and assume a mild bounded moment condition on the response variable. Notably, we do not require any specific model assumption on the response variable $y$ in the way that it relates to the feature vector $\mathbf{x}$. Thus, all of our results are applicable in a model-free setting.

---

[2]The original theorem in LeJeune et al. (2022) was given for complex $\lambda$ and $\mu$, but the stated version follows by analytic continuation to the real line.

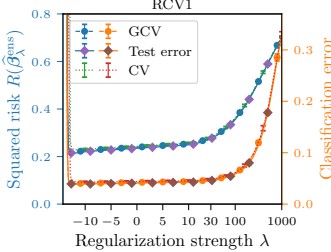 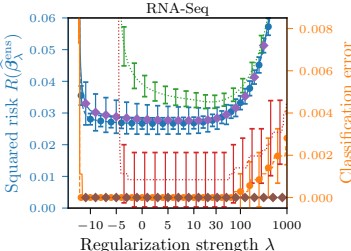

Figure 2: **GCV provides very accurate risk estimates for real-world data.** We fit ridge regression ensembles of size $K = 5$ using CountSketch (Charikar et al., 2004) on binary $\pm 1$ labels from RCV1 (Lewis et al., 2004) ($n = 20000$, $p = 30617$, $q = 515$) (**left**) and RNA-Seq (Weinstein et al., 2013) ($n = 356$, $p = 20223$, $q = 99$) (**right**). GCV (dashed, circles) matches test risk (solid, diamonds) and improves upon 2-fold CV (dotted) for both squared error (blue, green) and classification error (orange, red). CV provides poorer estimates for less positive $\lambda$, heavily exaggerated when $n$ is small such as in RNA-Seq. Error bars denote std. dev. over 10 trials.

**Assumption B** (Data structure). *The feature vector decomposes as* $\mathbf{x} = \mathbf{\Sigma}^{1/2}\mathbf{z}$, *where* $\mathbf{z} \in \mathbb{R}^p$ *contains i.i.d. entries with mean* 0, *variance* 1, *bounded moments of order* $4 + \delta$ *for some* $\delta > 0$, *and* $\mathbf{\Sigma} \in \mathbb{R}^{p \times p}$ *is a symmetric matrix with eigenvalues uniformly bounded between* $r_{\min} > 0$ *and* $r_{\max} < \infty$. *The response* $y$ *has mean* 0 *and bounded moment of order* $4 + \delta$ *for some* $\delta > 0$.

The assumption of zero mean in the features and response is only done for mathematical simplicity. To deal with non-zero mean, one can add an (unregularized) intercept to the predictor, and all of our results can be suitably adapted. We apply such an intercept in our experiments on real-world data.

It has been recently shown that GCV for unsketched ridge regression is an asymptotically consistent estimator of risk (Patil et al., 2021), so given our bias–variance decomposition in (8), the only question is whether the variance term from GCV is a consistent estimator of the variance term of risk. This indeed turns out to be the case, as we state in the following theorem for squared risk.

**Theorem 3** (GCV consistency). *Under Assumptions A and B, for* $\lambda > \lambda_0$ *and all* $K$, *it holds that*

$$\mu' \simeq \mu'', \quad \text{and therefore} \quad \widehat{R}(\widehat{\boldsymbol{\beta}}_\lambda^{\text{ens}}) \simeq R(\widehat{\boldsymbol{\beta}}_\lambda^{\text{ens}}). \tag{9}$$

The remarkableness of this result is its generality: we have made no assumption on a particular choice of sketching matrix (see Figure 1) or the size $K$ of the ensemble. We also make no assumption other than boundedness on the covariance $\mathbf{\Sigma}$, and we do not require any model on the relation of the response to the data. Furthermore, this result is not marginal but rather conditional on $\mathbf{X}, \mathbf{y}, (\mathbf{S}_k)_{k=1}^K$, meaning that we can trust GCV to be consistent for tuning on a single learning problem. We also emphasize that our results holds for positive, zero, and even negative $\lambda$ generally speaking. This is important, as negative regularization can be optimal in ridge regression in certain circumstances (Kobak et al., 2020; Wu & Xu, 2020; Richards et al., 2021) and even more commonly in sketched ridge ensembles (LeJeune et al., 2022), as we demonstrate in Figure 2.

Observe that for $K = 1$, that is, sketched ridge regression, one can absorb the sketching matrix $\mathbf{S}$ into the data matrix $\mathbf{X}$ such that the transformed data $\widetilde{\mathbf{X}} = \mathbf{XS}$ satisfies Assumption B. We therefore directly obtain the consistency of GCV in this case using results of Patil et al. (2021). The novel aspect of Theorem 3 is thus that the consistency of GCV holds for ensembles of any $K$, which is not obvious, due to the interactions across predictors in squared error. The non-triviality of this result is perhaps subtle: one may wonder whether GCV is always consistent under any sketching setting. However, as we discuss later in Proposition 7, when sketching observations, GCV fails to be consistent, and so we cannot blindly assert that sketching and GCV are always compatible.

## 4 GENERAL FUNCTIONAL CONSISTENCY

In the previous section, we obtained an elegant decomposition for squared risk and the GCV estimator that cleanly captures the effect of ensembling as controlling the variance from an equivalent unsketched implicit ridge regression risk at a rate of $1/K$. However, we are also interested in using GCV for evaluating other risk functionals, which do not yield bias–variance decompositions.

Fortunately, however, we can leverage the close connection between GCV and LOOCV to prove the consistency for a broad class of *subquadratic* risk functionals. As a result, we also certify that the *distribution* of the GCV-corrected predictions converges to the test distribution. We show convergence

Figure 3: **GCV provides consistent prediction intervals (PIs) and distribution estimates. Left:** We construct GCV prediction intervals for SRDCT ensembles of size $K = 5$ to synthetic data ($n = 1500$, $p = 1000$) with nonlinear responses $y = \mathrm{soft\,threshold}(\mathbf{x}^\top \boldsymbol{\beta}_0)$. **Mid-left:** We use GCV to tune our model to optimize prediction interval width. **Right:** The empirical GCV estimate $\widehat{P}^{\mathrm{ens}}_\lambda$ (here for $\alpha = 0.68$) closely matches the true joint response–prediction distribution $P^{\mathrm{ens}}_\lambda$. Error bars denote standard deviation over 30 trials.

for all error functions $t$ in (3) satisfying the following subquadratic growth condition, commonly used in the approximate message passing (AMP) literature (see, e.g., Bayati & Montanari, 2011).

**Assumption C** (Test error structure). The error function $t \colon \mathbb{R}^2 \to \mathbb{R}$ is pseudo-Lipschitz of order 2. That is, there exists a constant $L > 0$ such that for all $\mathbf{u}, \mathbf{v} \in \mathbb{R}^2$, the following bound holds true: $|t(\mathbf{u}) - t(\mathbf{v})| \leqslant L(1 + \|\mathbf{u}\|_2 + \|\mathbf{v}\|_2)\|\mathbf{u} - \mathbf{v}\|_2$.

The growth condition on $t$ in the assumption above is ultimately tied to our assumptions on the bounded moment of order $4 + \delta$ for some $\delta > 0$ on the entries of the feature vector and the response variable. By imposing stronger the moment assumptions, one can generalize these results for error functions with higher growth rates at the expense of less data generality.

We remark that this extends the class of functionals previously shown to be consistent for GCV in ridge regression (Patil et al., 2022), which were of the residual form $t(y - z)$. While the tools needed for this extension are not drastically different, it is nonetheless a conceptually important extension. In particular, this is useful for classification problems where metrics do not have a residual structure and for adaptive prediction interval construction. We now state our main consistency result.

**Theorem 4** (Functional consistency). *Under Assumptions A to C, for $\lambda > \lambda_0$ and all $K$,*

$$\widehat{T}(\widehat{\boldsymbol{\beta}}^{\mathrm{ens}}_\lambda) \simeq T(\widehat{\boldsymbol{\beta}}^{\mathrm{ens}}_\lambda). \tag{10}$$

Since $t(y, z) = (y - z)^2$ satisfies Assumption C, this result is strict generalization of Theorem 3. This class of risk functionals is very broad: it includes for example robust risks such as the mean absolute error or Huber loss, and even classification risks such as hinge loss and logistic loss.

Furthermore, this class of error functions is sufficiently rich as to guarantee that not only do risk functionals converge, but in fact the GCV-corrected predictions also converge in distribution to the predictions of test data. This simple corollary captures the fact that empirical convergence of pseudo-Lipschitz functionals of order 2, being equivalent to weak convergence plus convergence in second moment, is equivalent to Wasserstein convergence (Villani, 2008, Chapter 6).

**Corollary 5** (Distributional consistency). *Under Assumptions A and B, for $\lambda > \lambda_0$ and all $K$, it holds that $\widehat{P}^{\mathrm{ens}}_\lambda \overset{2}{\Rightarrow} P^{\mathrm{ens}}_\lambda$, where $\overset{2}{\Rightarrow}$ denotes convergence in Wasserstein $W_2$ metric.*

Distributional convergence further enriches our choices of consistent estimators that we can construct with GCV, in that we can now construct estimators of sets and their probabilities. One example is classification error $\mathbb{E}[\mathbb{1}\{y_0 \neq \mathrm{sign}(\mathbf{x}_0^\top \widehat{\boldsymbol{\beta}}^{\mathrm{ens}}_\lambda)\}]$, which can be expressed in terms of conditional probability over discrete $y_0$. In our real data experiments in Figure 2, we also compute classification error using GCV and find it yields highly consistent estimates, which is useful as squared error (and hence ridge) is known to be a competitive loss function for classification (Hui & Belkin, 2021).

Of statistical interest, we can also do things such as construct prediction intervals using the GCV-corrected empirical distribution. For example, for $\tau \in (0, 1)$, consider the level-$\tau$ quantile $\widehat{Q}(\tau) = \inf\{z \colon \widehat{F}(z) \geqslant \tau\}$ and prediction interval $\mathcal{I} = [\mathbf{x}_0^\top \widehat{\boldsymbol{\beta}}^{\mathrm{ens}}_\lambda + \widehat{Q}(\tau_l), \mathbf{x}_0^\top \widehat{\boldsymbol{\beta}}^{\mathrm{ens}}_\lambda + \widehat{Q}(\tau_u)]$, where $\widehat{F}$ is the cumulative distribution function (CDF) of the GCV residuals $(y - z) \colon (y, z) \sim \widehat{P}^{\mathrm{ens}}_\lambda$. Then $\mathcal{I}$ is a prediction interval for $y_0$ built only from training data that has the right coverage $\tau_u - \tau_l$, conditional on the training data, asymptotically almost surely. Furthermore, we can tune our model based on prediction interval metrics such as interval width. We demonstrate this idea in the experiment in Figure 3. This idea could be further extended to produce tighter *locally adaptive* prediction intervals by leveraging the entire joint distribution $\widehat{P}^{\mathrm{ens}}_\lambda$ rather than only the residuals.

## 5    TUNING APPLICATIONS AND THEORETICAL IMPLICATIONS

The obvious implication of the consistency results for GCV stated above is that we can also consistently tune sketched ridge regression: for any finite collection of hyperparameters ($\lambda$, $\alpha$, sketching family, $K$) over which we tune, consistency at each individual choice of hyperparameters implies that optimization over the hyperparameter set is also consistent. Thus if the predictor that we want to fit to our data is a sketched ridge regression ensemble, direct GCV enables us to efficiently tune it.

However, suppose we have the computational budget to fit a single large predictor, such as unsketched ridge regression or a large ensemble. Due to the large cost of refitting, tuning this predictor directly might be unfeasible. Fortunately, thanks to the bias–variance decomposition in Theorem 2, we can use small sketched ridge ensembles to tune such large predictors.

The key idea is to recall that asymptotically, the sketched risk is simply a linear combination of the equivalent ridge risk and a variance term, and that we can control the mixing of these terms by choice of the ensemble size $K$. This means that by choosing multiple distinct values of $K$, we can solve for the equivalent ridge risk. As a concrete example, suppose we have an ensemble of size $K = 2$ with corresponding risk $R_2 = R(\widehat{\boldsymbol{\beta}}_\lambda^{\mathrm{ens}})$, and let $R_1$ be the risk corresponding to the individual members of the ensemble. Then we can eliminate the variance term and obtain the equivalent risk as

$$R(\widehat{\boldsymbol{\beta}}_\mu^{\mathrm{ridge}}) \simeq 2R_2 - R_1. \tag{11}$$

Subsequently using the subordination relation $\mu \simeq \lambda \mathscr{S}_{\mathbf{SS}^\top}\big( -\frac{1}{p}\mathrm{tr}[\mathbf{S}^\top\widehat{\boldsymbol{\Sigma}}\mathbf{S}(\mathbf{S}^\top\widehat{\boldsymbol{\Sigma}}\mathbf{S} + \lambda\mathbf{I}_q)^{-1}]\big)$ from Theorem 1, we can map our choice of $\lambda$ and $\mathbf{S}$ to the equivalent $\mu$. By Theorem 3, we can use the GCV risk estimators for $R_1$ and $R_2$ and have a consistent estimator for ridge risk at $\mu$. In this way, we obtain a consistent estimator of risk that can be computed entirely using only the $q$-dimensional sketched data rather than the full $p$-dimensional data, which can be computed in less time with a smaller memory footprint. See Appendix H for a detailed comparison of computational complexity.

We demonstrate this "ensemble trick" for estimating ridge risk in Figure 4, which is accurate even where the variance component of sketched ridge risk is large. Furthermore, even though GCV is not consistent for sketched observations instead of features (see Section 6), the ensemble trick still provides a consistent estimator for ridge risk since the bias term is unchanged. One limitation of this method when considering a fixed sketch $\mathbf{S}$, varying only $\lambda$, is that this limits the minimum value of $\mu$ that can be considered (see discussion by LeJeune et al., 2022). A solution to this is to consider varying sketch sizes, allowing the full range of $\mu > 0$, as captured by the following result.

**Proposition 6** (Optimized GCV versus optimized ridge). *Under Assumptions A and B, if $\mathbf{S}_k^\top\mathbf{S}_k$ is invertible, then for any $\mu > 0$, if $\lambda = 0$ and $K \to \infty$, it holds that*

$$\widehat{R}(\widehat{\boldsymbol{\beta}}_\lambda^{\mathrm{ens}}) \simeq R(\widehat{\boldsymbol{\beta}}_\lambda^{\mathrm{ens}}) \simeq R(\widehat{\boldsymbol{\beta}}_\mu^{\mathrm{ridge}}) \quad for \quad \alpha = \tfrac{1}{p}\mathrm{tr}[\widehat{\boldsymbol{\Sigma}}\big(\widehat{\boldsymbol{\Sigma}} + \mu\mathbf{I}_p\big)^{-1}]. \tag{12}$$

That is, for any desired level of equivalent regularization $\mu$, we can obtain a sketched ridge regressor with the same bias (equivalently, the same large ensemble risk as $K \to \infty$) by changing only the sketch size and fixing $\lambda = 0$. A narrower result was shown for subsampled ensembles by LeJeune et al. (2020), but our generalization provides equivalences for all $\mu > 0$ and holds for any full-rank sketching matrix, establishing that freely sketched predictors indeed cover the same predictive space as their unsketched counterparts. The result also has practical merit. It guarantees that, with a sufficiently large sketched ensemble, we retain the statistical properties of the unsketched ridge regression. Thus, practitioners can harness the computational benefits of sketching, such as reduced memory usage and enhanced parallelization capabilities, without a loss in statistical performance.

## 6    DISCUSSION

This paper establishes the consistency of GCV-based estimators of risk functionals. We show that GCV provides a method for consistent fast tuning of sketched ridge ensemble parameters. However, taking a step back, given the connection between the sketched pseudoinverse and implicit ridge regularization in the unsketched inverse (Assumption A) and the fact that GCV "works" for ridge regression (Patil et al., 2021; Wei et al., 2022), one might wonder if the results in this paper were "expected"? The introduction of the ensemble required additional analysis of course, but perhaps the results seem intuitively natural.

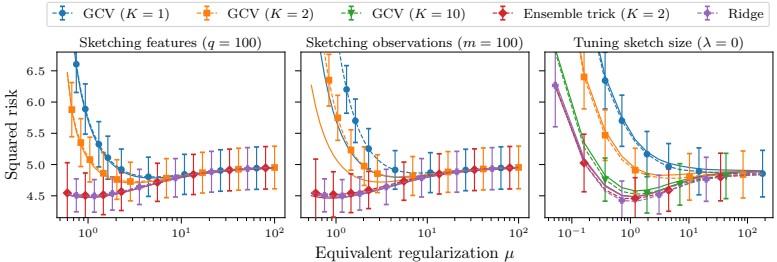

Figure 4: **GCV combined with sketching yields a fast method for tuning ridge.** We fit SRDCT ensembles on synthetic data ($n = 600$, $p = 800$), sketching features (**left** and **right**) or observations (**middle**). GCV (dashed) provides consistent estimates of test risk (solid) for feature sketching but not for observation sketching. However, the ensemble trick in (11) does not depend on the variance and thus works for both. For $\lambda = 0$, each equivalent $\mu > 0$ can be achieved by an appropriate choice of $\alpha$. Error bars denote standard deviation over 50 trials.

Surprisingly (even to the authors), if one changes the strategy from sketching features to sketching observations, we no longer have GCV consistency for finite ensembles! Consider a formulation where we now sketch observations with $K$ independent sketching matrices $\mathbf{T}_k \in \mathbb{R}^{n \times m}$ for $k \in [K]$:

$$\widetilde{\boldsymbol{\beta}}_\lambda^k = \arg\min_{\boldsymbol{\beta} \in \mathbb{R}^p} \tfrac{1}{n} \left\| \mathbf{T}_k^\top (\mathbf{y} - \mathbf{X}\boldsymbol{\beta}) \right\|_2^2 + \lambda \|\boldsymbol{\beta}\|_2^2 = \tfrac{1}{n} \left( \tfrac{1}{n} \mathbf{X}^\top \mathbf{T}_k \mathbf{T}_k^\top \mathbf{X} + \lambda \mathbf{I} \right)^{-1} \mathbf{X}^\top \mathbf{T}_k \mathbf{T}_k^\top \mathbf{y}. \quad (13)$$

Let the ensemble estimator $\widetilde{\boldsymbol{\beta}}_\lambda^{\mathrm{ens}}$ be defined analogously to (2). Note again that the ensemble estimator $\widetilde{\boldsymbol{\beta}}_\lambda^{\mathrm{ens}}$ is a linear smoother with smoothing matrix $\widetilde{\mathbf{L}}_\lambda^{\mathrm{ens}} = \tfrac{1}{K} \sum_{k=1}^K \widetilde{\mathbf{L}}_\lambda^k$, where $\widetilde{\mathbf{L}}_\lambda^k = \tfrac{1}{n} \mathbf{X} (\tfrac{1}{n} \mathbf{X}^\top \mathbf{T}_k \mathbf{T}_k^\top \mathbf{X} + \lambda \mathbf{I})^{-1} \mathbf{X}^\top \mathbf{T}_k \mathbf{T}_k^\top$. We can then define the GCV estimator $\widetilde{R}(\widetilde{\boldsymbol{\beta}}_\lambda^{\mathrm{ens}})$ of the squared risk analogous to (5). The following result shows that $\widetilde{R}$ is *inconsistent* for any $K$.

**Proposition 7** (GCV inconsistency for observation sketch). *Suppose assumptions of Theorem 2 hold for $\mathbf{T}\mathbf{T}^\top$. Then, for $\lambda > \widetilde{\lambda}_0 = -\liminf_{p \to \infty} \min_{k \in [K]} \lambda_{\min}^+ (\tfrac{1}{n} \mathbf{X}^\top \mathbf{T}_k \mathbf{T}_k^\top \mathbf{X})$ and all $K$,*

$$R(\widetilde{\boldsymbol{\beta}}_\lambda^{\mathrm{ens}}) \simeq R(\widetilde{\boldsymbol{\beta}}_\nu^{\mathrm{ridge}}) + \frac{\nu' \widetilde{\Delta}}{K} \quad and \quad \widetilde{R}(\widetilde{\boldsymbol{\beta}}_\lambda^{\mathrm{ens}}) \simeq \widetilde{R}(\widetilde{\boldsymbol{\beta}}_\nu^{\mathrm{ridge}}) + \frac{\nu'' \widetilde{\Delta}}{K}, \quad (14)$$

*where $\nu > -\lambda_{\min}^+ (\tfrac{1}{n} \mathbf{X}\mathbf{X}^\top)$ and satisfies $\nu = \lambda \mathscr{S}_{\mathbf{T}\mathbf{T}^\top} \left( - \tfrac{1}{n} \mathrm{tr}[\tfrac{1}{n} \mathbf{X}\mathbf{X}^\top (\tfrac{1}{n} \mathbf{X}\mathbf{X}^\top + \nu \mathbf{I}_n)^{-1}] \right)$, and $\widetilde{\Delta} = \tfrac{1}{n} \mathbf{y}^\top (\tfrac{1}{n} \mathbf{X}\mathbf{X}^\top + \nu \mathbf{I}_n)^{-2} \mathbf{y} \geqslant 0$, and $\widetilde{\nu}'$, $\widetilde{\nu}'' \geqslant 0$ are certain non-negative inflation factors. Furthermore, under Assumption B, in general we have $\nu' \not\eqsim \nu''$, and therefore $\widetilde{R}(\widetilde{\boldsymbol{\beta}}_\lambda^{\mathrm{ens}}) \not\eqsim R(\widetilde{\boldsymbol{\beta}}_\lambda^{\mathrm{ens}})$.*

For precise expressions of $\nu'$ and $\nu''$, we defer readers to Proposition 19 in Appendix F. Note that as $K \to \infty$, the variance terms vanish and we get back consistency; for this reason, the "ensemble trick" in (11) still works. This negative result highlights the subtleties in the results in this paper, and that the GCV consistency for sketched ensembles of finite $K$ is far from obvious and needs careful analysis to check whether it is consistent. This result is similar in spirit to the GCV inconsistency results of Bellec et al. (2023) and Patil et al. (2024) in subsampling and early stopping contexts, respectively. It is still possible to correct GCV in our case, as we detail in Appendix F.2, using unsketched data.

While our results are quite general in terms of being applicable to a wide variety of data and sketches, they are limited in that they apply only to ridge regression with isotropic regularization. However, we believe that the tools used in this work are useful in extending GCV consistency and the understanding of sketching to many other linear learning settings. It is straightforward to extend our results beyond isotropic ridge regularization. We might want to apply generalized anisotropic ridge regularization in real-world scenarios: generalized ridge achieves Bayes-optimal regression when the ground truth coefficients in a linear model come from an anisotropic prior. We can cover this case with a simple extension of our results; see Appendix F.3.

Going beyond ridge regression, we anticipate that GCV for sketched ensembles should also be consistent for generalized linear models with arbitrary convex regularizers, as was recently shown in the unsketched setting for Gaussian data (Bellec, 2023). The key difficulty in applying the analysis based on Theorem 1 to the general setting is that we can only characterize the effect of sketching as additional ridge regularization. One promising path forward is via viewing the optimization as iteratively reweighted least squares (IRLS). On the regularization side, IRLS can achieve many types of structure-promoting regularizers (see LeJeune et al., 2021 and references therein) via successive generalized ridge, and so we might expect GCV to also be consistent in this case. Furthermore, for general training losses, we believe that GCV can be extended appropriately to handle reweighting of observations and leverage the classical connection between IRLS and maximum likelihood estimation in generalized linear models. Furthermore, to slightly relax data assumptions, we can extend GCV to the closely related approximate leave-one-out (ALO) risk estimation (Xu et al., 2019; Rad & Maleki, 2020), which relies on fewer concentration assumptions for consistency.

ACKNOWLEDGMENTS

We are grateful to Ryan J. Tibshirani for helpful feedback on this work. We warmly thank Benson Au, Roland Speicher, Dimitri Shlyakhtenko for insightful discussions related to free probability theory and infinitesimal freeness. We also warmly thank Arun Kumar Kuchibhotla, Alessandro Rinaldo, Yuting Wei, Jin-Hong Du, Alex Wei for many useful discussions regrading the "dual" aspects of observation subsampling in the context of risk monotonization. As is the nature of direction reversing and side flipping dualities in general, the insights and perspectives gained from that observation side are naturally "mirrored" and "transposed" onto this feature side (with some important caveats)! Finally, we sincerely thank the anonymous reviewers for their insightful and constructive feedback, which improved the manuscript, particularly with the addition of Appendix H.

This collaboration was partially supported by Office of Naval Research MURI grant N00014-20-1-2787. DL was supported by Army Research Office grant 2003514594.

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

# SUPPLEMENTARY MATERIAL

This serves as a supplement to the paper "Asymptotically Free Sketched Ridge Ensembles: Risks, Cross-Validation, and Tuning." Below we first provide an outline for the supplement in Table 1. Then we list some of the general and specific notations used throughout the main paper and the supplement in Tables 2 and 3, respectively.

## OUTLINE

| Appendix | Content |
|---|---|
| Appendix A | Background on asymptotic freeness and empirical support for sketching freeness |
| Appendix B | Asymptotic equivalents for freely sketched resolvents used in the proofs throughout |
| Appendix C | Proofs of Theorem 2 and Theorem 3 (from Section 3) |
| Appendix D | Proofs of Theorem 4 and Corollary 5 (from Section 4) |
| Appendix E | Proof of Proposition 6 (from Section 5) |
| Appendix F | Proof of Proposition 7 and statements and other details for anisotropic sketching, generalized ridge regression, and observation sketch (from Section 6) |
| Appendix G | Additional experimental illustrations and setup details for Figures 1 to 4 |
| Appendix H | Comprehensive complexity comparisons of various cross-validation methods |

Table 1: Roadmap of the supplement.

## GENERAL NOTATION

| Notation | Description |
|---|---|
| Non-bold | Denotes scalars, functions, distributions etc. (e.g., $k$, $f$, $P$) |
| Lowercase bold | Denotes vectors (e.g., $\mathbf{x}$, $\mathbf{y}$, $\boldsymbol{\beta}$) |
| Uppercase bold | Denotes matrices (e.g., $\mathbf{X}$, $\mathbf{S}$, $\boldsymbol{\Sigma}$) |
| $\mathbb{R}$, $\mathbb{R}_{\geqslant 0}$ | Set of real and non-negative real numbers |
| $\mathbb{C}$, $\mathbb{C}^+$, $\mathbb{C}^-$ | Set of complex numbers, and upper and lower complex half-planes |
| $[n]$ | Set $\{1, \ldots, n\}$ for a natural number $n$ |
| $\mathbb{1}\{A\}$ | Indicator random variable associated with an event $A$ |
| $\|\mathbf{u}\|_p$, $\|f\|_{L_p}$ | The $\ell_p$ norm of a vector $\mathbf{u}$ and the $L_p$ norm of a function $f$ for $p \geqslant 1$ |
| $\|\mathbf{X}\|_{\mathrm{op}}$, $\|\mathbf{X}\|_{\mathrm{tr}}$ | Operator (or spectral) and trace (or nuclear) norm of a rectangular matrix $\mathbf{X} \in \mathbb{R}^{n \times p}$ |
| $\mathrm{tr}[\mathbf{A}]$, $\mathbf{A}^{-1}$ | Trace and inverse (if invertible) of a square matrix $\mathbf{A} \in \mathbb{R}^{p \times p}$ |
| $\mathrm{rank}(\mathbf{B})$, $\mathbf{B}^\top$, $\mathbf{B}^\dagger$ | Rank, transpose and Moore-Penrose inverse of a rectangular matrix $\mathbf{B} \in \mathbb{R}^{n \times p}$ |
| $\mathbf{C}^{1/2}$ | Principal square root of a positive semidefinite matrix $\mathbf{C} \in \mathbb{R}^{p \times p}$ |
| $\mathbf{I}_n$ or $\mathbf{I}$ | The $n \times n$ identity matrix |
| $\mathcal{O}$, $o$ | Deterministic big-O and little-o notation |
| $\mathbf{u} \leqslant \mathbf{v}$ | Lexicographic ordering for real vectors $\mathbf{u}$ and $\mathbf{v}$ |
| $\mathbf{A} \preceq \mathbf{B}$ | Loewner ordering for symmetric matrices $\mathbf{A}$ and $\mathbf{B}$ |
| $\mathcal{O}_p$, $o_p$ | Probabilistic big-O and little-o notation |
| $\mathbf{A} \simeq \mathbf{B}$ | Asymptotic equivalence of matrices $\mathbf{A}$ and $\mathbf{B}$ (see Appendix B for details) |
| $\xrightarrow{\text{a.s.}}$, $\xrightarrow{\text{p}}$, $\xrightarrow{\text{d}}$ | Almost sure convergence, convergence in probability, and weak convergence |
| $\overset{2}{\Rightarrow}$ | Convergence in Wasserstein $W_2$ metric |

Table 2: Summary of the general notation used throughout the paper and the supplement.

SPECIFIC NOTATION

| Symbol | Meaning |
| --- | --- |
| $((\mathbf{x}_i, y_i))_{i=1}^n$ | Train dataset containing $n$ i.i.d. observations in $\mathbb{R}^p \times \mathbb{R}$ |
| $(\mathbf{X}, \mathbf{y})$ | Train data matrix $(\mathbf{x}_1, \ldots, \mathbf{x}_n)^\top$ in $\mathbb{R}^{n \times p}$ and response vector $\mathbf{y} = (y_1, \ldots, y_n)$ in $\mathbb{R}^n$ |
| $(\mathbf{x}_0, y_0)$ | Test point in $\mathbb{R}^p \times \mathbb{R}$ drawn independently from the train data distribution |
| $\mathbf{\Sigma}$ | Population covariance matrix in $\mathbb{R}^{p \times p}$: $\mathbb{E}[\mathbf{x}_0 \mathbf{x}_0^\top]$ |
| $\boldsymbol{\beta}_0$ | Coefficients of population linear projection of $y_0$ onto $\mathbf{x}_0$ in $\mathbb{R}^p$: $\mathbf{\Sigma}^{-1} \mathbb{E}[\mathbf{x}_0 y_0]$ |
| $\widehat{\mathbf{\Sigma}}$ | Sample covariance matrix in $\mathbb{R}^{p \times p}$: $\frac{1}{n} \mathbf{X}^\top \mathbf{X}$ |
| $\widehat{\boldsymbol{\beta}}_\lambda^{\mathrm{ridge}}$ | Ridge estimator on full data $(\mathbf{X}, \mathbf{y})$ at regularization level $\lambda$: $(\widehat{\mathbf{\Sigma}} + \lambda \mathbf{I}_p)^{-1} \frac{1}{n} \mathbf{X}^\top \mathbf{y}$ |
| $K$ | Ensemble size |
| $(\mathbf{S}_k)_{k=1}^K$ | Sketching matrices in $\mathbb{R}^{p \times q}$ (for feature sketch) |
| $\alpha$ | Sketching aspect ratio $\alpha = \frac{q}{p}$ |
| $\widehat{\boldsymbol{\beta}}_\lambda^{\mathbf{S}_k}$ | $k$-th component estimator in the sketched ensemble in $\mathbb{R}^q$ (sketch space) at regularization level $\lambda$ |
| $\widehat{\boldsymbol{\beta}}_\lambda^k$ | $k$-th component estimator in the sketched ensemble in $\mathbb{R}^p$ (feature space) |
| $\mathbf{L}_\lambda^k$ | Smoothing matrix of the $k$-th component estimator in the sketched ensemble in $\mathbb{R}^{n \times n}$ |
| $\widehat{\boldsymbol{\beta}}_\lambda^{\mathrm{ens}}$ | Final sketched ensemble estimator in $\mathbb{R}^p$: $\frac{1}{K} \sum_{k=1}^K \widehat{\boldsymbol{\beta}}_\lambda^k$ |
| $\mathbf{L}_\lambda^{\mathrm{ens}}$ | Smoothing matrix of the sketched ensemble estimator in $\mathbb{R}^{n \times n}$: $\frac{1}{K} \sum_{k=1}^K \mathbf{L}_\lambda^k$ |
| $P_\lambda^{\mathrm{ens}}$ | Joint distribution of test response and test predicted values of the sketched ensemble estimator (for feature sketch) at regularization level $\lambda$ |
| $R(\widehat{\boldsymbol{\beta}}_\lambda^{\mathrm{ens}})$ | Squared risk of the sketched ensemble estimator |
| $T(\widehat{\boldsymbol{\beta}}_\lambda^{\mathrm{ens}})$ | General linear risk functional of the sketched ensemble estimator |
| $\widehat{P}_\lambda^{\mathrm{ens}}$ | Estimated joint distribution of test response and test predicted values of the sketched ensemble (for feature sketch) at regularization level $\lambda$ using GCV residuals |
| $\widehat{R}(\widehat{\boldsymbol{\beta}}_\lambda^{\mathrm{ens}})$ | Estimated squared risk of the sketched ensemble estimator |
| $\widehat{T}(\widehat{\boldsymbol{\beta}}_\lambda^{\mathrm{ens}})$ | Estimated general linear functional of the sketched ensemble estimator |
| $\mu$ | Effective induced regularization level of the sketched ensemble estimator (for feature sketch) with original regularization level $\lambda$ |
| $\mu'$ | Inflation factor in the squared risk decomposition of the sketched ensemble estimator |
| $\mu''$ | Inflation factor in the GCV decomposition for the sketched ensemble estimator |
| $(\mathbf{T}_k)_{k=1}^K$ | Sketching matrices $\mathbb{R}^{n \times m}$ (for observation sketch) |
| $\eta$ | Sketching aspect ratio $\eta = \frac{m}{n}$ |
| $\widetilde{\boldsymbol{\beta}}_\lambda^k$ | $k$-th component estimator in the sketched ensemble in $\mathbb{R}^p$ (feature space) at regularization level $\lambda$ |
| $\widetilde{\mathbf{L}}_\lambda^k$ | Smoothing matrix of the $k$-th component estimator in the sketched ensemble in $\mathbb{R}^{n \times n}$ |
| $\widetilde{\boldsymbol{\beta}}_\lambda^{\mathrm{ens}}$ | Final sketched ensemble estimator in $\mathbb{R}^p$: $\frac{1}{K} \sum_{k=1}^K \widetilde{\boldsymbol{\beta}}_\lambda^k$ |
| $\widetilde{\mathbf{L}}_\lambda^{\mathrm{ens}}$ | Smoothing matrix of the sketched ensemble estimator in $\mathbb{R}^{n \times n}$: $\frac{1}{K} \sum_{k=1}^K \widetilde{\mathbf{L}}_\lambda^k$ |
| $R(\widetilde{\boldsymbol{\beta}}_\lambda^{\mathrm{ens}})$ | Squared risk of the sketched ensemble estimator (for observation sketch) at regularization level $\lambda$ |
| $\widetilde{R}(\widetilde{\boldsymbol{\beta}}_\lambda^{\mathrm{ens}})$ | Estimated squared risk of the sketched ensemble estimator using GCV |
| $\nu$ | Effective induced regularization level of the sketched ensemble estimator (for observation sketch) with original regularization level $\lambda$ |
| $\nu'$ | Inflation factor in the squared risk decomposition of the sketched ensemble estimator |
| $\nu''$ | Inflation factor in the GCV decomposition for the sketched ensemble estimator |
| $\mathscr{S}_{\mathbf{SS}^\top}$ | S-transform of the spectrum of $\mathbf{SS}^\top \in \mathbb{R}^{p \times p}$ (for feature sketch) |
| $\mathscr{S}_{\mathbf{TT}^\top}$ | S-transform of the spectrum of $\mathbf{TT}^\top \in \mathbb{R}^{n \times n}$ (for observation sketch) |

Table 3: Summary of the specific notation used throughout the paper and the supplement.

## A    BACKGROUND ON ASYMPTOTIC FREENESS AND FREE SKETCHING SUPPORT

Free probability (Voiculescu, 1997) is a mathematical framework that deals with non-commutative random variables. One of the key concepts in free probability is asymptotic freeness, which studies the behavior of random matrices in the limit as their dimension tends to infinity. This notion enables us to understand how independent random matrices become uncorrelated and behave as if they were freely independent in the high-dimensional limit. Good full-length references on free probability theory include: Mingo & Speicher (2017), Bose (2021). Chapters 2.4 and 2.5 from Tulino & Verdú (2004) and Tao (2023), respectively, are enjoyable introductions.

### A.1    FREE PROBABILITY THEORY

We begin with a few definitions from Mingo & Speicher (2017).

**Definition 8** ($C^*$-probability space and state). A pair $(\mathcal{A}, \varphi)$ is called a non-commutative $C^*$-*probability space* if $\mathcal{A}$ is a unital $C^*$-algebra and the linear functional $\varphi \colon \mathcal{A} \to \mathbb{C}$ is a unital *state*: i.e., $\varphi(1) = 1$ and $\varphi(a^*a) \geqslant 0$ for all $a \in \mathcal{A}$.

**Definition 9** (Freeness). Let $(\mathcal{A}, \varphi)$ be a $C^*$-probability space and let $(\mathcal{A}_1, \ldots, \mathcal{A}_s)$ be unital subalgebras of $\mathcal{A}$. Then $(\mathcal{A}_1, \ldots, \mathcal{A}_s)$ are *free* with respect to $\varphi$ if, for any $r \geqslant 2$ and $a_1, \ldots, a_r \in \mathcal{A}$ such that $\varphi(a_i) = 0$ for all $1 \leqslant i \leqslant r$ and $a_i \in \mathcal{A}_{j_i}$ for $j_i \neq j_{i+1}$ for all $1 \leqslant i \leqslant r - 1$, we have $\varphi(a_1 \cdots a_r) = 0$. Furthermore, we say that elements $a_1, \ldots, a_s \in \mathcal{A}$ are *free* with respect to $\varphi$ if the corresponding generated unital algebras $\mathcal{A}_1, \ldots, \mathcal{A}_s$ are free.

That is, we say that elements of the algebra are free if any alternating product of centered polynomials is also centered.

In this work, we will consider $\varphi$ to be the normalized trace—that is, the generalization of $\frac{1}{p}\mathrm{tr}[\mathbf{A}]$ for $\mathbf{A} \in \mathbb{C}^{p \times p}$ to elements of a $C^*$-algebra $\mathcal{A}$. Specifically, for any self-adjoint $a \in \mathcal{A}$ and any polynomial $p$,

$$\varphi(p(a)) = \int p(z) \, \mathrm{d}\mu_a(z),$$

where $\mu_a$ is the probability measure characterizing the spectral distribution of $a$.

**Definition 10** (Convergence in spectral distribution). Let $(\mathcal{A}, \varphi)$ be a $C^*$-probability space. We say that $\mathbf{A}_1, \ldots, \mathbf{A}_m \in \mathbb{C}^{p \times p}$ *converge in spectral distribution* to elements $a_1, \ldots, a_m \in \mathcal{A}$ if for all $1 \leqslant \ell < \infty$ and $1 \leqslant i_j \leqslant m$ for $1 \leqslant j \leqslant \ell$, we have

$$\tfrac{1}{p}\mathrm{tr}[\mathbf{A}_{i_1} \cdots \mathbf{A}_{i_\ell}] \to \varphi(a_{i_1} \cdots a_{i_\ell}).$$

One limitation of standard free probability theory is that it does not allow us to consider general expressions of the form $\mathrm{tr}[\mathbf{\Theta A}]$ when $\mathbf{\Theta}$ has bounded trace norm, as this would require us to use an unbounded operator $\widetilde{\mathbf{\Theta}} = p\mathbf{\Theta}$ to evaluate $\frac{1}{p}\mathrm{tr}[\widetilde{\mathbf{\Theta}}\mathbf{A}]$, but such an unbounded $\widetilde{\mathbf{\Theta}}$ cannot be an element of a $C^*$-algebra. However, evaluation of such expressions is possible with an extension called *infinitesimal* free probability (Shlyakhtenko, 2018), which is used in Theorem 1 from LeJeune et al. (2022) that our results build upon.

**Definition 11** (Infinitesimal freeness). Unital subalgebras $\mathcal{A}_1, \mathcal{A}_2 \subseteq \mathcal{A}$ are *infinitesimally free* with respect to $(\varphi, \varphi')$ if, for any $r \geqslant 2$ and $a_1, \ldots, a_r \in \mathcal{A}$ where $a_i \in \mathcal{A}_{j_i}$ for $j_i \neq j_{i+1}$ for all $1 \leqslant i \leqslant r - 1$, we have

$$\varphi_t((a_1 - \varphi_t(a_1)) \cdots (a_r - \varphi_t(a_r))) = o(t),$$

where $\varphi_t = \varphi + t\varphi'$.

We lastly introduce a series of invertible transformations for an element $a$ of a $C^*$-probability space:

$$G_a(z) = \varphi\big((z - a)^{-1}\big) \;\longleftrightarrow\; M_a(z) = \frac{1}{z}G_a\left(\frac{1}{z}\right) - 1 \;\longleftrightarrow\; \mathscr{S}_a(z) = \frac{1 + z}{z}M_a^{\langle -1 \rangle}(z),$$

which are the Cauchy transform (negative of the Stieltjes transform), moment generating series $M_a(z) = \sum_{k=1}^{\infty} \varphi(a^k)z^k$, and S-transform of $a$, respectively. Here $M_a^{\langle -1 \rangle}$ denotes inverse under composition of $M_a$.

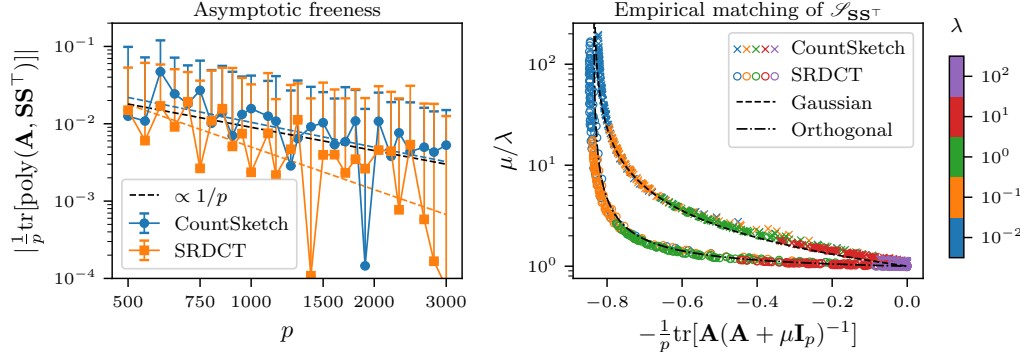

Figure A.1: **Empirical support for asymptotic freeness and subordination relation**. **Left:** We plot the absolute value of the average of the normalized traces of polynomials, which converge to zero. We also plot best fit lines on the log–log scale (dashed). Error bars denote one standard deviation over 10 trials, collected over both polynomials. **Right:** We numerically compute $\mu$ and plot the empirical subordination relation, which are decreasing continuous functions that closely match the theoretical S-transforms of Gaussian (dashed) for CountSketch ($\times$) and orthogonal (dash–dot) for SRDCT ($\circ$). Each mark in the scatter plots corresponds to a single $(\mathbf{A}, \lambda)$ pair, and we solve for the corresponding $\mu$.

## A.2 ASYMPTOTIC FREENESS

Freeness is characterized by a certain non-commutative centered alternating product condition (see Definition 9) with respect to a state function. With some slight abuse of notation, we consider the state function $\frac{1}{p}\mathrm{tr}[\cdot]$. Then two matrices $\mathbf{A}, \mathbf{B} \in \mathbb{R}^{p \times p}$ would be said to be free if

$$\frac{1}{p}\mathrm{tr}\left[\prod_{\ell=1}^{L}\mathrm{poly}_{\ell}^{\mathbf{A}}(\mathbf{A})\mathrm{poly}_{\ell}^{\mathbf{B}}(\mathbf{B})\right] = 0,$$

for all $L \geqslant 1$ and all centered polynomials—i.e., $\frac{1}{p}\mathrm{tr}[\mathrm{poly}_{\ell}^{\mathbf{A}}(\mathbf{A})] = 0$. The reason this is an abuse of notation is that finite matrices cannot satisfy this condition; however, they can satisfy it asymptotically as $p \to \infty$, and in this case we say that $\mathbf{A}$ and $\mathbf{B}$ are *asymptotically free*.

We test this property for CountSketch and SRDCT for polynomials of the form

$$\mathrm{poly}_{r}(\mathbf{A}) = \mathbf{A}^{r} - \frac{1}{p}\mathrm{tr}[\mathbf{A}^{r}]\mathbf{I}_{p}.$$

Specifically, we arbitrarily pick two choices

$$\mathrm{poly}(\mathbf{A}, \mathbf{B}) = \mathrm{poly}_{1}(\mathbf{A})\mathrm{poly}_{2}(\mathbf{B})\mathrm{poly}_{2}(\mathbf{A})\mathrm{poly}_{3}(\mathbf{B})$$

and

$$\mathrm{poly}(\mathbf{A}, \mathbf{B}) = \mathrm{poly}_{3}(\mathbf{A})\mathrm{poly}_{1}(\mathbf{B})\mathrm{poly}_{4}(\mathbf{A})\mathrm{poly}_{2}(\mathbf{B})$$

and evaluate $\frac{1}{p}\mathrm{tr}[\mathrm{poly}(\mathbf{A}, \mathbf{SS}^{\top})]$ for increasing $p$ over 10 trials, where $\mathbf{A}$ is a diagonal matrix with values linearly interpolating between 0.5 and 1.5 along the diagonal. As we see in Figure A.1 (left), for both sketches, this normalized trace is quite small and tending to zero. This strongly supports the assumption that CountSketch and SRDCT are both asymptotically free from diagonal matrices, and we expect the same to hold if $\mathbf{A}$ is rotated to be non-diagonal independently of the sampling of the sketching matrix $\mathbf{S}$.

## A.3 EMPIRICAL SUBORDINATION RELATIONS

### A.3.1 EXPERIMENTS ON SYNTHETIC DATASETS

Suppose $\hat{\mathbf{\Sigma}}$ and $\mathbf{SS}^{\top}$ are free and Theorem 1 holds. This means that a subordination relation via $\mathscr{S}_{\mathbf{SS}^{\top}}$ should characterize the implicit regularization, so we test this implication empirically as well. Specifically, without using any known form for $\mathscr{S}_{\mathbf{SS}^{\top}}$, we empirically verify that this mapping does not depend on $\mathbf{X}$ and compare it to known S-transforms.

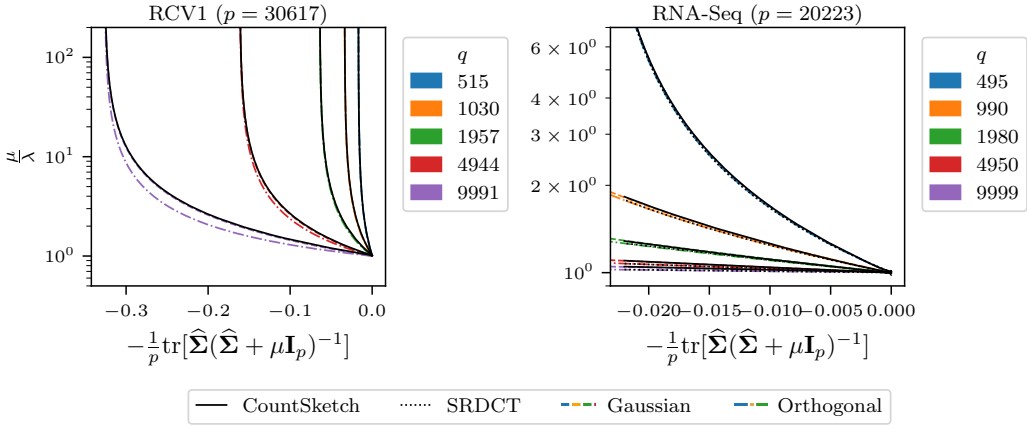

Figure A.2: **Empirical support for subordination relation with real data.** We numerically compute $\mu$ and plot the empirical subordination relation for sketches of real data (RCV1 (**left**) and RNA-Seq (**right**)) using CountSketch (black, solid) and SRDCT (black, dotted), which almost exactly match the theoretical S-transforms for Gaussian (dashed) and orthogonal (dash–dot) sketching, shown here for a single trial of random sketching for each value of $q$. As $q/p$ tends to zero, the subordination relation of all four sketches becomes indistinguishable.

As in the previous section, we will simplify our tests by considering a diagonal $\mathbf{A}$ instead of $\widehat{\mathbf{\Sigma}}$. We generate a family of $\mathbf{A} = \mathrm{diag}\,(\mathbf{a})$ parameterized by $a_0, s_0 > 0$, and $t_0 \in [0, 1]$ as

$$a_i = \frac{a_0}{1 - e^{-(t_i - t_0)/s_0}}, \quad \text{where} \quad t_i = \frac{i - 1}{p - 1}.$$

This family spans a variety of spectral distributions and provides a rich class of matrices over which Theorem 1 must hold simultaneously. For a fixed $700 \times 585$ sketching matrix $\mathbf{S}$ that we sample for CountSketch and for SRDCT, we sampled $\mathbf{A}$ over a $5 \times 5 \times 5$ grid of $a_0$ and $s_0$ logarithmically spaced between $0.1$ and $10$ and $t_0$ linearly spaced between $0$ and $1$. For each $\mathbf{A}$, we used numerical root finding to determine $\mu$ such that

$$\frac{1}{p}\mathrm{tr}\left[\mathbf{A}\mathbf{S}\big(\mathbf{S}^\top\mathbf{A}\mathbf{S} + \lambda\mathbf{I}_q\big)^{-1}\mathbf{S}^\top\right] = \frac{1}{p}\mathrm{tr}\left[\mathbf{A}\big(\mathbf{A} + \mu\mathbf{I}_p\big)^{-1}\right]$$

for each $\lambda \in \{0.01, 0.1, 1, 10, 100\}$. Then we construct a scatter plot of $\mu/\lambda$ and $\mu\frac{1}{p}\mathrm{tr}[(\mathbf{A} + \mu\mathbf{I}_p)^{-1}]$ in Figure A.1 (right), which should match $\mathscr{S}_{\mathbf{SS}^\top}$ and be a decreasing continuous function. We see that this is indeed the case, and furthermore by also plotting the known S-transform for Gaussian and orthogonal sketches from Table 4, we see that CountSketch matches the Gaussian function and SRDCT matches the orthogonal function.

### A.3.2 EXPERIMENTS ON REAL DATASETS

We repeat the experiment in Figure A.1 for real data in Figure A.2, using $\frac{1}{n}\mathbf{X}^\top\mathbf{X}$ instead of $\mathbf{A}$. For RCV1, since $\frac{1}{n}\mathbf{X}^\top\mathbf{X}$ is very high dimensional and matrix inversion is costly, we perform randomized trace estimation using the formula

$$\frac{1}{p}\mathrm{tr}\left[\frac{1}{n}\mathbf{X}^\top\mathbf{X}\big(\frac{1}{n}\mathbf{X}^\top\mathbf{X} + \mu\mathbf{I}_p\big)^{-1}\right] \approx \frac{1}{p}\mathbf{z}^\top\big(\frac{1}{n}\mathbf{X}^\top\mathbf{X} + \mu\mathbf{I}_p\big)^{-1}\frac{1}{n}\mathbf{X}^\top\mathbf{X}\mathbf{z},$$

where $\mathbf{z} \in \mathbb{R}^p$ is an i.i.d. Rademacher vector and the inversion is computed using the conjugate gradient method. Additionally, due to the size of RCV1, we only evaluate the subordination relation for CountSketch which can be efficiently applied due to the sparsity of the data, and do not evaluate the SRDCT. We precompute the traces for both sketched and unsketched sides of the subordination relation for a range of values of $\lambda$ and $\mu$ and then construct the mapping via linear interpolation. Since $n < p$ for RNA-Seq (see Appendix G.2, the normalized traces are upper bounded by $n/p = 446/20223 \approx 0.022$, which limits the operating range of the subordination relation and therefore the $x$-axis of the plot from $-0.022$ to $0$.

## A.4 KNOWN S-TRANSFORMS

We state some known S-transforms in the following table, where we let $\alpha = q/p$. We also assume that $\mathbf{S}$ is normalized such that $\mathbf{SS}^\top \simeq \mathbf{I}_p$, following LeJeune et al. (2022). For the i.i.d. sketch, this is simply the S-transform of the Marchenko–Pastur distribution, and for the orthogonal sketch, it is the S-transform of a binary distribution on $\left\{0, \frac{1}{\alpha}\right\}$. The identity sketch refers to simply $\mathbf{S} = \mathbf{I}_p$. There is currently no known S-transform for CountSketch, although our experiments in the previous sections suggest it is similar as for i.i.d. sketches.[3]

| Sketching family: | IID | Orthogonal, SRFT | Identity |
|---|---|---|---|
| $\mathscr{S}_{\mathbf{SS}^\top}(w)$: | $\frac{\alpha}{\alpha+w}$ | $\frac{\alpha(1+w)}{\alpha+w}$ | $1$ |

Table 4: Known S-transforms for normalized sketches

## B ASYMPTOTIC EQUIVALENTS FOR FREELY SKETCHED RESOLVENTS

In this section, we provide a brief background on the language of asymptotic equivalents used in the proofs throughout the paper. We will state the definition of asymptotic equivalents and point to useful calculus rules. For more details, see Dobriban & Wager (2018); Dobriban & Sheng (2021); LeJeune et al. (2022).

We use the language of asymptotic equivalents throughout the paper, defined formally as follows.

**Definition 12** (Asymptotic equivalence). Consider sequences $(\mathbf{A}_p)_{p \geqslant 1}$ and $(\mathbf{B}_p)_{p \geqslant 1}$ of (random or deterministic) matrices of growing dimension. We say that $\mathbf{A}_p$ and $\mathbf{B}_p$ are equivalent and write $\mathbf{A}_p \simeq \mathbf{B}_p$ if $\lim_{p \to \infty} |\mathrm{tr}[\mathbf{C}_p(\mathbf{A}_p - \mathbf{B}_p)]| = 0$ almost surely for any sequence $\mathbf{C}_p$ matrices with bounded trace norm such that $\lim \sup \|\mathbf{C}_p\|_{\mathrm{tr}} < \infty$ as $p \to \infty$.

The notion of deterministic equivalents obeys various calculus rules such as sum, product, differentiation, conditioning, substitution, among others. We refer readers to Patil et al. (2023) for a comprehensive list of these calculus rules, their proofs, and other related details.

### B.1 KNOWN ASYMPTOTIC EQUIVALENTS FOR ORDINARY RIDGE RESOLVENTS

In this section, we collect known asymptotic equivalents for the first- and second-order ordinary ridge resolvents. See Hastie et al. (2022); Patil et al. (2023) for more details.

**Lemma 13** (Deterministic equivalents for first-order and second-order ordinary ridge resolvents). *Suppose Assumption B holds. Then, for $\mu > -\lambda_{\min}^+(\frac{1}{n}\mathbf{X}^\top\mathbf{X})$, the following statements hold:*

1. *First-order ordinary ridge resolvent:*
$$\mu(\widehat{\mathbf{\Sigma}} + \mu\mathbf{I}_p)^{-1} \simeq (v\mathbf{\Sigma} + \mathbf{I}_p)^{-1}, \tag{B.1}$$
   *where $v^{-1}$ is the most positive solution to*
$$\mu = v^{-1} - \gamma\frac{1}{p}\mathrm{tr}[\mathbf{\Sigma}(v\mathbf{\Sigma} + \mathbf{I}_p)^{-1}]. \tag{B.2}$$

2. *Second-order ordinary ridge resolvent (population version):*
$$\mu^2(\widehat{\mathbf{\Sigma}} + \mu\mathbf{I}_p)^{-1}\mathbf{\Sigma}(\widehat{\mathbf{\Sigma}} + \mu\mathbf{I}_p)^{-1} \simeq (1 + \widetilde{v}_b)(v\mathbf{\Sigma} + \mathbf{I}_p)^{-1}\mathbf{\Sigma}(v\mathbf{\Sigma} + \mathbf{I}_p)^{-1}, \tag{B.3}$$
   *where $v$ as defined in (B.2), and $\widetilde{v}_b$ is defined in terms of $v$ by the equation*
$$\widetilde{v}_b = \frac{\gamma\frac{1}{p}\mathrm{tr}[\mathbf{\Sigma}^2(v\mathbf{\Sigma} + \mathbf{I}_p)^{-2}]}{v^{-2} - \gamma\frac{1}{p}\mathrm{tr}[\mathbf{\Sigma}^2(v\mathbf{\Sigma} + \mathbf{I}_p)^{-2}]}. \tag{B.4}$$

3. *Second-order ordinary ridge resolvent (empirical version):*
$$(\widehat{\mathbf{\Sigma}} + \mu\mathbf{I}_p)^{-1}\widehat{\mathbf{\Sigma}}(\widehat{\mathbf{\Sigma}} + \mu\mathbf{I}_p)^{-1} \simeq \widetilde{v}_v(v\mathbf{\Sigma} + \mathbf{I}_p)^{-1}\mathbf{\Sigma}(v\mathbf{\Sigma} + \mathbf{I}_p)^{-1}, \tag{B.5}$$
   *where $v$ is as defined in (B.2), and $\widetilde{v}_v$ is defined in terms of $v$ by the equation*
$$\widetilde{v}_v^{-1} = v^{-2} - \gamma\frac{1}{p}\mathrm{tr}[\mathbf{\Sigma}^2(v\mathbf{\Sigma} + \mathbf{I}_p)^{-2}]. \tag{B.6}$$

---

[3]See also the recent work Chenakkod et al. (2023) that show certain Gaussian universality results for sparse sketches like CountSketch.

## B.2 New asymptotic equivalents for freely sketched ridge resolvents

In this section, we derive first- and second-order equivalences for (both feature and observation) sketched resolvents. Their proofs are provided just after the statements.

**Lemma 14** (General first order equivalence for freely sketched ridge resolvents). *The following statements hold:*

1. *First-order sketched ridge resolvent (for feature sketch): Suppose Assumption A holds for* $\mathbf{S}\mathbf{S}^\top$. *Then, for all* $\lambda > \lambda_0$,

$$\mathbf{S}(\mathbf{S}^\top \tfrac{1}{n}\mathbf{X}^\top\mathbf{X}\mathbf{S} + \lambda\mathbf{I}_q)^{-1}\mathbf{S}^\top \simeq (\tfrac{1}{n}\mathbf{X}^\top\mathbf{X} + \mu\mathbf{I}_p)^{-1}, \tag{B.7}$$

   *where* $\mu$ *solves*

$$\mu = \lambda\mathscr{S}_{\mathbf{S}\mathbf{S}^\top}\big(-\tfrac{1}{p}\mathrm{tr}\big[\tfrac{1}{n}\mathbf{X}^\top\mathbf{X}\big(\tfrac{1}{n}\mathbf{X}^\top\mathbf{X} + \mu\mathbf{I}_p\big)^{-1}\big]\big). \tag{B.8}$$

2. *First-order sketched ridge resolvent (for observation sketch): Suppose Assumption A holds for* $\mathbf{T}\mathbf{T}^\top$. *Then, for all* $\lambda > \widetilde{\lambda}_0$,

$$(\tfrac{1}{n}\mathbf{X}^\top\mathbf{T}\mathbf{T}^\top\mathbf{X} + \lambda\mathbf{I}_p)^{-1}\mathbf{X}^\top\mathbf{T}\mathbf{T}^\top \simeq \mathbf{X}^\top(\tfrac{1}{n}\mathbf{X}\mathbf{X}^\top + \nu\mathbf{I}_n)^{-1}, \tag{B.9}$$

   *where* $\nu$ *solves*

$$\nu = \lambda\mathscr{S}_{\mathbf{T}\mathbf{T}^\top}\big(-\tfrac{1}{n}\mathrm{tr}\big[\tfrac{1}{n}\mathbf{X}\mathbf{X}^\top\big(\tfrac{1}{n}\mathbf{X}\mathbf{X}^\top + \nu\mathbf{I}_n\big)^{-1}\big]\big). \tag{B.10}$$

**Lemma 15** (General second order equivalence for freely sketched ridge resolvents). *Under the settings of Lemma 14, for any positive semidefinite* $\mathbf{\Psi}$ *with uniformly bounded operator norm, the following statements hold:*

1. *Second-order sketched ridge resolvent (for feature sketch): For all* $\lambda > \lambda_0$,

$$\mathbf{S}\big(\mathbf{S}^\top \tfrac{1}{n}\mathbf{X}^\top\mathbf{X}\mathbf{S} + \lambda\mathbf{I}_q\big)^{-1}\mathbf{S}^\top\mathbf{\Psi}\mathbf{S}\big(\mathbf{S}^\top \tfrac{1}{n}\mathbf{X}^\top\mathbf{X}\mathbf{S} + \lambda\mathbf{I}_q\big)^{-1}\mathbf{S}^\top$$
$$\simeq \big(\tfrac{1}{n}\mathbf{X}^\top\mathbf{X} + \mu\mathbf{I}_p\big)^{-1}\big(\mathbf{\Psi} + \mu'_{\mathbf{\Psi}}\mathbf{I}_p\big)\big(\tfrac{1}{n}\mathbf{X}^\top\mathbf{X} + \mu\mathbf{I}_p\big)^{-1}, \tag{B.11}$$

   *where* $\mu'_{\mathbf{\Psi}} \geqslant 0$ *is given by:*

$$\mu'_{\mathbf{\Psi}} = -\frac{\partial\mu}{\partial\lambda}\lambda^2\mathscr{S}'_{\mathbf{S}\mathbf{S}^\top}\big(-\tfrac{1}{p}\mathrm{tr}\big[\tfrac{1}{n}\mathbf{X}^\top\mathbf{X}\big(\tfrac{1}{n}\mathbf{X}^\top\mathbf{X} + \mu\mathbf{I}_p\big)^{-1}\big]\big)$$
$$\cdot \tfrac{1}{p}\mathrm{tr}\big[\mathbf{\Psi}\big(\tfrac{1}{n}\mathbf{X}^\top\mathbf{X} + \mu\mathbf{I}_p\big)^{-2}\big]. \tag{B.12}$$

2. *Second-order sketched ridge resolvent (for observation sketch): For all* $\lambda > \widetilde{\lambda}_0$,

$$\tfrac{1}{n}\mathbf{T}\mathbf{T}^\top\mathbf{X}\big(\tfrac{1}{n}\mathbf{X}^\top\mathbf{T}\mathbf{T}^\top\mathbf{X} + \lambda\mathbf{I}_p\big)^{-1}\mathbf{\Psi}\big(\tfrac{1}{n}\mathbf{X}^\top\mathbf{T}\mathbf{T}^\top\mathbf{X} + \lambda\mathbf{I}_p\big)^{-1}\mathbf{X}^\top\mathbf{T}\mathbf{T}^\top$$
$$\simeq \big(\tfrac{1}{n}\mathbf{X}\mathbf{X}^\top + \nu\mathbf{I}_n\big)^{-1}\big(\tfrac{1}{n}\mathbf{X}\mathbf{\Psi}\mathbf{X}^\top + \nu'_{\mathbf{\Psi}}\mathbf{I}_n\big)\big(\tfrac{1}{n}\mathbf{X}\mathbf{X}^\top + \nu\mathbf{I}_n\big)^{-1}, \tag{B.13}$$

   *where* $\nu'_{\mathbf{\Psi}} \geqslant 0$ *is given by:*

$$\nu'_{\mathbf{\Psi}} = -\frac{\partial\mu}{\partial\lambda}\lambda^2\mathscr{S}'_{\mathbf{T}\mathbf{T}^\top}\big(-\tfrac{1}{n}\mathrm{tr}\big[\tfrac{1}{n}\mathbf{X}\mathbf{X}^\top\big(\tfrac{1}{n}\mathbf{X}\mathbf{X}^\top + \nu\mathbf{I}_n\big)^{-1}\big]\big)$$
$$\cdot \tfrac{1}{p}\mathrm{tr}\big[\tfrac{1}{n}\mathbf{X}\mathbf{\Psi}\mathbf{X}^\top\big(\tfrac{1}{n}\mathbf{X}\mathbf{X}^\top + \nu\mathbf{I}_n\big)^{-2}\big]. \tag{B.14}$$

*Proof of Lemma 14.* There are two cases two show.

1. The statement for the feature sketch follows from Theorem 1.

2. For the statement for the observation sketch, we use the Woodbury matrix identity to write

$$(\tfrac{1}{n}\mathbf{X}^\top\mathbf{T}\mathbf{T}^\top\mathbf{X} + \lambda\mathbf{I}_p)^{-1}\mathbf{X}^\top\mathbf{T}\mathbf{T}^\top = \mathbf{X}^\top\mathbf{T}(\tfrac{1}{n}\mathbf{T}^\top\mathbf{X}\mathbf{X}^\top\mathbf{T} + \lambda\mathbf{I}_m)^{-1}\mathbf{T}^\top.$$

   Now we can use the result from feature sketch with $\mathbf{T}$ playing the role of $\mathbf{S}$ and $\mathbf{X}^\top$ playing the role of $\mathbf{X}$.

This completes the two cases and finishes the proof. $\qquad\square$

*Proof of Lemma 15.* There are again two cases to prove.

1. We begin with feature sketch. Let $\mathbf{A}_z = \frac{1}{n}\mathbf{X}^\top\mathbf{X} + z\boldsymbol{\Psi}$ with corresponding $\mu_z$ from Assumption A. Following the same strategy as the proof of (LeJeune et al., 2022, Theorem 4.8), the two sides of (B.11) are equal to

$$-\frac{\partial}{\partial z}\mathbf{S}\big(\mathbf{S}^\top\mathbf{A}_z\mathbf{S} + \lambda\mathbf{I}_q\big)^{-1}\mathbf{S}^\top \simeq -\frac{\partial}{\partial z}\big(\mathbf{A}_z + \mu_z\mathbf{I}_p\big)^{-1}$$

at $z = 0$, and therefore $\mu' = \partial\mu_z/\partial z$ at $z = 0$. Letting

$$\mathscr{S}' = \mathscr{S}'_{\mathbf{S}\mathbf{S}^\top}\big(-\tfrac{1}{p}\mathrm{tr}\big[\tfrac{1}{n}\mathbf{X}^\top\mathbf{X}\big(\tfrac{1}{n}\mathbf{X}^\top\mathbf{X} + \mu\mathbf{I}_p\big)^{-1}\big]\big)$$

for brevity, and noting that

$$-\mathbf{A}_z\big(\mathbf{A}_z + \mu_z\mathbf{I}_p\big)^{-1} = \mu_z\big(\mathbf{A}_z + \mu_z\mathbf{I}_p\big)^{-1} - 1,$$

we differentiate (6) to obtain for $z = 0$

$$\mu' = \lambda\mathscr{S}' \cdot \Big(\mu'\tfrac{1}{p}\mathrm{tr}\big[\big(\tfrac{1}{n}\mathbf{X}^\top\mathbf{X} + \mu\mathbf{I}_p\big)^{-1}\big] - \mu\tfrac{1}{p}\mathrm{tr}\big[\big(\tfrac{1}{n}\mathbf{X}^\top\mathbf{X} + \mu\mathbf{I}_p\big)^{-2}(\boldsymbol{\Psi} + \mu'\mathbf{I}_p)\big]\Big).$$

Solving for $\mu'$, we get

$$\mu' = \frac{-\lambda\mu\mathscr{S}'\tfrac{1}{p}\mathrm{tr}\big[\boldsymbol{\Psi}\big(\tfrac{1}{n}\mathbf{X}^\top\mathbf{X} + \mu\mathbf{I}_p\big)^{-2}\big]}{\lambda\mu\mathscr{S}'\tfrac{1}{p}\mathrm{tr}\big[\big(\tfrac{1}{n}\mathbf{X}^\top\mathbf{X} + \mu\mathbf{I}_p\big)^{-2}\big] - \lambda\mathscr{S}'\tfrac{1}{p}\mathrm{tr}\big[\big(\tfrac{1}{n}\mathbf{X}^\top\mathbf{X} + \mu\mathbf{I}_p\big)^{-1}\big] + 1}.$$

Meanwhile, if we take partial derivatives with respect to $\lambda$ (after dividing by $\lambda$ on both sides),

$$\frac{\partial\mu}{\partial\lambda}\frac{1}{\lambda} - \frac{\mu}{\lambda^2} = \mathscr{S}' \cdot \Big(\tfrac{1}{p}\mathrm{tr}\big[\big(\tfrac{1}{n}\mathbf{X}^\top\mathbf{X} + \mu\mathbf{I}_p\big)^{-1}\big] - \mu\tfrac{1}{p}\mathrm{tr}\big[\big(\tfrac{1}{n}\mathbf{X}^\top\mathbf{X} + \mu\mathbf{I}_p\big)^{-2}\big]\Big)\frac{\partial\mu}{\partial\lambda}.$$

Combining these two equations gives the stated result for the feature sketch.

2. For the observation sketch, we once again use the Woodbury matrix identity to write

$$\tfrac{1}{n}\mathbf{T}\mathbf{T}^\top\mathbf{X}\big(\tfrac{1}{n}\mathbf{X}^\top\mathbf{T}\mathbf{T}^\top\mathbf{X} + \lambda\mathbf{I}_p\big)^{-1}\boldsymbol{\Psi}\big(\tfrac{1}{n}\mathbf{X}^\top\mathbf{T}\mathbf{T}^\top\mathbf{X} + \lambda\mathbf{I}_p\big)^{-1}\mathbf{X}^\top\mathbf{T}\mathbf{T}^\top$$
$$= \tfrac{1}{n}\mathbf{T}\big(\tfrac{1}{n}\mathbf{T}^\top\mathbf{X}\mathbf{X}^\top\mathbf{T} + \lambda\mathbf{I}_m\big)^{-1}\mathbf{T}^\top\mathbf{X}\boldsymbol{\Psi}\mathbf{X}^\top\mathbf{T}\big(\tfrac{1}{n}\mathbf{T}^\top\mathbf{X}\mathbf{X}^\top\mathbf{T} + \lambda\mathbf{I}_m\big)^{-1}\mathbf{T}^\top.$$

The equivalence in (B.13) and the inflation parameter in (B.14) now follow from the second-order result for feature sketch by substituting $\mathbf{T}$ for $\mathbf{S}$, $\mathbf{X}$ for $\mathbf{X}^\top$, and $\frac{1}{n}\mathbf{X}\boldsymbol{\Psi}\mathbf{X}^\top$ for $\boldsymbol{\Psi}$ in (B.11).

This finishes the two cases and concludes the proof. $\qquad\square$

## C PROOFS IN SECTION 3

### C.1 PROOF OF THEOREM 2

Below we first provide the complete statement of Theorem 2, which includes expressions for $\mu'$ and $\mu''$ that are excluded from the main paper.

**Theorem 16** (Squared risk and GCV asymptotics for feature sketch)**.** *Suppose Assumption A hold. Then, for all $\lambda > \lambda_0 = -\liminf_{p\to\infty}\min_{k\in[K]}\lambda_{\min}^+(\mathbf{S}_k^\top\widehat{\boldsymbol{\Sigma}}\mathbf{S}_k)$ and all $K$,*

$$R\big(\widehat{\boldsymbol{\beta}}_\lambda^{\mathrm{ens}}\big) \simeq R\big(\widehat{\boldsymbol{\beta}}_\mu^{\mathrm{ridge}}\big) + \frac{\mu'\Delta}{K} \quad and \quad \widehat{R}\big(\widehat{\boldsymbol{\beta}}_\lambda^{\mathrm{ens}}\big) \simeq \widehat{R}\big(\widehat{\boldsymbol{\beta}}_\mu^{\mathrm{ridge}}\big) + \frac{\mu''\Delta}{K}, \tag{C.1}$$

*where $\mu$ is an implicit regularization parameter that solves (B.8), $\Delta = \frac{1}{n}\mathbf{y}^\top\big(\frac{1}{n}\mathbf{X}\mathbf{X}^\top + \mu\mathbf{I}_n\big)^{-2}\mathbf{y}$, and $\mu' \geqslant 0$ is an inflation factor in the risk decomposition given by:*

$$\mu' = -\frac{\partial\mu}{\partial\lambda}\lambda^2\mathscr{S}'_{\mathbf{S}\mathbf{S}^\top}\big(-\tfrac{1}{p}\mathrm{tr}\big[\tfrac{1}{n}\mathbf{X}^\top\mathbf{X}\big(\tfrac{1}{n}\mathbf{X}^\top\mathbf{X} + \mu\mathbf{I}_p\big)^{-1}\big]\big)\tfrac{1}{p}\mathrm{tr}\big[\boldsymbol{\Sigma}\big(\tfrac{1}{n}\mathbf{X}^\top\mathbf{X} + \mu\mathbf{I}_p\big)^{-2}\big], \tag{C.2}$$

*while $\mu'' \geqslant 0$ is an inflation factor in the GCV decomposition given by:*

$$\mu'' = \frac{-\frac{\partial \mu}{\partial \lambda} \lambda^2 \mathscr{S}'_{\mathbf{SS}^\top}\left(-\frac{1}{p}\mathrm{tr}\left[\frac{1}{n}\mathbf{X}^\top\mathbf{X}\left(\frac{1}{n}\mathbf{X}^\top\mathbf{X} + \mu\mathbf{I}_p\right)^{-1}\right]\right)\frac{1}{p}\mathrm{tr}\left[\frac{1}{n}\mathbf{X}^\top\mathbf{X}\left(\frac{1}{n}\mathbf{X}^\top\mathbf{X} + \mu\mathbf{I}_p\right)^{-2}\right]}{\left(1 - \frac{1}{n}\mathrm{tr}\left[\frac{1}{n}\mathbf{X}^\top\mathbf{X}\left(\frac{1}{n}\mathbf{X}^\top\mathbf{X} + \mu\mathbf{I}_p\right)^{-1}\right]\right)^2}. \quad \text{(C.3)}$$

*Proof.* The core component of the proof is Lemma 17. We shall break down the proof into two parts: risk asymptotics and GCV asymptotics. Before proceeding, let us introduce some essential notation first.

*Notation*: We decompose the unknown response $y_0$ into its linear predictor and residual. Specifically, let $\boldsymbol{\beta}_0$ be the optimal projection parameter given by $\boldsymbol{\beta}_0 = \boldsymbol{\Sigma}^{-1}\mathbb{E}[\mathbf{x}_0 y_0]$. Then, we can express the response as a sum of its best linear predictor, $\mathbf{x}^\top\boldsymbol{\beta}_0$, and the residual, $y_0 - \mathbf{x}_0^\top\boldsymbol{\beta}_0$. Denote the variance of this residual by $\sigma^2 = \mathbb{E}[(y_0 - \mathbf{x}_0^\top\boldsymbol{\beta}_0)^2]$.

**Part 1: Risk asymptotics.** It is easy to see that the risk decomposes as follows:

$$R(\widehat{\boldsymbol{\beta}}_\lambda^{\mathrm{ens}}) = \mathbb{E}\big[(y_0 - \mathbf{x}_0^\top\widehat{\boldsymbol{\beta}}_\lambda^{\mathrm{ens}})^2 \mid \mathbf{X}, \mathbf{y}, (\mathbf{S}_k)_{k=1}^k\big] = (\widehat{\boldsymbol{\beta}}_\lambda^{\mathrm{ens}} - \boldsymbol{\beta}_0)^\top\boldsymbol{\Sigma}(\widehat{\boldsymbol{\beta}}_\lambda^{\mathrm{ens}} - \boldsymbol{\beta}_0) + \sigma^2.$$

Here, we used the fact that $(y_0 - \mathbf{x}_0^\top\boldsymbol{\beta}_0)$ is uncorrelated with $\mathbf{x}_0$, that is $\mathbb{E}[\mathbf{x}_0(y_0 - \mathbf{x}_0^\top\boldsymbol{\beta}_0)] = \mathbf{0}_p$. We note that $\|\beta_0\|_2 < \infty$ and $\boldsymbol{\Sigma}$ has uniformly bounded operator norm from Assumption B. Applying Lemma 17 then yields:

$$R(\widehat{\boldsymbol{\beta}}_\lambda^{\mathrm{ens}}) = (\widehat{\boldsymbol{\beta}}_\lambda^{\mathrm{ens}} - \boldsymbol{\beta}_0)^\top\boldsymbol{\Sigma}(\widehat{\boldsymbol{\beta}}_\lambda^{\mathrm{ens}} - \boldsymbol{\beta}_0) + \sigma^2$$

$$\simeq (\widehat{\boldsymbol{\beta}}_\mu^{\mathrm{ridge}} - \boldsymbol{\beta}_0)^\top\boldsymbol{\Sigma}(\widehat{\boldsymbol{\beta}}_\mu^{\mathrm{ridge}} - \boldsymbol{\beta}_0) + \sigma^2 + \frac{\mu'\Delta}{K}$$

$$= R(\widehat{\boldsymbol{\beta}}_\mu^{\mathrm{ridge}}) + \frac{\mu'\Delta}{K},$$

where $\mu'$ is as defined in (C.2). This completes the first part of the proof for the risk asymptotics decomposition.

**Part 2: GCV asymptotics.** We will work on the numerator and denominator asymptotics separately, and combine them to get the final expression for the GCV asymptotics.

*Numerator:* We start with the numerator. Similar decomposition as the risk yields

$$\frac{1}{n}\|\mathbf{y} - \mathbf{X}\widehat{\boldsymbol{\beta}}_\lambda^{\mathrm{ens}}\|_2^2$$
$$= \frac{1}{n}\|\mathbf{X}(\boldsymbol{\beta}_0 - \widehat{\boldsymbol{\beta}}_\lambda^{\mathrm{ens}}) + (\mathbf{y} - \mathbf{X}\boldsymbol{\beta}_0)\|_2^2$$
$$= \frac{1}{n}\|\mathbf{X}(\boldsymbol{\beta}_0 - \widehat{\boldsymbol{\beta}}_\lambda^{\mathrm{ens}})\|_2^2 + \frac{1}{n}\|(\mathbf{y} - \mathbf{X}\boldsymbol{\beta}_0)\|_2^2 + \frac{2}{n}(\mathbf{y} - \mathbf{X}\boldsymbol{\beta}_0)^\top\mathbf{X}(\boldsymbol{\beta}_0 - \widehat{\boldsymbol{\beta}}_\lambda^{\mathrm{ens}}).$$

From Lemma 14, note that

$$\frac{2}{n}(\mathbf{y} - \mathbf{X}\boldsymbol{\beta}_0)^\top\mathbf{X}(\boldsymbol{\beta}_0 - \widehat{\boldsymbol{\beta}}_\lambda^{\mathrm{ens}}) \simeq \frac{2}{n}(\mathbf{y} - \mathbf{X}\boldsymbol{\beta}_0)^\top\mathbf{X}(\boldsymbol{\beta}_0 - \widehat{\boldsymbol{\beta}}_\mu^{\mathrm{ridge}}).$$

Next we expand

$$\frac{1}{n}\|\mathbf{X}(\boldsymbol{\beta}_0 - \widehat{\boldsymbol{\beta}}_\lambda^{\mathrm{ens}})\|_2^2 = (\widehat{\boldsymbol{\beta}}_\lambda^{\mathrm{ens}} - \boldsymbol{\beta}_0)^\top\frac{1}{n}\mathbf{X}^\top\mathbf{X}(\widehat{\boldsymbol{\beta}}_\lambda^{\mathrm{ens}} - \boldsymbol{\beta}_0)$$

$$\simeq (\widehat{\boldsymbol{\beta}}_\mu^{\mathrm{ridge}} - \boldsymbol{\beta}_0)^\top\frac{1}{n}\mathbf{X}^\top\mathbf{X}(\widehat{\boldsymbol{\beta}}_\mu^{\mathrm{ridge}} - \boldsymbol{\beta}_0) + \frac{\mu''_{\mathrm{num}}\Delta}{K},$$

where $\mu''_{\mathrm{num}}$ is given by:

$$\mu''_{\mathrm{num}} = -\frac{\partial \mu}{\partial \lambda} \lambda^2 \mathscr{S}'_{\mathbf{SS}^\top}\left(-\frac{1}{p}\mathrm{tr}\left[\frac{1}{n}\mathbf{X}^\top\mathbf{X}\left(\frac{1}{n}\mathbf{X}^\top\mathbf{X} + \mu\mathbf{I}_p\right)^{-1}\right]\right)\frac{1}{p}\mathrm{tr}\left[\frac{1}{n}\mathbf{X}^\top\mathbf{X}\left(\frac{1}{n}\mathbf{X}^\top\mathbf{X} + \mu\mathbf{I}_p\right)^{-2}\right].$$

Now appealing to Lemma 17, we have

$$\frac{1}{n}\|\mathbf{y} - \mathbf{X}\widehat{\boldsymbol{\beta}}_\lambda^{\mathrm{ens}}\|_2^2 \simeq (\widehat{\boldsymbol{\beta}}_\mu^{\mathrm{ridge}} - \boldsymbol{\beta}_0)^\top\frac{1}{n}\mathbf{X}^\top\mathbf{X}(\widehat{\boldsymbol{\beta}}_\mu^{\mathrm{ridge}} - \boldsymbol{\beta}_0) + \frac{2}{n}(\mathbf{y} - \mathbf{X}\boldsymbol{\beta}_0)^\top\mathbf{X}(\boldsymbol{\beta}_0 - \widehat{\boldsymbol{\beta}}_\mu^{\mathrm{ridge}})$$

$$+ \frac{1}{n}\|(\mathbf{y} - \mathbf{X}\boldsymbol{\beta}_0)\|_2^2 + \frac{\mu''_{\mathrm{num}}\Delta}{K}. \quad \text{(C.4)}$$

Note also that

$$\frac{1}{n}\|\mathbf{y} - \mathbf{X}\widehat{\boldsymbol{\beta}}_\mu^{\mathrm{ridge}}\|_2^2$$

$$= \frac{1}{n}\|(\mathbf{y} - \mathbf{X}\boldsymbol{\beta}_0) + \mathbf{X}(\boldsymbol{\beta}_0 - \widehat{\boldsymbol{\beta}}_\mu^{\mathrm{ridge}})\|_2^2$$

$$= \frac{1}{n}\|\mathbf{y} - \mathbf{X}\boldsymbol{\beta}_0\|_2^2 + \frac{2}{n}(\mathbf{y} - \mathbf{X}\boldsymbol{\beta}_0)^\top \mathbf{X}(\boldsymbol{\beta}_0 - \widehat{\boldsymbol{\beta}}_\mu^{\mathrm{ridge}}) + \frac{1}{n}\|\mathbf{X}(\boldsymbol{\beta}_0 - \widehat{\boldsymbol{\beta}}_\mu^{\mathrm{ridge}})\|_2^2. \qquad (\mathrm{C}.5)$$

Combining (C.4) and (C.5), we arrive at the following asymptotic decomposition for the numerator:

$$\frac{1}{n}\|\mathbf{y} - \mathbf{X}\widehat{\boldsymbol{\beta}}_\lambda^{\mathrm{ens}}\|_2^2 \simeq \frac{1}{n}\|\mathbf{y} - \mathbf{X}\widehat{\boldsymbol{\beta}}_\mu^{\mathrm{ridge}}\|_2^2 + \frac{\mu_{\mathrm{num}}''\Delta}{K}. \qquad (\mathrm{C}.6)$$

*Denominator:* Next we work on the denominator. For the ensemble smoothing matrix, observe that

$$\mathbf{L}_\lambda^{\mathrm{ens}} = \frac{1}{K}\sum_{k=1}^K \frac{1}{n}\mathbf{X}\mathbf{S}_k(\frac{1}{n}\mathbf{S}_k^\top \mathbf{X}^\top \mathbf{X}\mathbf{S}_k + \lambda\mathbf{I}_q)^{-1}\mathbf{S}_k^\top \mathbf{X}^\top \simeq \frac{1}{n}\mathbf{X}(\frac{1}{n}\mathbf{X}^\top \mathbf{X} + \mu\mathbf{I}_p)^{-1}\mathbf{X}^\top,$$

where we used Lemma 14 to write the asymptotic equivalence in the last line. Thus, we have

$$\mathrm{tr}[\mathbf{L}_\lambda^{\mathrm{ens}}] \simeq \mathrm{tr}[(\frac{1}{n}\mathbf{X}^\top \mathbf{X} + \mu\mathbf{I}_p)^{-1}\frac{1}{n}\mathbf{X}^\top \mathbf{X}] = \mathrm{tr}[\mathbf{L}_\mu^{\mathrm{ridge}}]. \qquad (\mathrm{C}.7)$$

Therefore, combining (C.6) and (C.7), for the GCV estimator, we obtain

$$\widehat{R}(\widehat{\boldsymbol{\beta}}_\lambda^{\mathrm{ens}}) = \frac{\frac{1}{n}\|\mathbf{y} - \mathbf{X}\widehat{\boldsymbol{\beta}}_\lambda^{\mathrm{ens}}\|_2^2}{(1 - \frac{1}{n}\mathrm{tr}[\mathbf{L}_\lambda^{\mathrm{ens}}])^2}$$

$$\simeq \frac{\frac{1}{n}\|\mathbf{y} - \mathbf{X}\widehat{\boldsymbol{\beta}}_\mu^{\mathrm{ridge}}\|_2^2 + \dfrac{\mu_{\mathrm{num}}''\Delta}{K}}{(1 - \frac{1}{n}\mathrm{tr}[\mathbf{L}_\lambda^{\mathrm{ens}}])^2}$$

$$\simeq \frac{\frac{1}{n}\|\mathbf{y} - \mathbf{X}\widehat{\boldsymbol{\beta}}_\mu^{\mathrm{ridge}}\|_2^2 + \dfrac{\mu_{\mathrm{num}}''\Delta}{K}}{(1 - \frac{1}{n}\mathrm{tr}[\mathbf{L}_\mu^{\mathrm{ridge}}])^2}$$

$$= \widehat{R}(\widehat{\boldsymbol{\beta}}_\mu^{\mathrm{ridge}}) + \frac{\mu''\Delta}{K},$$

where $\mu''$ is as defined in (C.3). This finishes the second part of the proof for the GCV asymptotics decomposition and concludes the proof. $\qquad\square$

HELPER LEMMA FOR THE PROOF OF THEOREM 2

**Lemma 17** (Quadratic risk decomposition for the ensemble estimator for feature sketch)**.** *Assume the conditions of Lemma 15. Let $\boldsymbol{\Psi}$ be any positive semidefinite matrix with uniformly bounded operator norm, that is independent of $(\mathbf{S}_k)_{k=1}^K$. Let $\boldsymbol{\beta}_0 \in \mathbb{R}^p$ be any vector with uniformly bounded Euclidean norm, that is independent of $(\mathbf{S}_k)_{k=1}^K$. Then, under Assumptions A and B, for $\lambda > \lambda_0$ and all $K$,*

$$(\widehat{\boldsymbol{\beta}}_\lambda^{\mathrm{ens}} - \boldsymbol{\beta}_0)^\top \boldsymbol{\Psi}(\widehat{\boldsymbol{\beta}}_\lambda^{\mathrm{ens}} - \boldsymbol{\beta}_0) \simeq (\widehat{\boldsymbol{\beta}}_\mu^{\mathrm{ridge}} - \boldsymbol{\beta}_0)^\top \boldsymbol{\Psi}(\widehat{\boldsymbol{\beta}}_\mu^{\mathrm{ridge}} - \boldsymbol{\beta}_0) + \frac{\mu_\Psi'\Delta}{K},$$

*where $\mu$ is as defined in (B.8), $\mu_\Psi'$ is as defied in (B.12), and $\Delta$ is as defined in Theorem 16.*

*Proof.* We start with a decomposition. Observe that

$$(\widehat{\boldsymbol{\beta}}_\lambda^{\mathrm{ens}} - \boldsymbol{\beta}_0)^\top \boldsymbol{\Psi}(\widehat{\boldsymbol{\beta}}_\lambda^{\mathrm{ens}} - \boldsymbol{\beta}_0)$$

$$= \left(\frac{1}{K}\sum_{k=1}^K \widehat{\boldsymbol{\beta}}_\lambda^k - \boldsymbol{\beta}_0\right)^\top \boldsymbol{\Psi}\left(\frac{1}{K}\sum_{k=1}^K \widehat{\boldsymbol{\beta}}_\lambda^k - \boldsymbol{\beta}_0\right)$$

$$= \frac{1}{K^2}\sum_{k,\ell=1}^K (\widehat{\boldsymbol{\beta}}_\lambda^k)^\top \boldsymbol{\Psi}\widehat{\boldsymbol{\beta}}_\lambda^\ell - \frac{2}{K}\sum_{k=1}^K \boldsymbol{\beta}_0^\top \boldsymbol{\Sigma}\widehat{\boldsymbol{\beta}}_\lambda^k + \boldsymbol{\beta}_0^\top \boldsymbol{\Sigma}\boldsymbol{\beta}_0$$

$$= \frac{1}{K^2}\sum_{k,\ell=1}^K (\widehat{\boldsymbol{\beta}}_\lambda^k)^\top \boldsymbol{\Psi}\widehat{\boldsymbol{\beta}}_\lambda^\ell - (\widehat{\boldsymbol{\beta}}_\mu^{\mathrm{ridge}})^\top \boldsymbol{\Psi}\widehat{\boldsymbol{\beta}}_\mu^{\mathrm{ridge}} + (\widehat{\boldsymbol{\beta}}_\mu^{\mathrm{ridge}})^\top \boldsymbol{\Psi}\widehat{\boldsymbol{\beta}}_\mu^{\mathrm{ridge}} - \frac{2}{K}\sum_{k=1}^K \boldsymbol{\beta}_0^\top \boldsymbol{\Psi}\widehat{\boldsymbol{\beta}}_\lambda^k + \boldsymbol{\beta}_0^\top \boldsymbol{\Psi}\boldsymbol{\beta}_0.$$

By Lemma 14, note that

$$\frac{1}{K}\sum_{k=1}^{K}\widehat{\boldsymbol{\beta}}_{\lambda}^{k} \simeq \widehat{\boldsymbol{\beta}}_{\mu}^{\mathrm{ridge}}.$$

Similarly, by two applications of Lemma 14, we know that $(\widehat{\boldsymbol{\beta}}_{\lambda}^{k})^{\top}\boldsymbol{\Psi}\widehat{\boldsymbol{\beta}}_{\lambda}^{\ell} - (\widehat{\boldsymbol{\beta}}_{\mu}^{\mathrm{ridge}})^{\top}\boldsymbol{\Psi}\widehat{\boldsymbol{\beta}}_{\mu}^{\mathrm{ridge}} \xrightarrow{\mathrm{a.s.}} 0$ when $k \neq \ell$ since $\mathbf{S}_k$ and $\mathbf{S}_\ell$ are independent. The remaining $K$ terms where $k = \ell$ converge identically, so

$$(\widehat{\boldsymbol{\beta}}_{\lambda}^{\mathrm{ens}} - \boldsymbol{\beta}_0)^{\top}\boldsymbol{\Psi}(\widehat{\boldsymbol{\beta}}_{\lambda}^{\mathrm{ens}} - \boldsymbol{\beta}_0)$$
$$- \left( (\widehat{\boldsymbol{\beta}}_{\mu}^{\mathrm{ridge}} - \boldsymbol{\beta}_0)^{\top}\boldsymbol{\Psi}(\widehat{\boldsymbol{\beta}}_{\mu}^{\mathrm{ridge}} - \boldsymbol{\beta}_0) + \frac{1}{K}\big(\widehat{\boldsymbol{\beta}}_{\lambda}^{\top}\boldsymbol{\Psi}\widehat{\boldsymbol{\beta}}_{\lambda} - (\widehat{\boldsymbol{\beta}}_{\mu}^{\mathrm{ridge}})^{\top}\boldsymbol{\Psi}\widehat{\boldsymbol{\beta}}_{\mu}^{\mathrm{ridge}}\big) \right) \xrightarrow{\mathrm{a.s.}} 0.$$

Thus, it suffices to evaluate the difference $\widehat{\boldsymbol{\beta}}_{\lambda}^{\top}\boldsymbol{\Psi}\widehat{\boldsymbol{\beta}}_{\lambda} - (\widehat{\boldsymbol{\beta}}_{\mu}^{\mathrm{ridge}})^{\top}\boldsymbol{\Psi}\widehat{\boldsymbol{\beta}}_{\mu}^{\mathrm{ridge}}$, which we will do next.

By linearity of the trace, we have

$$\widehat{\boldsymbol{\beta}}_{\lambda}^{\top}\boldsymbol{\Psi}\widehat{\boldsymbol{\beta}}_{\lambda} - (\widehat{\boldsymbol{\beta}}_{\mu}^{\mathrm{ridge}})^{\top}\boldsymbol{\Psi}\widehat{\boldsymbol{\beta}}_{\mu}^{\mathrm{ridge}} = \tfrac{1}{n}\mathrm{tr}\left[ \big(\mathbf{X}_{\lambda}^{\ddagger\top}\boldsymbol{\Psi}\mathbf{X}_{\lambda}^{\ddagger} - \mathbf{X}_{\lambda}^{\dagger\top}\boldsymbol{\Psi}\mathbf{X}_{\lambda}^{\dagger}\big)\mathbf{y}\mathbf{y}^{\top} \right],$$

where

$$\mathbf{X}_{\lambda}^{\ddagger} = \tfrac{1}{\sqrt{n}}\mathbf{S}\big(\tfrac{1}{n}\mathbf{S}^{\top}\mathbf{X}^{\top}\mathbf{X}\mathbf{S} + \lambda\mathbf{I}_q\big)^{-1}\mathbf{S}^{\top}\mathbf{X}^{\top} \text{ and } \mathbf{X}_{\lambda}^{\dagger} = \tfrac{1}{\sqrt{n}}\big(\tfrac{1}{n}\mathbf{X}^{\top}\mathbf{X} + \mu\mathbf{I}_p\big)^{-1}\mathbf{X}^{\top}.$$

The result now follows by evaluating the second-order asymptotic equivalences from Lemma 15. This concludes the proof. $\qquad\square$

## C.2   PROOF OF THEOREM 3

The main ingredient of the proof will be Lemma 18. Comparing the expressions of $\mu'$ and $\mu''$, it suffices to show the following equivalence:

$$\tfrac{1}{p}\mathrm{tr}\big[\boldsymbol{\Sigma}\big(\tfrac{1}{n}\mathbf{X}^{\top}\mathbf{X} + \mu\mathbf{I}_p\big)^{-2}\big] \simeq \frac{\tfrac{1}{p}\mathrm{tr}\big[\tfrac{1}{n}\mathbf{X}^{\top}\mathbf{X}\big(\tfrac{1}{n}\mathbf{X}^{\top}\mathbf{X} + \mu\mathbf{I}_p\big)^{-2}\big]}{\big(1 - \tfrac{1}{n}\mathrm{tr}\big[\tfrac{1}{n}\mathbf{X}^{\top}\mathbf{X}(\tfrac{1}{n}\mathbf{X}^{\top}\mathbf{X} + \mu\mathbf{I}_p)^{-1}\big]\big)^2}.$$

We show this equivalence in Lemma 18 to finish the proof.

A side remark that is worth stressing about the proof of Theorem 3: The inflation in both the test error and train errors are such that the same GCV denominator cancels them appropriately! Thus, while one may expect that the GCV for infinite ensemble may work given the equivalence to the ridge regression, the fact that GCV works for a single instance of sketch is (even to the authors) quite remarkable!

HELPER LEMMA FOR THE PROOF OF THEOREM 3

**Lemma 18** (Equivalence of risk and GCV inflation factors for feature sketch). *Under Assumption B,*

$$\big(\tfrac{1}{n}\mathbf{X}^{\top}\mathbf{X} + \mu\mathbf{I}_p\big)^{-1}\boldsymbol{\Sigma}\big(\tfrac{1}{n}\mathbf{X}^{\top}\mathbf{X} + \mu\mathbf{I}_p\big)^{-1} \simeq \frac{(\tfrac{1}{n}\mathbf{X}^{\top}\mathbf{X} + \mu\mathbf{I}_p)^{-1}\tfrac{1}{n}\mathbf{X}^{\top}\mathbf{X}(\tfrac{1}{n}\mathbf{X}^{\top}\mathbf{X} + \mu\mathbf{I}_p)^{-1}}{\big(1 - \tfrac{1}{n}\mathrm{tr}\big[\tfrac{1}{n}\mathbf{X}^{\top}\mathbf{X}(\tfrac{1}{n}\mathbf{X}^{\top}\mathbf{X} + \mu\mathbf{I}_p)^{-1}\big]\big)^2}. \quad \text{(C.8)}$$

*Proof.* We will first derive asymptotic equivalents for both the left-hand and right-hand sides of (C.8). We will then show that the asymptotic equivalents match appropriately.

**Asymptotic equivalent for left-hand side.** From the second part of Lemma 13, we have

$$\mu^2\big(\tfrac{1}{n}\mathbf{X}^{\top}\mathbf{X} + \mu\mathbf{I}_p\big)^{-1}\boldsymbol{\Sigma}\big(\tfrac{1}{n}\mathbf{X}^{\top}\mathbf{X} + \mu\mathbf{I}_p\big)^{-1} \simeq (1 + \widetilde{v}_b)(v\boldsymbol{\Sigma} + \mathbf{I}_p)^{-1}\boldsymbol{\Sigma}(v\boldsymbol{\Sigma} + \mathbf{I}_p)^{-1},$$

where the parameter $\widetilde{v}_b$ from (B.4) is given by:

$$\widetilde{v}_b = \frac{\gamma\tfrac{1}{p}\mathrm{tr}[\boldsymbol{\Sigma}^2(v\boldsymbol{\Sigma} + \mathbf{I}_p)^{-2}]}{v^{-2} - \gamma\tfrac{1}{p}\mathrm{tr}[\boldsymbol{\Sigma}^2(v\boldsymbol{\Sigma} + \mathbf{I}_p)^{-2}]}.$$

Now, note that

$$1 + \widetilde{v}_b = \frac{v^{-2}}{v^{-2} + \gamma \frac{1}{p}\mathrm{tr}[\boldsymbol{\Sigma}^2(v\boldsymbol{\Sigma} + \mathbf{I}_p)^{-2}]},$$

which leads to

$$\frac{1 + \widetilde{v}_b}{\mu^2} = \frac{1}{\mu^2}\frac{v^{-2}}{v^{-2} + \gamma \frac{1}{p}\mathrm{tr}[\boldsymbol{\Sigma}^2(v\boldsymbol{\Sigma} + \mathbf{I}_p)^{-2}]}.$$

Thus, we have that

$$(\tfrac{1}{n}\mathbf{X}^\top\mathbf{X} + \mu\mathbf{I}_p)^{-1}\boldsymbol{\Sigma}(\tfrac{1}{n}\mathbf{X}^\top\mathbf{X} + \mu\mathbf{I}_p)^{-1}$$
$$\simeq \frac{1}{\mu^2}\frac{v^{-2}}{v^{-2} + \gamma \frac{1}{p}\mathrm{tr}[\boldsymbol{\Sigma}^2(v\boldsymbol{\Sigma} + \mathbf{I}_p)^{-2}]}(v\boldsymbol{\Sigma} + \mathbf{I}_p)^{-1}\boldsymbol{\Sigma}(v\boldsymbol{\Sigma} + \mathbf{I}_p)^{-1}. \tag{C.9}$$

**Asymptotic equivalent for right-hand side.** From the third part of Lemma 13, we have

$$(\tfrac{1}{n}\mathbf{X}^\top\mathbf{X} + \mu\mathbf{I}_p)^{-1}\tfrac{1}{n}\mathbf{X}^\top\mathbf{X}(\tfrac{1}{n}\mathbf{X}^\top\mathbf{X} + \mu\mathbf{I}_p)^{-1} \simeq \widetilde{v}_v(v\boldsymbol{\Sigma} + \mathbf{I}_p)^{-1}\boldsymbol{\Sigma}(v\boldsymbol{\Sigma} + \mathbf{I}_p)^{-1}, \tag{C.10}$$

where the parameter $\widetilde{v}_v$ from (B.6) is given by:

$$\widetilde{v}_v = \frac{1}{v^{-2} - \gamma \frac{1}{p}\mathrm{tr}[\boldsymbol{\Sigma}^2(v\boldsymbol{\Sigma} + \mathbf{I}_p)^{-2}]}. \tag{C.11}$$

From the first of Lemma 13, we have

$$
\begin{aligned}
1 - \tfrac{1}{n}\mathrm{tr}[\tfrac{1}{n}\mathbf{X}^\top\mathbf{X}(\tfrac{1}{n}\mathbf{X}^\top\mathbf{X} + \mu\mathbf{I}_p)^{-1}] &= 1 - \tfrac{1}{n}\mathrm{tr}[\mathbf{I}_p - \mu(\tfrac{1}{n}\mathbf{X}^\top\mathbf{X} + \mu\mathbf{I}_p)^{-1}] \\
&= 1 - \gamma\big(1 - \tfrac{1}{p}\mu\mathrm{tr}[(\tfrac{1}{n}\mathbf{X}^\top\mathbf{X} + \mu\mathbf{I}_p)^{-1}]\big) \\
&\simeq 1 - \gamma\big(1 - \tfrac{1}{p}\mathrm{tr}[(v\boldsymbol{\Sigma} + \mathbf{I}_p)^{-1}]\big) \\
&= 1 - \gamma + \gamma\tfrac{1}{p}\mathrm{tr}[(v\boldsymbol{\Sigma} + \mathbf{I}_p)^{-1}]. 
\end{aligned}
\tag{C.12}$$

Combining (C.10), (C.11), and (C.12), we obtain

$$
\frac{(\tfrac{1}{n}\mathbf{X}^\top\mathbf{X} + \mu\mathbf{I}_p)^{-1}\tfrac{1}{n}\mathbf{X}^\top\mathbf{X}(\tfrac{1}{n}\mathbf{X}^\top\mathbf{X} + \mu\mathbf{I}_p)^{-1}}{\big(1 - \tfrac{1}{n}\mathrm{tr}[\tfrac{1}{n}\mathbf{X}^\top\mathbf{X}(\tfrac{1}{n}\mathbf{X}^\top\mathbf{X} + \mu\mathbf{I}_p)^{-1}]\big)^2}
$$
$$
\simeq \frac{1}{v^{-2} + \gamma \frac{1}{p}\mathrm{tr}[\boldsymbol{\Sigma}^2(v\boldsymbol{\Sigma} + \mathbf{I}_p)^{-2}]} \cdot \frac{(v\boldsymbol{\Sigma} + \mathbf{I}_p)^{-1}\boldsymbol{\Sigma}(v\boldsymbol{\Sigma} + \mathbf{I}_p)^{-1}}{\big(1 - \gamma + \gamma\tfrac{1}{p}\mathrm{tr}[(v\boldsymbol{\Sigma} + \mathbf{I}_p)^{-1}]\big)^2}. \tag{C.13}
$$

**Matching of asymptotic equivalents.** Note from the fixed-point equation (B.2) that

$$\mu = v^{-1} - \gamma\tfrac{1}{p}\mathrm{tr}[\boldsymbol{\Sigma}(v\boldsymbol{\Sigma} + \mathbf{I}_p)^{-1}].$$

Multiplying by $v$ on both sides yields

$$\mu v = 1 - \gamma\tfrac{1}{p}\mathrm{tr}[v\boldsymbol{\Sigma}(v\boldsymbol{\Sigma} + \mathbf{I}_p)^{-1}] = 1 - \gamma + \gamma\tfrac{1}{p}\mathrm{tr}[(v\boldsymbol{\Sigma} + \mathbf{I}_p)^{-1}]. \tag{C.14}$$

Thus, combining (C.9), (C.14), and (C.13), we have

$$
(\tfrac{1}{n}\mathbf{X}^\top\mathbf{X} + \mu\mathbf{I}_p)^{-1}\boldsymbol{\Sigma}(\tfrac{1}{n}\mathbf{X}^\top\mathbf{X} + \mu\mathbf{I}_p)^{-1}
$$
$$
\simeq \frac{1}{\mu^2}\frac{v^{-2}}{v^{-2} + \gamma \frac{1}{p}\mathrm{tr}[\boldsymbol{\Sigma}^2(v\boldsymbol{\Sigma} + \mathbf{I}_p)^{-2}]}(v\boldsymbol{\Sigma} + \mathbf{I}_p)^{-1}\boldsymbol{\Sigma}(v\boldsymbol{\Sigma} + \mathbf{I}_p)^{-1}
$$
$$
= \frac{1}{(\mu v)^2}\frac{1}{v^{-2} + \gamma \frac{1}{p}\mathrm{tr}[\boldsymbol{\Sigma}^2(v\boldsymbol{\Sigma} + \mathbf{I}_p)^{-2}]}(v\boldsymbol{\Sigma} + \mathbf{I}_p)^{-1}\boldsymbol{\Sigma}(v\boldsymbol{\Sigma} + \mathbf{I}_p)^{-1}
$$
$$
= \frac{1}{(1 - \gamma + \gamma\frac{1}{p}\mathrm{tr}[(v\boldsymbol{\Sigma} + \mathbf{I}_p)^{-1}])^2}\frac{1}{v^{-2} + \gamma \frac{1}{p}\mathrm{tr}[\boldsymbol{\Sigma}^2(v\boldsymbol{\Sigma} + \mathbf{I}_p)^{-2}]}(v\boldsymbol{\Sigma} + \mathbf{I}_p)^{-1}\boldsymbol{\Sigma}(v\boldsymbol{\Sigma} + \mathbf{I}_p)^{-1}
$$
$$
\simeq \frac{(\tfrac{1}{n}\mathbf{X}^\top\mathbf{X} + \mu\mathbf{I}_p)^{-1}\tfrac{1}{n}\mathbf{X}^\top\mathbf{X}(\tfrac{1}{n}\mathbf{X}^\top\mathbf{X} + \mu\mathbf{I}_p)^{-1}}{\big(1 - \tfrac{1}{n}\mathrm{tr}[\tfrac{1}{n}\mathbf{X}^\top\mathbf{X}(\tfrac{1}{n}\mathbf{X}^\top\mathbf{X} + \mu\mathbf{I}_p)^{-1}]\big)^2}.
$$

In the chain above, the first equivalence follows from (C.9), the penultimate equality follows from (C.14), and the final equivalence follows from (C.13). This finishes the proof. $\qquad\square$

# D  PROOFS IN SECTION 4

## D.1  PROOF OF THEOREM 4

As mentioned in the main paper, the idea of the proof is to exploit the close connection between GCV and LOOCV to prove the consistency for general functionals. In particular, we will consider an intermediate functional, constructed based on LOO-reweighted residuals. We will then connect the functional constructed based on GCV-reweighted residuals to that based on LOO-reweighted residuals.

**Step 1: LOOCV consistency.** Let $\mathbf{X}_{-i}$ denote the the feature matrix obtained by removing the $i$-th row from $\mathbf{X}$, and $\mathbf{y}_{-i}$ is the response vector obtained by removing the $i$-th entry from $\mathbf{y}$. Let $\widehat{\boldsymbol{\beta}}_{-i,\lambda}^{\text{ens}}$ denote the LOO ensemble estimator. It is defined using $K$ constituent sketched LOO estimators $\widehat{\boldsymbol{\beta}}_{-i,\lambda}^{k}$ for $k \in [K]$ as follows:

$$\widehat{\boldsymbol{\beta}}_{-i,\lambda}^{\text{ens}} = \frac{1}{K} \sum_{k=1}^{K} \widehat{\boldsymbol{\beta}}_{-i,\lambda}^{k}, \quad \text{where} \quad \widehat{\boldsymbol{\beta}}_{-i,\lambda}^{k} = \frac{1}{n} \mathbf{S}_k (\frac{1}{n} \mathbf{S}_k \mathbf{X}_{-i}^{\top} \mathbf{X}_{-i} \mathbf{S}_k + \lambda \mathbf{I}_q)^{-1} \mathbf{S}_k^{\top} \mathbf{X}_{-i}^{\top} \mathbf{y}_{-i}.$$

The LOOCV functional is the defined using the predictions of the LOO ensemble estimator as:

$$\widehat{T}_{\lambda}^{\text{loo}} = \frac{1}{n} \sum_{i=1}^{n} t(y_i, \mathbf{x}_i^{\top} \widehat{\boldsymbol{\beta}}_{-i,\lambda}^{\text{ens}}). \tag{D.1}$$

It follows from the proof of Theorem 3 of Patil et al. (2022) that $|R(\widehat{\boldsymbol{\beta}}_{\lambda}^{\text{ens}}) - \widehat{T}_{\lambda}^{\text{loo}}| \xrightarrow{\text{a.s.}} 0$. Next we will show that $|\widehat{T}_{\lambda}^{\text{loo}} - \widehat{T}_{\lambda}| \xrightarrow{\text{a.s.}} 0$, where $\widehat{T}_{\lambda}$ is the GCV functional.

**Step 2: From LOOCV to GCV consistency.** To go from LOOCV to GCV, we first rewrite the LOO errors. From Woodbury matrix identity, observe that

$$y_i - \mathbf{x}_i^{\top} \widehat{\boldsymbol{\beta}}_{-i,\lambda}^{k} = \frac{y_i - \mathbf{x}_i^{\top} \widehat{\boldsymbol{\beta}}_{\lambda}^{k}}{1 - [\mathbf{L}_{\lambda}^{k}]_{ii}}.$$

This is the so-called exact shortcut for LOO errors for ridge regression, that also works for sketched ridge regression. In other words, we have

$$\mathbf{x}_i^{\top} \widehat{\boldsymbol{\beta}}_{-i,\lambda}^{k} = y_i - \frac{y_i - \mathbf{x}_i^{\top} \widehat{\boldsymbol{\beta}}_{\lambda}^{k}}{1 - [\mathbf{L}_{\lambda}^{k}]_{ii}}.$$

This implies that

$$\mathbf{x}_i^{\top} \widehat{\boldsymbol{\beta}}_{-i,\lambda}^{\text{ens}} = y_i - \frac{1}{K} \sum_{k=1}^{K} \frac{y_i - \mathbf{x}_i^{\top} \widehat{\boldsymbol{\beta}}_{\lambda}^{k}}{1 - [\mathbf{L}_{\lambda}^{k}]_{ii}}.$$

Thus, we can write the LOO function (D.1) in the shortcut form as follows:

$$\widehat{T}_{\lambda}^{\text{loo}} = \frac{1}{n} \sum_{i=1}^{n} t \left( y_i, y_i - \frac{1}{K} \sum_{k=1}^{K} \frac{y_i - \mathbf{x}_i^{\top} \widehat{\boldsymbol{\beta}}_{\lambda}^{k}}{1 - [\mathbf{L}_{\lambda}^{k}]_{ii}} \right).$$

We will now bound the difference:

$$|\widehat{T}_{\lambda}^{\text{loo}} - \widehat{T}_{\lambda}|$$
$$= \frac{1}{n} \sum_{i=1}^{n} \left| t \left( y_i, y_i - \frac{1}{K} \sum_{k=1}^{K} \frac{y_i - \mathbf{x}_i^{\top} \widehat{\boldsymbol{\beta}}_{\lambda}^{k}}{1 - [\mathbf{L}_{\lambda}^{k}]_{ii}} \right) - t \left( y_i, y_i - \frac{y_i - \mathbf{x}_i^{\top} \widehat{\boldsymbol{\beta}}_{\lambda}^{\text{ens}}}{1 - \frac{1}{n} \text{tr}[\mathbf{L}_{\lambda}^{\text{ens}}]} \right) \right|$$
$$= \frac{1}{n} \sum_{i=1}^{n} \left| t \left( y_i, y_i - \frac{1}{K} \sum_{k=1}^{K} \frac{y_i - \mathbf{x}_i^{\top} \widehat{\boldsymbol{\beta}}_{\lambda}^{k}}{1 - [\mathbf{L}_{\lambda}^{k}]_{ii}} \right) - t \left( y_i, y_i - \frac{1}{K} \sum_{k=1}^{K} \frac{y_i - \mathbf{x}_i^{\top} \widehat{\boldsymbol{\beta}}_{\lambda}^{k}}{1 - \frac{1}{n} \text{tr}[\mathbf{L}_{\lambda}^{\text{ens}}]} \right) \right|.$$

The final equality follows from using the definition of the ensemble estimator (2). For notational simplicity, denote by:

- $r_i^k = y_i - \mathbf{x}_i^\top \widehat{\boldsymbol{\beta}}_\lambda^k$ for $k \in [K]$ and $i \in [n]$;
- $d_i^k = 1 - [\mathbf{L}_\lambda^k]_{ii}$ for $k \in [K]$ and $i \in [n]$, and $\bar{d} = 1 - \frac{1}{n}\mathrm{tr}[\mathbf{L}_\lambda^{\mathrm{ens}}]$;
- $\mathbf{u}_i = (y_i, y_i - \frac{1}{K}\sum_{k=1}^K \frac{r_i^k}{d_i^k})$, $\mathbf{v}_i = (y_i, y_i - \frac{1}{K}\sum_{k=1}^K \frac{r_i^k}{\bar{d}})$.

Now, consider the desired difference:

$$
|\widehat{T}_\lambda^{\mathrm{loo}} - \widehat{T}_\lambda| = \frac{1}{n}\sum_{i=1}^n |t(\mathbf{u}_i) - t(\mathbf{v}_i)|
$$

$$
\leqslant \frac{1}{n}\sum_{i=1}^n L(1 + \|\mathbf{u}_i\|_2 + \|\mathbf{v}_i\|_2)(\|\mathbf{u}_i - \mathbf{v}_i\|_2)
$$

$$
= \frac{1}{n}\sum_{i=1}^n L(1 + \|\mathbf{u}_i\|_2 + \|\mathbf{v}_i\|_2)\left| \frac{1}{K}\sum_{k=1}^K r_i^k\left(\frac{1}{d_i^k} - \frac{1}{\bar{d}}\right)\right|. \tag{D.2}
$$

Above, we used Assumption C in the inequality. Using Hölder's inequality, we can bound

$$
\left| \frac{1}{K}\sum_{k=1}^K r_i^k\left(\frac{1}{d_i^k} - \frac{1}{\bar{d}}\right)\right| \leqslant \frac{1}{K}\sum_{k=1}^K |r_i^k| \cdot \max_{1\leqslant k\leqslant K}\left| \frac{1}{d_i^k} - \frac{1}{\bar{d}}\right|
$$

$$
\leqslant \frac{1}{\sqrt{K}}\|\mathbf{r}_i\|_2 \cdot \max_{1\leqslant k\leqslant K}\left| \frac{1}{d_i^k} - \frac{1}{\bar{d}}\right|, \tag{D.3}
$$

where we denote by $\mathbf{r}_i = (r_i^1, \ldots, r_i^K)$ for $i \in [n]$. In the second inequality, we used the fact that $\|\mathbf{r}_i\|_1 \leqslant \sqrt{K}\|\mathbf{r}_i\|_2$. Combining (D.2) with (D.3) yields

$$
|\widehat{T}_\lambda^{\mathrm{loo}} - \widehat{T}_\lambda| \leqslant \frac{1}{n}\sum_{i=1}^n \left\{ L(1 + \|\mathbf{u}_i\|_2 + \|\mathbf{v}_i\|_2)\left(\frac{1}{\sqrt{K}}\|\mathbf{r}_i\|_2\right)\right\}\left\{ \max_{1\leqslant k\leqslant K}\left| \frac{1}{d_i^k} - \frac{1}{\bar{d}}\right|\right\}
$$

$$
\leqslant L\left\{ \frac{1}{n}\sum_{i=1}^n (1 + \|\mathbf{u}_i\|_2 + \|\mathbf{v}_i\|_2)\left(\frac{1}{\sqrt{K}}\|\mathbf{r}_i\|_2\right)\right\}\left\{ \max_{1\leqslant i\leqslant n}\max_{1\leqslant k\leqslant K}\left| \frac{1}{d_i^k} - \frac{1}{\bar{d}}\right|\right\}.
$$

Further denote by:

- $\mathbf{u} = (\|\mathbf{u}_1\|_2, \ldots, \|\mathbf{u}_n\|_2)$;
- $\mathbf{v} = (\|\mathbf{v}_1\|_2, \ldots, \|\mathbf{v}_n\|_2)$;
- $\mathbf{r} = \frac{1}{\sqrt{K}}(\|\mathbf{r}_1\|_2, \ldots, \|\mathbf{r}_n\|_2)$;
- $\delta_i = \max_{1\leqslant k\leqslant K}|1/d_i^k - 1/\bar{d}|$ for $i \in [n]$, and $\boldsymbol{\delta} = (\delta_1, \ldots, \delta_n)$.

Continuing from above, using the Cauchy-Schwartz inequality on the first term and triangle inequality, we obtain

$$
|\widehat{T}_\lambda^{\mathrm{loo}} - \widehat{T}_\lambda| \leqslant L \cdot \left(1 + \frac{\|\mathbf{u}\|_2}{\sqrt{n}} + \frac{\|\mathbf{v}\|_2}{\sqrt{n}}\right) \cdot \frac{\|\mathbf{r}\|_2}{\sqrt{n}} \cdot \|\boldsymbol{\delta}\|_\infty \leqslant C\|\boldsymbol{\delta}\|_\infty.
$$

From a short calculations, it follows that $\frac{1}{\sqrt{n}}\|\mathbf{u}\|_2$, $\frac{1}{\sqrt{n}}\|\mathbf{v}\|_2$, $\frac{1}{\sqrt{n}}\|\mathbf{r}\|_2$ are all eventually almost surely bounded. We will next show that $\|\boldsymbol{\delta}\|_\infty \xrightarrow{\mathrm{a.s.}} 0$.

*Sup-norm concentration*: We will show that

$$
\max_{1\leqslant i\leqslant n}\left| \frac{1}{K}\sum_{k=1}^K \frac{1}{1 - \mathrm{tr}[\mathbf{L}_\lambda^k]_{ii}} - \frac{1}{1 - \frac{1}{n}\mathrm{tr}[\mathbf{L}_\lambda^{\mathrm{ens}}]}\right| \xrightarrow{\mathrm{a.s.}} 0.
$$

From Lemma 14, observe that

$$
\frac{1}{K}\sum_{k=1}^K \frac{1}{1 - [\mathbf{L}_\lambda^k]_{ii}} = \frac{1}{K}\sum_{k=1}^K \frac{1}{1 - [\frac{1}{n}\mathbf{X}\mathbf{S}_k(\frac{1}{n}\mathbf{S}_k^\top\mathbf{X}^\top\mathbf{X}\mathbf{S}_k + \lambda\mathbf{I}_q)^{-1}\mathbf{S}_k^\top\mathbf{X}^\top]_{ii}}
$$

$$
\simeq \frac{1}{K}\sum_{k=1}^K \frac{1}{1 - [\mathbf{L}_\mu^{\mathrm{ridge}}]_{ii}} = \frac{1}{1 - [\mathbf{L}_\mu^{\mathrm{ridge}}]_{ii}}.
$$

Similarly, using Lemma 14 again, we also have

$$\frac{1}{1 - \frac{1}{n}\text{tr}[\mathbf{L}_\lambda^{\text{ens}}]} = \frac{1}{1 - \frac{1}{n}\frac{1}{K}\sum_{k=1}^K \text{tr}[\mathbf{L}_\lambda^k]}$$

$$= \frac{1}{1 - \frac{1}{n}\frac{1}{K}\sum_{k=1}^K \text{tr}[\frac{1}{n}\mathbf{X}\mathbf{S}_k(\frac{1}{n}\mathbf{S}_k^\top\mathbf{X}^\top\mathbf{X}\mathbf{S}_k + \lambda\mathbf{I}_q)^{-1}\mathbf{S}_k^\top\mathbf{X}^\top]}$$

$$\simeq \frac{1}{1 - \frac{1}{n}\frac{1}{K}\sum_{k=1}^K \text{tr}[\mathbf{L}_\mu^{\text{ridge}}]} = \frac{1}{1 - \frac{1}{n}\text{tr}[\mathbf{L}_\mu^{\text{ridge}}]}.$$

Thus, we have the desired difference to be

$$\max_{1 \leqslant i \leqslant n} \left| \frac{1}{K}\sum_{k=1}^K \frac{1}{1 - \text{tr}[\mathbf{L}_\lambda^k]_{ii}} - \frac{1}{1 - \frac{1}{n}\text{tr}[\mathbf{L}_\lambda^{\text{ens}}]} \right| \simeq \max_{1 \leqslant i \leqslant n} \left| \frac{1}{1 - [\mathbf{L}_\mu^{\text{ridge}}]_{ii}} - \frac{1}{1 - \frac{1}{n}\text{tr}[\mathbf{L}_\mu^{\text{ridge}}]} \right|.$$

From the proof of Theorem 3 of Patil et al. (2022), the right-hand side of the display above almost surely vanishes. This completes the proof.

### D.2 Proof of Corollary 5

This is a simple consequence of Theorem 4 using the definition of convergence in Wasserstein $W_2$ metric. See, e.g., Chapter 6 of Villani (2008).

## E Proofs in Section 5

### Proof of Proposition 6

We begin by noting that when $\lambda = 0$, it suffices to prove the result for i.i.d. Gaussian sketches only. Consider first any sketch $\mathbf{S} = \mathbf{U}\mathbf{D}\mathbf{V}^\top$ where $\mathbf{U}$ is a uniformly distributed unitary matrix and $\mathbf{D}$ is invertible. Then

$$\mathbf{S}(\mathbf{S}^\top\hat{\boldsymbol{\Sigma}}\mathbf{S})^{-1}\mathbf{S}^\top = \mathbf{U}(\mathbf{U}^\top\hat{\boldsymbol{\Sigma}}\mathbf{U})^{-1}\mathbf{U}^\top,$$

and so the result does not depend on $\mathbf{D}$ and $\mathbf{V}$, and we can choose them to have the same distribution as in i.i.d. Gaussian sketching. For general free sketching, $\mathbf{S}$ may not have $\mathbf{U}$ uniformly distributed in finite dimensions, but the subordination relations in Theorem 1 depend only on the spectrum of $\mathbf{S}\mathbf{S}^\top$, so we can without loss of generality assume that $\mathbf{U}$ are uniformly distributed and obtain the exact same equivalence relationship.

The subordination relation

$$\mu \simeq \lambda \mathscr{S}_{\mathbf{S}\mathbf{S}^\top}\left(-\frac{1}{p}\text{tr}[\hat{\boldsymbol{\Sigma}}(\hat{\boldsymbol{\Sigma}} + \mu\mathbf{I}_p)^{-1}]\right)$$

for $\mu > 0$ and $\lambda = 0$ requires that the the S-transform must go to $\infty$. From Table 4, we know

$$\mathscr{S}_{\mathbf{S}\mathbf{S}^\top}(w) = \frac{\alpha}{\alpha + w}$$

for i.i.d. sketching, which means that we must send the denominator to 0. Thus we obtain the condition

$$\alpha = \frac{1}{p}\text{tr}\left[\hat{\boldsymbol{\Sigma}}(\hat{\boldsymbol{\Sigma}} + \mu\mathbf{I}_p)^{-1}\right].$$

By letting $K \to \infty$, the variance term vanishes, and only the bias term remains, proving the result.

## F Proofs in Section 6

### F.1 Proof of Proposition 7

We provide below the complete statement of Proposition 7, which includes expressions for $\nu'$ and $\nu''$. These are excluded from the main paper.

**Proposition 19** (Squared risk and GCV asymptotics and GCV inconsistency for observation sketch). *Suppose Assumption A holds for $\mathbf{T}\mathbf{T}^\top$, and that the operator norm of $\mathbf{\Sigma}$ and second moment of $y_0$ are uniformly bounded in $p$. Then, for $\lambda > \widetilde{\lambda}_0$ and any $K$,*

$$R(\widetilde{\boldsymbol{\beta}}_\lambda^{\mathrm{ens}}) \simeq R(\widetilde{\boldsymbol{\beta}}_\nu^{\mathrm{ridge}}) + \frac{\nu'\widetilde{\Delta}}{K} \quad and \quad \widetilde{R}(\widetilde{\boldsymbol{\beta}}_\lambda^{\mathrm{ens}}) \simeq \widetilde{R}(\widetilde{\boldsymbol{\beta}}_\nu^{\mathrm{ridge}}) + \frac{\nu''\widetilde{\Delta}}{K}, \tag{F.1}$$

*where $\nu$ is an implicit regularization parameter that solves (B.10), $\widetilde{\Delta} = \frac{1}{n}\mathbf{y}^\top(\frac{1}{n}\mathbf{X}\mathbf{X}^\top + \nu\mathbf{I}_n)^{-2}\mathbf{y}$, and $\nu' \geq 0$ is an inflation factor in the risk decomposition given by:*

$$\nu' =$$
$$-\frac{\partial\mu}{\partial\lambda}\lambda^2\mathscr{S}'_{\mathbf{T}\mathbf{T}^\top}\big(-\tfrac{1}{n}\mathrm{tr}\big[\tfrac{1}{n}\mathbf{X}\mathbf{X}^\top\big(\tfrac{1}{n}\mathbf{X}\mathbf{X}^\top + \nu\mathbf{I}_n\big)^{-1}\big]\big)\tfrac{1}{p}\mathrm{tr}\big[\tfrac{1}{n}\mathbf{X}\mathbf{\Sigma}\mathbf{X}^\top\big(\tfrac{1}{n}\mathbf{X}\mathbf{X}^\top + \nu\mathbf{I}_n\big)^{-2}\big], \tag{F.2}$$

*while $\nu'' \geq 0$ is an inflation the GCV decomposition given by:*

$$\nu'' =$$
$$\frac{-\frac{\partial\mu}{\partial\lambda}\lambda^2\mathscr{S}'_{\mathbf{T}\mathbf{T}^\top}\big(-\tfrac{1}{n}\mathrm{tr}\big[\tfrac{1}{n}\mathbf{X}\mathbf{X}^\top\big(\tfrac{1}{n}\mathbf{X}\mathbf{X}^\top + \nu\mathbf{I}_n\big)^{-1}\big]\big)\tfrac{1}{p}\mathrm{tr}\big[\tfrac{1}{n}\mathbf{X}\big(\tfrac{1}{n}\mathbf{X}^\top\mathbf{X}\big)\mathbf{X}^\top\big(\tfrac{1}{n}\mathbf{X}\mathbf{X}^\top + \nu\mathbf{I}_n\big)^{-2}\big]}{\big(1 - \tfrac{1}{n}\mathbf{X}\mathbf{X}^\top\big(\tfrac{1}{n}\mathbf{X}\mathbf{X}^\top + \nu\mathbf{I}_n\big)^{-1}\big)^2}.$$
$$\tag{F.3}$$

*Furthermore, under Assumption B, in general we have $\nu' \neq \nu''$, and therefore $\widetilde{R}(\widetilde{\boldsymbol{\beta}}_\lambda^{\mathrm{ens}}) \neq R(\widetilde{\boldsymbol{\beta}}_\lambda^{\mathrm{ens}})$.*

*Proof.* The proof for the decomposition in (F.1) is similar to the proof of Theorem 16. Our main workforce will be Lemma 20.

*Notation*: We will use the same strategy and notation as in the proof of Theorem 16. We will decompose the unknown response $y_0$ into the linear predictor corresponding to best linear projection parameter and the residual. Let $\boldsymbol{\beta}_0$ denote the best projection parameter: $\boldsymbol{\beta}_0 = \mathbf{\Sigma}^{-1}\mathbb{E}[\mathbf{x}_0 y_0]$. We decompose the response into the best linear predictor $\mathbf{x}^\top\boldsymbol{\beta}_0$ and the residual error $y_0 - \mathbf{x}_0^\top\boldsymbol{\beta}_0$. We will denote by $\sigma^2 = \mathbb{E}[(y_0 - \mathbf{x}_0^\top\boldsymbol{\beta}_0)^2]$.

**Part 1: Risk asymptotics.** As done in the proof of Theorem 16, we have

$$R(\widetilde{\boldsymbol{\beta}}_\lambda^{\mathrm{ens}}) = (\widetilde{\boldsymbol{\beta}}_\lambda^{\mathrm{ens}} - \boldsymbol{\beta}_0)^\top\mathbf{\Sigma}(\widetilde{\boldsymbol{\beta}}_\lambda^{\mathrm{ens}} - \boldsymbol{\beta}_0) + \sigma^2 \simeq (\widetilde{\boldsymbol{\beta}}_\mu^{\mathrm{ridge}} - \boldsymbol{\beta}_0)^\top\mathbf{\Sigma}(\widetilde{\boldsymbol{\beta}}_\mu^{\mathrm{ridge}}) + \sigma^2 + \frac{\nu'\widetilde{\Delta}}{K},$$

where $\nu'$ is as defined in (F.2). In the second step, we now instead used Lemma 20 to obtain the desired equivalence. This completes the proof for the risk asymptotics decomposition.

**Part 2: GCV asymptotics.** We will obtain asymptotic equivalents for the numerator and denominator of GCV separately below.

*Numerator*: Similar to the proof of Theorem 16, we first decompose

$$\tfrac{1}{n}\|\mathbf{y} - \mathbf{X}\widetilde{\boldsymbol{\beta}}_\lambda^{\mathrm{ens}}\|_2^2 = \tfrac{1}{n}\|\mathbf{X}(\boldsymbol{\beta}_0 - \widetilde{\boldsymbol{\beta}}_\lambda^{\mathrm{ens}})\|_2^2 + \tfrac{1}{n}\|(\mathbf{y} - \mathbf{X}\boldsymbol{\beta}_0)\|_2^2 + \tfrac{2}{n}(\mathbf{y} - \mathbf{X}\boldsymbol{\beta}_0)^\top\mathbf{X}(\boldsymbol{\beta}_0 - \widetilde{\boldsymbol{\beta}}_\lambda^{\mathrm{ens}}).$$

An application of Lemma 14 yields

$$\tfrac{2}{n}(\mathbf{y} - \mathbf{X}\boldsymbol{\beta}_0)^\top\mathbf{X}(\boldsymbol{\beta}_0 - \widetilde{\boldsymbol{\beta}}_\lambda^{\mathrm{ens}}) \simeq \tfrac{2}{n}(\mathbf{y} - \mathbf{X}\boldsymbol{\beta}_0)^\top\mathbf{X}(\boldsymbol{\beta}_0 - \widetilde{\boldsymbol{\beta}}_\mu^{\mathrm{ridge}}).$$

Notice that

$$\tfrac{1}{n}\|\mathbf{X}(\boldsymbol{\beta}_0 - \widetilde{\boldsymbol{\beta}}_\lambda^{\mathrm{ens}})\|_2^2 = (\widetilde{\boldsymbol{\beta}}_\lambda^{\mathrm{ens}} - \boldsymbol{\beta}_0)^\top\tfrac{1}{n}\mathbf{X}^\top\mathbf{X}(\widetilde{\boldsymbol{\beta}}_\lambda^{\mathrm{ens}} - \boldsymbol{\beta}_0)$$

$$\simeq (\widetilde{\boldsymbol{\beta}}_\mu^{\mathrm{ridge}} - \boldsymbol{\beta}_0)^\top\tfrac{1}{n}\mathbf{X}^\top\mathbf{X}(\widetilde{\boldsymbol{\beta}}_\mu^{\mathrm{ridge}} - \boldsymbol{\beta}_0) + \frac{\nu''_{\mathrm{num}}\widetilde{\Delta}}{K},$$

where $\nu''_{\mathrm{num}}$ is expressed as:

$$\nu''_{\mathrm{num}} = -\frac{\partial\mu}{\partial\lambda}\lambda^2\mathscr{S}'_{\mathbf{T}\mathbf{T}^\top}\big(-\tfrac{1}{n}\mathrm{tr}\big[\tfrac{1}{n}\mathbf{X}\mathbf{X}^\top\big(\tfrac{1}{n}\mathbf{X}\mathbf{X}^\top + \nu\mathbf{I}_n\big)^{-1}\big]\big).$$

Using Lemma 17, we get

$$\tfrac{1}{n}\|\mathbf{y} - \mathbf{X}\widetilde{\boldsymbol{\beta}}_\lambda^{\mathrm{ens}}\|_2^2 \simeq (\widetilde{\boldsymbol{\beta}}_\mu^{\mathrm{ridge}} - \boldsymbol{\beta}_0)^\top \tfrac{1}{n}\mathbf{X}^\top\mathbf{X}(\widetilde{\boldsymbol{\beta}}_\mu^{\mathrm{ridge}} - \boldsymbol{\beta}_0) + \tfrac{2}{n}(\mathbf{y} - \mathbf{X}\boldsymbol{\beta}_0)^\top\mathbf{X}(\boldsymbol{\beta}_0 - \widetilde{\boldsymbol{\beta}}_\mu^{\mathrm{ridge}})$$

$$+ \tfrac{1}{n}\|(\mathbf{y} - \mathbf{X}\boldsymbol{\beta}_0)\|_2^2 + \frac{\nu''_{\mathrm{num}}\widetilde{\Delta}}{K}. \tag{F.4}$$

On the other hand, we also have

$$\tfrac{1}{n}\|\mathbf{y} - \mathbf{X}\widetilde{\boldsymbol{\beta}}_\mu^{\mathrm{ridge}}\|_2^2$$

$$= \tfrac{1}{n}\|(\mathbf{y} - \mathbf{X}\boldsymbol{\beta}_0) + \mathbf{X}(\boldsymbol{\beta}_0 - \widehat{\boldsymbol{\beta}}_\mu^{\mathrm{ridge}})\|_2^2$$

$$= \tfrac{1}{n}\|\mathbf{y} - \mathbf{X}\boldsymbol{\beta}_0\|_2^2 + \tfrac{2}{n}(\mathbf{y} - \mathbf{X}\boldsymbol{\beta}_0)^\top\mathbf{X}(\boldsymbol{\beta}_0 - \widetilde{\boldsymbol{\beta}}_\mu^{\mathrm{ridge}}) + \tfrac{1}{n}\|\mathbf{X}(\boldsymbol{\beta}_0 - \widetilde{\boldsymbol{\beta}}_\mu^{\mathrm{ridge}})\|_2^2. \tag{F.5}$$

Combining (F.4) and (F.5), we deduce that

$$\tfrac{1}{n}\|\mathbf{y} - \mathbf{X}\widetilde{\boldsymbol{\beta}}_\lambda^{\mathrm{ens}}\|_2^2 \simeq \tfrac{1}{n}\|\mathbf{y} - \mathbf{X}\widetilde{\boldsymbol{\beta}}_\mu^{\mathrm{ridge}}\|_2^2 + \frac{\nu''_{\mathrm{num}}\widetilde{\Delta}}{K}. \tag{F.6}$$

*Denominator*: We will now derive an asymptotic equivalent for the GCV denominator. By repeated applications of the Woodbury matrix identity along with the first-order equivalence for observation sketch from Lemma 14, we get

$$\widetilde{\mathbf{L}}_\lambda^{\mathrm{ens}} = \frac{1}{K}\sum_{k=1}^K \tfrac{1}{n}\mathbf{X}(\tfrac{1}{n}\mathbf{X}^\top\mathbf{T}_k\mathbf{T}_k^\top\mathbf{X} + \lambda\mathbf{I}_p)^{-1}\mathbf{X}^\top\mathbf{T}_k\mathbf{T}_k^\top$$

$$= \frac{1}{K}\sum_{k=1}^K \tfrac{1}{n}\mathbf{X}\mathbf{X}^\top\mathbf{T}_k(\tfrac{1}{n}\mathbf{T}_k^\top\mathbf{X}\mathbf{X}^\top\mathbf{T}_k + \lambda\mathbf{I}_m)^{-1}\mathbf{T}_k^\top$$

$$\simeq \tfrac{1}{n}\mathbf{X}\mathbf{X}^\top(\tfrac{1}{n}\mathbf{X}\mathbf{X}^\top + \nu\mathbf{I}_n)^{-1}$$

$$= \tfrac{1}{n}\mathbf{X}(\tfrac{1}{n}\mathbf{X}^\top\mathbf{X} + \nu\mathbf{I}_p)^{-1}\mathbf{X}^\top.$$

In the chain above, we used the Woodbury matrix identity for equality in the second and forth lines, and Lemma 14 for the equivalence in the third line. Hence, we get

$$\mathrm{tr}[\widetilde{\mathbf{L}}_\lambda^{\mathrm{ens}}] \simeq \mathrm{tr}[\tfrac{1}{n}\mathbf{X}(\tfrac{1}{n}\mathbf{X}^\top\mathbf{X} + \nu\mathbf{I}_p)^{-1}\mathbf{X}^\top] = \mathrm{tr}[(\tfrac{1}{n}\mathbf{X}^\top\mathbf{X} + \nu\mathbf{I}_p)^{-1}\tfrac{1}{n}\mathbf{X}^\top\mathbf{X}] = \mathrm{tr}[\widetilde{\mathbf{L}}_\nu^{\mathrm{ridge}}]. \tag{F.7}$$

Therefore, combining (F.6) and (F.7), for the GCV estimator, we obtain

$$\widetilde{R}(\widetilde{\boldsymbol{\beta}}_\lambda^{\mathrm{ens}}) = \frac{\tfrac{1}{n}\|\mathbf{y} - \mathbf{X}\widetilde{\boldsymbol{\beta}}_\lambda^{\mathrm{ens}}\|_2^2}{(1 - \tfrac{1}{n}\mathrm{tr}[\widetilde{\mathbf{L}}_\lambda^{\mathrm{ens}}])^2}$$

$$\simeq \frac{\tfrac{1}{n}\|\mathbf{y} - \mathbf{X}\widetilde{\boldsymbol{\beta}}_\mu^{\mathrm{ridge}}\|_2^2 + \dfrac{\nu''_{\mathrm{num}}\widetilde{\Delta}}{K}}{(1 - \tfrac{1}{n}\mathrm{tr}[\widetilde{\mathbf{L}}_\lambda^{\mathrm{ens}}])^2}$$

$$\simeq \frac{\tfrac{1}{n}\|\mathbf{y} - \mathbf{X}\widetilde{\boldsymbol{\beta}}_\mu^{\mathrm{ridge}}\|_2^2 + \dfrac{\nu''_{\mathrm{num}}\widetilde{\Delta}}{K}}{(1 - \tfrac{1}{n}\mathrm{tr}[\widetilde{\mathbf{L}}_\nu^{\mathrm{ridge}}])^2}$$

$$= \widetilde{R}(\widetilde{\boldsymbol{\beta}}_\mu^{\mathrm{ridge}}) + \frac{\nu''\widetilde{\Delta}}{K},$$

where $\nu''$ is as defined in (F.3). This finishes the proof of the GCV asymptotics decomposition.

**Part 3: GCV inconsistency.** The inconsistency follows from the asymptotic mismatch of $\nu'$ and $\nu''$. To show the mismatch, it suffices to show that

$$\tfrac{1}{p}\mathrm{tr}\big[\tfrac{1}{n}\mathbf{X}\boldsymbol{\Sigma}\mathbf{X}^\top\big(\tfrac{1}{n}\mathbf{X}\mathbf{X}^\top + \nu\mathbf{I}_n\big)^{-2}\big] \not\simeq \frac{\tfrac{1}{p}\mathrm{tr}\big[\tfrac{1}{n}\mathbf{X}(\tfrac{1}{n}\mathbf{X}^\top\mathbf{X})\mathbf{X}^\top\big(\tfrac{1}{n}\mathbf{X}\mathbf{X}^\top + \nu\mathbf{I}_n\big)^{-2}\big]}{\big(1 - \tfrac{1}{n}\mathbf{X}\mathbf{X}^\top(\tfrac{1}{n}\mathbf{X}\mathbf{X}^\top + \nu\mathbf{I}_n)^{-1}\big)^2}.$$

This is shown in Lemma 21.

This concludes all the three parts and finishes the proof. □

HELPER LEMMAS FOR THE PROOF OF PROPOSITION 7

**Lemma 20** (Quadratic risk decomposition for the ensemble estimator for observation sketch). *Assume the conditions of Lemma 15. Let $\boldsymbol{\Psi}$ be any positive semidefinite matrix with uniformly bounded operator norm, that is independent of $(\mathbf{T}_k)_{k=1}^K$. Let $\boldsymbol{\beta}_0 \in \mathbb{R}^p$ be any vector with uniformly bounded Euclidean norm, that is independent of $(\mathbf{T}_k)_{k=1}^K$. Consider the ensemble estimator obtained with observation sketch as defined in* (13). *Then, under Assumptions A and B, for $\lambda > \widetilde{\lambda}_0$ and all $K$,*

$$(\widetilde{\boldsymbol{\beta}}_\lambda^{\mathrm{ens}} - \boldsymbol{\beta}_0)^\top \boldsymbol{\Psi} (\widetilde{\boldsymbol{\beta}}_\lambda^{\mathrm{ens}} - \boldsymbol{\beta}_0) \simeq (\widetilde{\boldsymbol{\beta}}_\nu^{\mathrm{ridge}} - \boldsymbol{\beta}_0)^\top \boldsymbol{\Psi} (\widetilde{\boldsymbol{\beta}}_\nu^{\mathrm{ridge}} - \boldsymbol{\beta}_0) + \frac{\nu'_{\boldsymbol{\Psi}} \widetilde{\Delta}}{K},$$

*where $\nu$ is as defined in* (B.10)*, $\nu'_{\boldsymbol{\Psi}}$ is as defined in* (B.14)*, $\widetilde{\Delta}$ is as defined in Proposition 19.*

*Proof.* The proof follows analogously to the proof of Lemma 17, except now we use the first- and second-order equivalences from Lemmas 14 and 15 for observation sketch (instead of feature sketch). We omit the details. $\qquad\square$

**Lemma 21** (Asymptotic non-equivalence of risk and GCV inflation factors for observation sketch). *Under Assumption B,*

$$\frac{1}{n}\mathrm{tr}\big[(\tfrac{1}{n}\mathbf{X}\mathbf{X}^\top + \nu\mathbf{I}_n)^{-1}(\tfrac{1}{n}\mathbf{X}\boldsymbol{\Sigma}\mathbf{X}^\top)(\tfrac{1}{n}\mathbf{X}\mathbf{X}^\top + \nu\mathbf{I}_n)^{-1}\big]$$
$$\not\simeq \frac{\frac{1}{n}\mathrm{tr}\big[(\tfrac{1}{n}\mathbf{X}\mathbf{X}^\top + \nu\mathbf{I}_n)^{-1}(\tfrac{1}{n}\mathbf{X}\widehat{\boldsymbol{\Sigma}}\mathbf{X}^\top)(\tfrac{1}{n}\mathbf{X}\mathbf{X}^\top + \nu\mathbf{I}_n)^{-1}\big]}{\big(1 - \frac{1}{n}\mathrm{tr}[\widehat{\boldsymbol{\Sigma}}(\widehat{\boldsymbol{\Sigma}} + \nu\mathbf{I}_p)^{-1}]\big)^2}. \tag{F.8}$$

*Proof.* We will first derive asymptotic equivalents for both the left- and right-hand sides of (F.8). Then we will show that the difference in their asymptotic equivalents is non-zero.

**Asymptotic equivalent for left-hand side.** Using Woodbury matrix identity, we can write

$$\mathrm{tr}\big[(\tfrac{1}{n}\mathbf{X}\mathbf{X}^\top + \nu\mathbf{I}_n)^{-1}(\tfrac{1}{n}\mathbf{X}\boldsymbol{\Sigma}\mathbf{X}^\top)(\tfrac{1}{n}\mathbf{X}\mathbf{X}^\top + \nu\mathbf{I}_n)^{-1}\big]$$
$$= \mathrm{tr}\big[(\widehat{\boldsymbol{\Sigma}} + \nu\mathbf{I}_p)^{-1}\widehat{\boldsymbol{\Sigma}}(\widehat{\boldsymbol{\Sigma}} + \nu\mathbf{I}_p)^{-1}\boldsymbol{\Sigma}\big]$$
$$= \mathrm{tr}\big[(\widehat{\boldsymbol{\Sigma}} + \nu\mathbf{I}_p)^{-1}\boldsymbol{\Sigma}(\mathbf{I}_p - \nu(\widehat{\boldsymbol{\Sigma}} + \nu\mathbf{I}_p)^{-1})\big]$$
$$= \mathrm{tr}\big[(\widehat{\boldsymbol{\Sigma}} + \nu\mathbf{I}_p)^{-1}\boldsymbol{\Sigma}\big] - \nu\mathrm{tr}\big[(\widehat{\boldsymbol{\Sigma}} + \nu\mathbf{I}_p)^{-1}\boldsymbol{\Sigma}(\widehat{\boldsymbol{\Sigma}} + \nu\mathbf{I}_p)^{-1}\big].$$

**Asymptotic equivalent for right-hand side.** Similarly, the numerator of GCV can be expressed as

$$\mathrm{tr}\big[(\tfrac{1}{n}\mathbf{X}\mathbf{X}^\top + \nu\mathbf{I}_n)^{-1}(\tfrac{1}{n}\mathbf{X}\widehat{\boldsymbol{\Sigma}}\mathbf{X}^\top)(\tfrac{1}{n}\mathbf{X}\mathbf{X}^\top + \nu\mathbf{I}_n)^{-1}\big]$$
$$= \mathrm{tr}\big[(\widehat{\boldsymbol{\Sigma}} + \nu\mathbf{I}_p)^{-1}\widehat{\boldsymbol{\Sigma}}(\widehat{\boldsymbol{\Sigma}} + \nu\mathbf{I}_p)^{-1}\widehat{\boldsymbol{\Sigma}}\big]$$
$$= \mathrm{tr}\big[(\widehat{\boldsymbol{\Sigma}} + \nu\mathbf{I}_p)^{-1}\widehat{\boldsymbol{\Sigma}}\big] - \nu\mathrm{tr}\big[(\widehat{\boldsymbol{\Sigma}} + \nu\mathbf{I}_p)^{-1}\widehat{\boldsymbol{\Sigma}}(\widehat{\boldsymbol{\Sigma}} + \nu\mathbf{I}_p)^{-1}\big].$$

From Lemma 18, we have that

$$\nu\mathrm{tr}\big[(\widehat{\boldsymbol{\Sigma}} + \nu\mathbf{I}_p)^{-1}\boldsymbol{\Sigma}(\widehat{\boldsymbol{\Sigma}} + \nu\mathbf{I}_p)^{-1}\big] \simeq \frac{\nu\mathrm{tr}\big[(\widehat{\boldsymbol{\Sigma}} + \nu\mathbf{I}_p)^{-1}\widehat{\boldsymbol{\Sigma}}(\widehat{\boldsymbol{\Sigma}} + \nu\mathbf{I}_p)^{-1}\big]}{\big(1 - \frac{1}{n}\mathrm{tr}[\widehat{\boldsymbol{\Sigma}}(\widehat{\boldsymbol{\Sigma}} + \nu\mathbf{I}_p)^{-1}]\big)^2}.$$

**Mismatching of asymptotic equivalents.** Observe that

$$\frac{1}{n}\mathrm{tr}\big[(\tfrac{1}{n}\mathbf{X}\mathbf{X}^\top + \nu\mathbf{I}_n)^{-1}(\tfrac{1}{n}\mathbf{X}\boldsymbol{\Sigma}\mathbf{X}^\top)(\tfrac{1}{n}\mathbf{X}\mathbf{X}^\top + \nu\mathbf{I}_n)^{-1}\big]$$

$$- \frac{\frac{1}{n}\mathrm{tr}\big[(\tfrac{1}{n}\mathbf{X}\mathbf{X}^\top + \nu\mathbf{I}_n)^{-1}(\tfrac{1}{n}\mathbf{X}\widehat{\boldsymbol{\Sigma}}\mathbf{X}^\top)(\tfrac{1}{n}\mathbf{X}\mathbf{X}^\top + \nu\mathbf{I}_n)^{-1}\big]}{\big(1 - \frac{1}{n}\mathrm{tr}[\widehat{\boldsymbol{\Sigma}}(\widehat{\boldsymbol{\Sigma}} + \nu\mathbf{I}_p)^{-1}]\big)^2}$$

$$\simeq \frac{1}{n}\mathrm{tr}\big[(\widehat{\boldsymbol{\Sigma}} + \nu\mathbf{I}_p)^{-1}\boldsymbol{\Sigma}\big] - \frac{\frac{1}{n}\mathrm{tr}\big[(\widehat{\boldsymbol{\Sigma}} + \nu\mathbf{I}_p)^{-1}\widehat{\boldsymbol{\Sigma}}\big]}{\big(1 - \frac{1}{n}\mathrm{tr}[\widehat{\boldsymbol{\Sigma}}(\widehat{\boldsymbol{\Sigma}} + \nu\mathbf{I}_p)^{-1}]\big)^2}$$

$$\simeq \frac{1}{1 - \frac{1}{n}\mathrm{tr}[\widehat{\boldsymbol{\Sigma}}(\widehat{\boldsymbol{\Sigma}} + \nu\mathbf{I}_p)^{-1}]} - 1 - \frac{\frac{1}{n}\mathrm{tr}\big[(\widehat{\boldsymbol{\Sigma}} + \nu\mathbf{I}_p)^{-1}\widehat{\boldsymbol{\Sigma}}\big]}{\big(1 - \frac{1}{n}\mathrm{tr}[\widehat{\boldsymbol{\Sigma}}(\widehat{\boldsymbol{\Sigma}} + \nu\mathbf{I}_p)^{-1}]\big)^2}$$

$$= -\left(1 - \frac{\frac{1}{n}\mathrm{tr}\big[(\widehat{\boldsymbol{\Sigma}} + \nu\mathbf{I}_p)^{-1}\widehat{\boldsymbol{\Sigma}}\big]}{1 - \frac{1}{n}\mathrm{tr}\big[(\widehat{\boldsymbol{\Sigma}} + \nu\mathbf{I}_p)^{-1}\widehat{\boldsymbol{\Sigma}}\big]}\right)^2.$$

The last line is in general not equal to $0$, proving the desired asymptotic mismatch. This finishes the proof. $\qquad\square$

### F.2 Correction using ensemble trick for GCV with observation sketch

Below we outline a method that corrects GCV for the sketched ensemble estimator with observation sketch. The idea of the method is to estimate the error term in the mismatch in Lemma 21 using a combination of the ensemble trick and our second-order sketched equivalences. The correction takes a complicated form involving both the unsketched and sketched data. We are not aware of any method that uses only sketched data.

1. Estimate $\nu$ from the data and sketch using the subordination relation (B.10).

2. Estimate the following two quantities that appear in the inflation of the GCV decomposition:

$$\widetilde{\Delta} = \frac{1}{n}\mathbf{y}^\top(\tfrac{1}{n}\mathbf{X}\mathbf{X}^\top + \nu\mathbf{I}_n)^{-2}\mathbf{y} \quad \text{and} \quad C_1 = \frac{\frac{1}{n}\mathrm{tr}\big[(\tfrac{1}{n}\mathbf{X}\mathbf{X}^\top + \nu\mathbf{I}_n)^{-1}(\tfrac{1}{n}\mathbf{X}\widehat{\boldsymbol{\Sigma}}\mathbf{X}^\top)(\tfrac{1}{n}\mathbf{X}\mathbf{X}^\top + \nu\mathbf{I}_n)^{-1}\big]}{\big(1 - \frac{1}{n}\mathrm{tr}[\widehat{\boldsymbol{\Sigma}}(\widehat{\boldsymbol{\Sigma}} + \nu\mathbf{I}_p)^{-1}]\big)^2}.$$

3. Use ensemble trick as explained in Section 5 on $\widetilde{R}(\widetilde{\boldsymbol{\beta}}_\lambda^{\mathrm{ens}}) = \widetilde{R}(\widetilde{\boldsymbol{\beta}}_\nu^{\mathrm{ridge}}) + \frac{\nu''\widetilde{\Delta}}{K}$ with $K = 1$ and $K = 2$ to estimate $\widetilde{R}(\widetilde{\boldsymbol{\beta}}_\nu^{\mathrm{ridge}})$ first and then estimate the following component:

$$C = \nu''\widetilde{\Delta} = -\frac{\partial\nu}{\partial\lambda}\lambda^2 \mathscr{S}'_{\mathbf{T}\mathbf{T}^\top}\big(-\tfrac{1}{n}\mathrm{tr}\big[\tfrac{1}{n}\mathbf{X}\mathbf{X}^\top\big(\tfrac{1}{n}\mathbf{X}\mathbf{X}^\top + \nu\mathbf{I}_n\big)^{-1}\big]\big)C_1\widetilde{\Delta}.$$

4. Eliminate $\widetilde{\Delta}$ from $C$ to get an estimate for the following component:

$$C_2 = -\frac{\partial\nu}{\partial\lambda}\lambda^2 \mathscr{S}'_{\mathbf{T}\mathbf{T}^\top}\big(-\tfrac{1}{n}\mathrm{tr}\big[\tfrac{1}{n}\mathbf{X}\mathbf{X}^\top\big(\tfrac{1}{n}\mathbf{X}\mathbf{X}^\top + \nu\mathbf{I}_n\big)^{-1}\big]\big).$$

5. Then use the following equivalence to estimate:

$$C_1' = \frac{1}{n}\mathrm{tr}\big[(\tfrac{1}{n}\mathbf{X}\mathbf{X}^\top + \nu\mathbf{I}_n)^{-1}(\tfrac{1}{n}\mathbf{X}\boldsymbol{\Sigma}\mathbf{X}^\top)(\tfrac{1}{n}\mathbf{X}\mathbf{X}^\top + \nu\mathbf{I}_n)^{-1}\big]$$

$$\simeq \frac{\frac{1}{n}\mathrm{tr}\big[(\tfrac{1}{n}\mathbf{X}\mathbf{X}^\top + \nu\mathbf{I}_n)^{-1}(\tfrac{1}{n}\mathbf{X}\widehat{\boldsymbol{\Sigma}}\mathbf{X}^\top)(\tfrac{1}{n}\mathbf{X}\mathbf{X}^\top + \nu\mathbf{I}_n)^{-1}\big]}{\big(1 - \frac{1}{n}\mathrm{tr}[\widehat{\boldsymbol{\Sigma}}(\widehat{\boldsymbol{\Sigma}} + \nu\mathbf{I}_p)^{-1}]\big)^2}$$

$$- \left(1 - \frac{\frac{1}{n}\mathrm{tr}\big[(\widehat{\boldsymbol{\Sigma}} + \nu\mathbf{I}_p)^{-1}\widehat{\boldsymbol{\Sigma}}\big]}{1 - \frac{1}{n}\mathrm{tr}\big[(\widehat{\boldsymbol{\Sigma}} + \nu\mathbf{I}_p)^{-1}\widehat{\boldsymbol{\Sigma}}\big]}\right)^2.$$

6. Finally, obtain the corrected estimate for risk using the GCV asymptotics decomposition from Proposition 19:

$$\widetilde{R}(\widetilde{\boldsymbol{\beta}}_\nu^{\mathrm{ridge}}) + \frac{C_2 C_1' \widetilde{\Delta}}{K}.$$

### F.3 Anisotropic sketching and generalized ridge regression

Using structural equivalences to anisotropic sketching matrices and generalized ridge regression, one can extend our results to anisotropic sketching and generalized ridge regression. Specifically, let $\mathbf{R} \in \mathbb{R}^{p \times p}$ be an invertible positive semidefinite matrix with bounded operator norm. Consider generalized ridge regression with a anisotropic sketching matrices $\widetilde{\mathbf{S}}_k = \mathbf{R}^{1/2} \mathbf{S}_k$ for $k \in [K]$:

$$\widehat{\boldsymbol{\beta}}_\lambda^k = \widetilde{\mathbf{S}}_k \widehat{\boldsymbol{\beta}}_\lambda^{\widetilde{\mathbf{S}}_k}, \quad \text{where} \quad \widehat{\boldsymbol{\beta}}_\lambda^{\widetilde{\mathbf{S}}_k} = \underset{\boldsymbol{\beta} \in \mathbb{R}^q}{\arg\min} \frac{1}{n} \left\| \mathbf{y} - \mathbf{X} \widetilde{\mathbf{S}}_k \boldsymbol{\beta} \right\|_2^2 + \lambda \left\| \mathbf{G}^{1/2} \boldsymbol{\beta} \right\|_2^2, \tag{F.9}$$

where $\mathbf{G} \in \mathbb{R}^{p \times p}$ is a positive definite matrix with bounded operator norm. Let $\widehat{\boldsymbol{\beta}}_\lambda^{\text{ens}}$ be the ensemble estimator defined analogously as in (2) and GCV defined analogously as in (5). Using Corollary 7.1 of LeJeune et al. (2022), all of our results carry in this case in a straightforward manner.

## G Experimental details

All experiments were run in less than 1 hour on a Macbook Air (M1, 2020) and coded in Python using standard scientific computing packages. CountSketch (Charikar et al., 2004) is implemented by generating a sparse matrix corresponding to the hash function, and due to rounding of the size parameters to match theoretical rates, we cannot choose arbitrary sketch sizes and are often restricted to non-standard sequences. Instead of the SRHT, which requires an implementation of the fast Walsh–Hadamard transform not readily available and platform-independent in Python (and also suffers statistically from zero-padding issues as described by LeJeune et al., 2022), we use a subsampled randomized discrete cosine transform (SRDCT), which is fast, widely available, and does not suffer the statistical drawbacks. All sketches are normalized such that $\mathbf{S}\mathbf{S}^\top \simeq \mathbf{I}_p$ and therefore $\frac{1}{p} \|\mathbf{S}^\top \mathbf{x}\|_2^2 \simeq \frac{1}{p} \|\mathbf{x}\|_2^2$.

### G.1 GCV paths in Figure 1

For this experiment, over 100 trials, we sampled $\mathbf{X}$ with each row $\mathbf{x} \sim \mathcal{N}(\mathbf{0}, \mathbf{I}_p)$ and generated $y = \mathbf{x}^\top \boldsymbol{\beta} + \xi$ for $\boldsymbol{\beta} \sim \mathcal{N}(\mathbf{0}, \frac{1}{p} \mathbf{I}_p)$ and independent noise $\xi \sim \mathcal{N}(0, 1)$. We have $n = 500$ and $p = 600$, and our sketching ensembles have $K = 5$. For the left plot, we fix $q = 441$, which is an allowed sketch size for CountSketch. For the right plot, we fix $\lambda = 0.2$ and sweep through the choices of $q$ which are allowed by CountSketch, which are $q \in \{63, 126, 189, 252, 315, 378, 441, 504, 567\}$.

### G.2 Real data in Figure 2

For both real data datasets, we fit our sketched ridge regressors on centered sketched data and responses and then added the mean of the training responses to any outputs. For both datasets, we sketched using CountSketch, which is among the most computationally efficient sketches, especially for sparse data as in RCV1. We plot risk on a `symlog` scale of $\lambda$ with linear region from $-10$ to $10$.

For RCV1 (Lewis et al., 2004), we downloaded the data from `scikit-learn` (Buitinck et al., 2013). We discarded all labels except for `GCAT` and `CCAT` and then discarded all examples that did not uniquely fall into one of these categories. These became our binary class labels. We then randomly subsampled 20000 training points and 5000 test points, and discarded any features that took value 0 for all train and test points. This left 30617 features, and we used $q = 515$ for CountSketch. We normalized each data vector $\mathbf{x}$ such that $\|\mathbf{x}\|_2 = \sqrt{p}$, preserving sparsity. We then fit ensembles of size $K = 5$, reporting error over 10 random trials.

For RNA-Seq (Weinstein et al., 2013), we downloaded the data from the UCI Machine Learning repository (Dua & Graff, 2017) at: https://archive.ics.uci.edu/ml/datasets/gene+expression+cancer+RNA-Seq. We discarded all examples that were labeled neither `BRCA` nor `KIRC`, the most common classes, leaving 446 observations, which were split into a training set of 356 and test set of 90. We then z-scored each of the 20223 features using the training data statistics. We fit ensembles of size $K = 5$, reporting error over 10 random trials.

### G.3   Prediction intervals in Figure 3

For this experiment, we use SRDCT sketches. For each choice of

$$q \in \{80, 180, 280, 380, 480, 580, 680, 780, 880, 980\},$$

we generated data over 30 trials in similar manner to the experiment in Figure 1: for $n = 1500$ and $p = 1000$ we sampled $\mathbf{X}$ with each row $\mathbf{x} \sim \mathcal{N}(\mathbf{0}, \mathbf{I}_p)$, but we generated $y = g(\mathbf{x}^\top \boldsymbol{\beta})$ for $\boldsymbol{\beta} \sim \mathcal{N}(\mathbf{0}, \frac{1}{p}\mathbf{I}_p)$ and $g$ the soft-thresholding operator:

$$g(u) = \begin{cases} u - 1 & \text{if } u > 1 \\ 0 & \text{if } -1 \leqslant u \leqslant 1 \\ u + 1 & \text{if } u < -1 \end{cases}.$$

We compute the 95% and 99% prediction intervals by identifying the 2.5% and 0.1% tail intervals of the GCV corrected residuals $(y - z)$: $(y, z) \sim \widehat{P}_\lambda^{\text{ens}}$ and evaluate coverage on 1500 test residuals $y_0 - \mathbf{x}_0^\top \widehat{\boldsymbol{\beta}}_\lambda^{\text{ens}}$. We plot 2D histograms of $P_\lambda^{\text{ens}}$ (empirical using test points) and $\widehat{P}_\lambda^{\text{ens}}$ (using training points) on a logarithmic color scale.

### G.4   Details for Figure 4

For this experiment, we use SRDCT sketches. For $n = 600$ and $p = 800$, for each trial, we generate Gaussian data with $\boldsymbol{\Sigma} = \text{diag}(\mathbf{a})$, where $a_i = 2/(1 + 30t_i)$, where $t_i$ are $p$ linearly spaced values from 0 to 1. We generate $y = \mathbf{x}^\top \boldsymbol{\beta} + \xi$ for $\boldsymbol{\beta}_{1:80} \sim \mathcal{N}(\mathbf{0}, \frac{1}{80}\mathbf{I}_{80})$ and $\boldsymbol{\beta}_{81:} = \mathbf{0}$ and $\xi \sim \mathcal{N}(0, 4)$.

We evaluate the mapping from $\mu \mapsto \lambda$ for feature sketching by inverting the subordination relation

$$\mu = \lambda \mathscr{S}_{\mathbf{SS}^\top} \left( -\tfrac{1}{p}\text{tr}\big[\mathbf{S}^\top \widehat{\boldsymbol{\Sigma}}\mathbf{S}\big(\mathbf{S}^\top \widehat{\boldsymbol{\Sigma}}\mathbf{S} + \lambda \mathbf{I}_q\big)^{-1}\big]\right)$$

for a single random generation of data and sketch. For observation sketching, we do the same but use the relation

$$\mu = \lambda \mathscr{S}_{\mathbf{SS}^\top} \left( -\tfrac{1}{n}\text{tr}\big[\tfrac{1}{n}\mathbf{X}^\top \mathbf{TT}^\top \mathbf{X}\big(\tfrac{1}{n}\mathbf{X}^\top \mathbf{TT}^\top \mathbf{X} + \lambda \mathbf{I}_p\big)^{-1}\big]\right).$$

We evaluate the mapping from $\mu \mapsto \alpha$ using the same method as in feature sketching, except we take $q$ as 20 values logarithmically spaced between 1 and 800, rounded down to the nearest integer. For computing the curves where we vary $\alpha$, we pre-sketch using $q = p$ and then subsample and normalize to obtain the sketched data for each desired $q$.

### G.5   Verification of convergence rate for sketched ensembles

We demonstrate that both GCV and risk for sketched ensembles converge at rate $1/K$ to the equivalent ridge for sketched ensembles in Figure G.1. For $n = 140$ and $p = 200$, for a single trial, we generate Gaussian data with $\boldsymbol{\Sigma} = \mathbf{I}_p$ and $y = \mathbf{x}^\top \boldsymbol{\beta} + \xi$ for $\boldsymbol{\beta} \sim \mathcal{N}(\mathbf{0}, \frac{1}{p}\mathbf{I}_p)$ and $\xi \sim \mathcal{N}(0, 1)$. We fit 1000 sketched predictors for $\lambda = 0.1$ for each sketch using $q = 156$, and then successively build ensembles of size $K$ by taking the first $K$ predictors. We then subtract the risk of the unsketched predictor at the equivalent $\mu$, determined numerically to be 0.283 for Gaussian sketching, 0.157 for orthogonal sketching, 0.281 for CountSketch, and 0.157 for SRDCT.

## H   Complexity comparisons

If the sketch size is sufficiently small, the ensemble trick in (11) can be more computationally efficient than computing GCV directly on the unsketched data or using $k$-fold CV. Memory savings are straightforward, as we only need to work with the $n \times q$ (feature sketching) or $m \times p$ (observation sketching) data matrix, so we focus here on computation time complexity. For concreteness, we consider dense $\mathbf{X}$. With additional care this could be extended to sparse $\mathbf{X}$ and $\mathbf{XS}$, although computation time improvements are harder to obtain with sketching when comparing to iterative solvers on very sparse data.

Letting $\widetilde{\mathbf{X}} \in \mathbb{R}^{\widetilde{n} \times \widetilde{p}}$ denote the unsketched, sketched, or $k$-fold training data depending on context, and similarly $\widetilde{\mathbf{y}}$ and $\widetilde{\lambda}$, the computations are dominated by computing the following two quantities:

$$\widetilde{\boldsymbol{\beta}} = \big(\tfrac{1}{\widetilde{n}}\widetilde{\mathbf{X}}^\top \widetilde{\mathbf{X}} + \widetilde{\lambda}\mathbf{I}_{\widetilde{p}}\big)^{-1}\tfrac{1}{\widetilde{n}}\widetilde{\mathbf{X}}^\top \widetilde{\mathbf{y}} \quad \text{and} \quad \tfrac{1}{\widetilde{n}}\text{tr}[\widetilde{\mathbf{L}}_\lambda] = \tfrac{1}{\widetilde{n}}\text{tr}\Big[\widetilde{\mathbf{X}}\big(\tfrac{1}{\widetilde{n}}\widetilde{\mathbf{X}}^\top \widetilde{\mathbf{X}} + \widetilde{\lambda}\mathbf{I}_{\widetilde{p}}\big)^{-1}\tfrac{1}{\widetilde{n}}\widetilde{\mathbf{X}}^\top\Big].$$

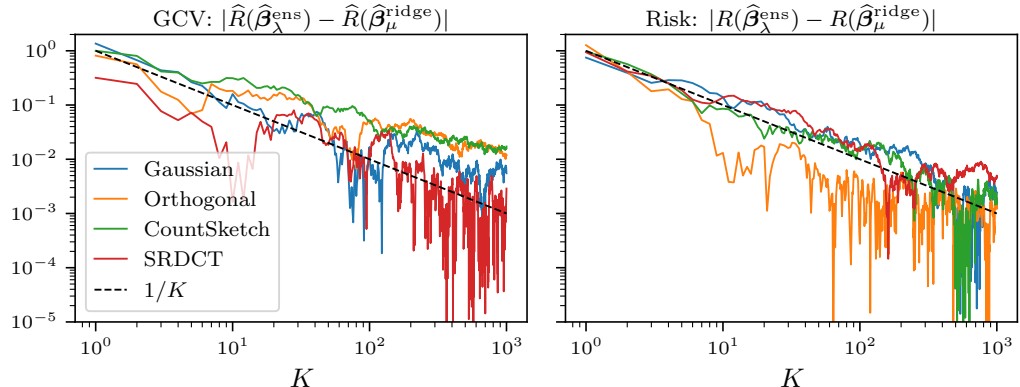

Figure G.1: **Both GCV and risk converge at rate** $1/K$ **to the equivalent ridge for sketched ensembles.** See Appendix G.5 for the setup details.

For the ensemble trick, we must know which value of $\mu$ corresponds to the value of $\lambda$ we use, but this can be computed using the subordination relation in Theorem 1 and $\frac{1}{\widetilde{n}}\mathrm{tr}[\widetilde{\mathbf{L}}_\lambda]$. In high dimensions, this trace is well approximated (Hutchinson, 1989) by:

$$\frac{1}{\widetilde{n}}\mathrm{tr}[\widetilde{\mathbf{L}}_\lambda] \approx \frac{1}{\widetilde{n}}\mathbf{z}^\top \big(\frac{1}{\widetilde{n}}\widetilde{\mathbf{X}}^\top\widetilde{\mathbf{X}} + \widetilde{\lambda}\mathbf{I}_{\widetilde{p}}\big)^{-1}\frac{1}{\widetilde{n}}\widetilde{\mathbf{X}}^\top\widetilde{\mathbf{X}}\mathbf{z},$$

where $\mathbf{z} \in \mathbb{R}^{\widetilde{p}\times\widetilde{p}}$ has i.i.d. Rademacher ($\pm 1$) entries (or if $\widetilde{n} < \widetilde{p}$, we could use $\mathbf{z} \in \mathbb{R}^{\widetilde{n}}$). Thus, the computations are dominated by solving linear system with the matrix $\frac{1}{\widetilde{n}}\widetilde{\mathbf{X}}^\top\widetilde{\mathbf{X}} + \widetilde{\lambda}\mathbf{I}_{\widetilde{p}}$ (or $\frac{1}{\widetilde{n}}\widetilde{\mathbf{X}}\widetilde{\mathbf{X}}^\top + \widetilde{\lambda}\mathbf{I}_{\widetilde{n}}$ if dimensions are preferable). We now specialize to a couple of different solvers that could be used to solve these systems, assuming that the cost of sketching is amortized or otherwise negligible compared to solving. In what follows, when we use big $\mathcal{O}$ notation, we mean that the rates scale with universal constants independent of $\widetilde{n}, \widetilde{p}, n, p, m, q, \lambda$.

## H.1 DIRECT SOLVER

A direct exact solve of this system has cost $\mathcal{O}(\widetilde{n}\widetilde{p}\min\{\widetilde{n}, \widetilde{p}\})$. For unsketched GCV ($\widetilde{n} = n$, $\widetilde{p} = p$), we must solve two of these systems, one for the parameter estimator and the other for the randomized trace, giving a total cost of $\mathcal{O}(2np\min\{n, p\})$. For unsketched $k$-fold CV, there is only the cost of computing the parameter estimator on the $\widetilde{n} = \frac{k-1}{k}n$ data points of dimension $\widetilde{p} = p$, which is then evaluated on the left-out fold, which is comparatively inexpensive. Since this must be done $k$ times, the total cost is $\mathcal{O}((k - 1)np\min\{\frac{k-1}{k}n, p\})$. For the ensemble trick, we must evaluate GCV on two separate parameter estimators ($\widetilde{n} = m$, $\widetilde{p} = q$), for a total cost of $\mathcal{O}(4mq\min\{m, q\})$.

## H.2 ITERATIVE SOLVER

An approximate solution is generally acceptable for a risk estimate. Thus, a direct solve is often unnecessary, and an iterative solver such as a Krylov subspace method can offer considerable computational gains. In particular, instead of a cost of $\mathcal{O}(\widetilde{n}\widetilde{p}\min\{\widetilde{n}, \widetilde{p}\})$, the cost becomes $\mathcal{O}(\widetilde{n}\widetilde{p}\sqrt{\widetilde{\kappa}_{\widetilde{\lambda}}}\log\frac{1}{\varepsilon})$ to reach an $\epsilon$-accurate solution, where $\widetilde{\kappa}_\lambda$ is the condition number of $\widetilde{\mathbf{X}}^\top\widetilde{\mathbf{X}} + \widetilde{\lambda}\mathbf{I}_{\widetilde{p}}$.

We can update the computations for the dense solve accordingly, except that for each case (unsketched GCV, unsketched $k$-fold CV, ensemble trick) the matrix will have a different condition number, which we denote as $\kappa_\mu$, $\kappa_{\mu,k}$, and $\kappa_{\lambda,m,q}$, respectively.

## H.3 COMPARING COMPLEXITIES

We list the computational complexities of the various methods for both direct and iterative solvers.

For the direct solver, the ensemble trick is always beneficial over unsketched GCV as long as $q < p/2$ (for feature sketching) or $m < n/2$ (for observation sketching), regardless of the relative sizes of $n$

| Method (regime) | Unsketched GCV | Unsketched $k$-fold CV | Ensemble trick |
|---|---|---|---|
| Direct solver | $\mathcal{O}(2np\min\{n,p\})$ | $\mathcal{O}((k-1)np\min\{\frac{k-1}{k}n,p\})$ | $\mathcal{O}(4mq\min\{m,q\})$ |
| Iterative solver | $\mathcal{O}(2np\sqrt{\kappa_\mu}\log\frac{1}{\epsilon})$ | $\mathcal{O}((k-1)np\sqrt{\kappa_{\mu,k}}\log\frac{1}{\epsilon})$ | $\mathcal{O}(4mq\sqrt{\kappa_{\mu,m,q}}\log\frac{1}{\epsilon})$ |

Table 5: Complexity comparison for risk estimation in ridge regression.

and $p$. If we also know that $p < n$, then the ensemble trick with feature sketching is beneficial as long as $q < p/\sqrt{2}$, and an analogous result holds for observation sketching if $m < n/\sqrt{2}$. If one were to sketch both observations and features by a factor of $\alpha$, it suffices to use $\alpha < 2^{-1/3} \approx 0.78$. Meanwhile, $k$-fold CV is never competitive for $k = 5$ or $10$.

For the iterative solver, in order to compare the complexity, we need to know how the condition numbers change with sketching, which is not obvious. We can gain some insight, however, by considering very small sketches. As we observed in Appendix A.3.2, all sketch types seem to have similar subordination relations for small sketches, so we can specialize to Gaussian sketches for simplicity. As the sketch size decreases, all eigenvalues of $\mathbf{S}^\top\widehat{\boldsymbol{\Sigma}}\mathbf{S}$ concentrate around a single value, in the extreme limit of $q = 1$ converging to $\mathbf{s}^\top\widehat{\boldsymbol{\Sigma}}\mathbf{s} \approx \mathrm{tr}[\widehat{\boldsymbol{\Sigma}}]$. This means that the condition number of $\mathbf{S}^\top\widehat{\boldsymbol{\Sigma}}\mathbf{S} + \lambda\mathbf{I}_q$ tends toward 1 as sketches get smaller, meaning that sketching better conditions the system such that $\kappa_{\mu,m,q} \leqslant \kappa_\mu$. We then again have the same conclusion as in the direct solver case, that as long as $m < n/2$ or $q < p/2$, then using the ensemble trick is beneficial over unsketched GCV. And similarly, $k$-fold CV is not competitive.

