# OpenReview forum: "Asymptotically Free Sketched Ridge Ensembles: Risks, Cross-Validation, and Tuning"
_ICLR.cc/2024/Conference — ICLR 2024 spotlight_

### Official Review · Reviewer_GR2e · 2023-10-20

**Soundness:** 3 good
**Presentation:** 3 good
**Contribution:** 3 good
**Rating:** 6
**Confidence:** 4

**Summary:**

This paper established consistency of generalized cross validation for estimating prediction risks of sketched ridge regression that enables it to fast tune ensemble parameters. The authors further proposed an resembling trick so that the risk for unsketched ridge regression can be estimated through GCV using small sketched ridge ensembles. Simulations are conducted to validate the theoretical results.

**Strengths:**

The paper gives asymptotic of squared risk and its GCF estimator for sketched ridge regression so that an it is intuitively understandable as the implicit unsketched ridge regression risk and an inflation term due to randomness of the sketch that is controlled by ensemble size. And this is exploited to provide a method to tune unsketched ridge regression using only sketched ensembles. None of the assumptions are very strong for these theoretical results.

**Weaknesses:**

While the results in this paper is interesting, the authors failed to illustrate while tuning (ensembled) sketched ridge is preferred over tuning unsketched ridge regression. It would be better if they provide some intuition and explanation to the results, especially for readers who are not that familiar with sketching.

**Questions:**

Because GCV and risk for sketched ensembles converge at rate 1/K to the equivalent ridge for sketched ensembles, does this imply the larger the K the faster the convergence and the better it is? It there a downside if K is too large?
Could the authors why the result that tuning (ensembled) sketched ridge is equivalent to tuning unsketched ridge regression is useful?

---

> ### Author Response · Authors · 2023-11-18
> **Specific response to the Reviewer GR2e**
>
> Thank you for your constructive feedback and for acknowledging the strengths of our paper.
> We appreciate the opportunity to clarify aspects of our work further.
> We hope that these responses address your concerns and clarify the practical contributions of our work.
>
> > **The authors failed to illustrate while tuning (ensembled) sketched ridge is preferred over tuning unsketched ridge regression.**
>
> We have now included a new Appendix H to clarify when one might prefer to tune ensembled sketched ridge regression over unsketched ridge regression from a computational perspective.
> As we show, the sketched ensemble trick for tuning unsketched ridge often always offers a computational advantage if $q < p/2$, and can sometimes be preferable more generally.
>
> Moreover, from a statistical standpoint, GCV is preferable to traditional CV due to the absence of data splitting, which preserves the consistency of the risk estimator compared to CV.
>
> > **Does this imply the larger the $K$ the faster the convergence and the better it is? Is there a downside if $K$ is too large?**
>
> While a larger $K$ indeed means being closer to equivalent unsketched ridge risk, there is a trade-off due to increased computational demands.
> In practice, we typically find a sweet spot where $K$ is large enough to yield a good approximation to the unsketched ridge (with a properly chosen sketch size), but not so large as to be computationally prohibitive.
> Our experiments suggest that $K$ between 10 and 50 is often sufficient, as long as the constant in the $1/K$ term is not excessively large.
> This ensures that any inflation due to the randomness of the sketch is controlled by the predictive power of the response.
>
> > **Could the authors why the result that tuning (ensembled) sketched ridge is equivalent to tuning unsketched ridge regression is useful?**
>
> The theoretical equivalence between the space of sketched and unsketched ridge predictors is not only of theoretical interest but also has practical implications.
> It reassures practitioners that using a large enough ensemble of sketched predictors does not compromise statistical efficiency compared to unsketched ridge regression.
> This means that one can benefit from the computational efficiencies (especially parallelization and reduced memory footprint) of sketching without a loss in statistical performance, which is particularly relevant for large-scale problems where computational resources or time are at a premium.

---

> > ### Comment · Reviewer_GR2e · 2023-11-22
> > **Reviewer response**
> >
> > The authors' response is satisfactory and it would be great if some of those explanations could be put in the main text.

---

> ### Author Response · Authors · 2023-11-22
> **Follow-up specific response to the Reviewer GR2e**
>
> Thanks for the comment! We have now squeezed in parts of the explanation above in the revised main paper. Please find pointers to Appendix H in Sections 2 and 5, and additional comments after Proposition 6. These are inked **blue** in the revision, as are all other changes indicated in the general response at the top.

---

### Official Review · Reviewer_WHDZ · 2023-11-02

**Soundness:** 3 good
**Presentation:** 3 good
**Contribution:** 3 good
**Rating:** 8
**Confidence:** 3

**Summary:**

The current paper considers generalized cross validation (GCV) for sketched ridge regression ensembles. Specifically, sketching is done across different features and ensembles are based on finite sketches. The paper first derives the asymptotics of squared risk and its GCV estimator (section 2) and then extends to more general subquadratic prediction risk functionals (section 3). The paper also proposes a method for estimating the risk of unsketched ridge regression using sketched ridge ensembles (section 4). All the theoretical results are illustrated using both synthetic and real datasets with CountSketch and subsampled randomized discrete cosine transforms.

**Strengths:**

1. Overall the paper is very well written and persuasive. The experimental results are very well summarized in the figures.

2. It is impressive that the current paper considers all asymptotically free sketched ensembles and allows for zero or negative penalty
 in ridge regularization.

3. Distributional consistency in Corollary 5 is nice: this allows for classification errors and construction of prediction intervals among other things.

4. It is interesting to know that the finite ensembles by sketching observations do not have GCV consistency, as given in Proposition 7.

**Weaknesses:**

It would be beneficial to provide more details regarding computational aspects in the main text. For example, I presume that it is not necessary to compute $\hat{\beta}_{\lambda}^k$ alone in equation (1).

In other words,  it is only necessary to compute the predicted values
$X \hat{\beta}_{\lambda}^k$.

This seems important because it is not necessary to explicitly premultiply $S_k$ to $\hat{\beta}_{\lambda}^{S_k}$ in equation (1). If I am correct, this point can be emphasized at the end of section 2 when it is discussed that matrix inversions are inexpensive after precomputing $X S_k$.

Generally speaking, it would be helpful to provide more details regarding computational aspects.

**Questions:**

1. In figure 1, it would be good to indicate that SRDCT refers to subsampled randomized discrete cosine transform because SRDCT first appears toward the end of page 2.

2. In proposition 6, $S_k^T S_k$ is assumed to be invertible. How strong is this assumption? Some further remarks might be helpful.

3. The ensemble trick in section 5 seems very useful. However, there is no explicit discussion of computational gains over unsketched GCV. Especially, when $K$ is large as in proposition 6, one might need to rely on parallel computing to fully speed up computations. More discussions might be desirable in terms of computational complexity.

4. Is there a known S-transform for CountSketch? Table 4 in the appendix does not include CountSketch.

[Update after the discussion period] The author(s) responded well with my questions. I am satisfied with their replies. This is indeed a good paper.

---

> ### Author Response · Authors · 2023-11-18
> **Specific response to the Reviewer WHDZ**
>
> Thank you for the thorough review and for the positive comments about our paper.
> We are pleased that you found our work to be well-written and the experimental results compelling.
>
> We appreciate your insightful feedback, which has helped us improve the clarity and depth of our manuscript.
> Your input on the computational aspects has been particularly valuable, and we have made the appropriate enhancements to the text to better communicate these important details.
> Thank you once again for your review and the positive assessment of our work. Below we respond to some of your specific points.
>
> > **Generally speaking, it would be helpful to provide more details regarding computational aspects.**
>
> We agree with you and the other reviewers regarding this point and have now added a new Appendix H providing a detailed computational comparison between unsketched GCV, $k$-fold CV, and the ensemble trick, clarifying that the computational demands decrease for smaller sketch sizes.
> Indeed, as you have written, it is not necessary to "broadcast" $\hat{\beta}_\lambda^{S_k}$ back to $p$-dimensional space, and all computation can (and should) be done only in the sketched domain; our presentation is simply made for mathematical convenience.
> In fact, for sketches based on hash functions such as CountSketch, computing the adjoint of the sketching operator is in general intractable, so this point is critical in sketched machine learning practice.
>
> > **In figure 1, it would be good to indicate that SRDCT refers to subsampled randomized discrete cosine transform**
>
> We have revised the caption of Figure 1 to include a definition for SRDCT, ensuring that it is clear on its first appearance in the text.
>
> > **In proposition 6, $S_k^\top S_k$
> is assumed to be invertible. How strong is this assumption?**
>
> The invertibility assumption for $\mathbf{S}_k^\top \mathbf{S}_k$ is merely technical.
> If it is not invertible, it is straightforward to verify that the sketched ridge regression problem is unchanged if we replace $\mathbf{S}_k \in \mathbb{R}^{p \times q}$ by its projection $\widetilde{\mathbf{S}} \in \mathbb{R}^{p \times r}$ onto its right principal singular vectors, where $r = \mathrm{rank}(\mathbf{S}_k)$ takes the place of $q$.
> Thus, Proposition 7 holds for any sketch if we replace $\alpha$ by $r/p$, but we thought it would be better to tie into the same $\alpha$ used in the rest of the paper.
> Additionally, to use a rank-deficient sketch in practice would generally be wasteful, so we don't anticipate this to be a common case.
>
> > **Is there a known S-transform for CountSketch?**
>
> The S-transform, or any other spectral properties as far as we are aware, for the CountSketch have not been studied in prior literature. One recent work [1] has considered a related sketch which they also call "CountSketch", but which uses only a single hash function, rather than standard CountSketch which uses a growing number of independent hash functions.
> As they show, it is straightforward to see that with a single hash function, the spectrum of the sketch should follow a Poisson distribution; however, even in that case the S-transform is still not known.
>
> Thus, understanding the spectral properties (including the S-transform) of CountSketch is an interesting direction for future work.
> Based on our empirical observations (in Figure A.1 and the new Figure A.2) and the fact that it combines many independent hash functions, we strongly suspect that CountSketch has a Marchenko–Pastur spectrum like Gaussian sketches. We have added a new sentence in Appendix A.4 to re-emphasize this observation.
>
> [1] Fan Yang, Sifan Liu, Edgar Dobriban, David P. Woodruff. How to reduce dimension with PCA and random projections?
> https://arxiv.org/abs/2005.00511

---

> ### Comment · Reviewer_WHDZ · 2023-11-22
>
> Thank you so much for responding to my previous comments and revising the paper accordingly. I very much appreciate the addition of the new appendix section for computational complexity, which is quite informative. I also fully agree regarding the response saying that _"Indeed, as you have written, ....; our presentation is simply made for mathematical convenience. In fact, for sketches based on hash functions such as CountSketch, computing the adjoint of the sketching operator is in general intractable, so this point is critical in sketched machine learning practice."_ I am wondering whether you have emphasized this fact in the revised version of the paper. Although it is common knowledge for the experts, it might be helpful to emphasize this in order to avoid any confusion for future readers.

---

> ### Author Response · Authors · 2023-11-23
> **Follow-up specific response to the Reviewer WHDZ**
>
> Thanks for the suggestion! We have discussed along these lines in the supplement but we agree that this point should be emphasized in the main paper also. We have now squeezed in a line in the main paper right after defining the ensemble estimator (2). It is inked **blue**, as are all other changes in the revision.

---

### Official Review · Reviewer_qfu6 · 2023-11-05

**Soundness:** 4 excellent
**Presentation:** 4 excellent
**Contribution:** 4 excellent
**Rating:** 8
**Confidence:** 4

**Summary:**

Motivating by hyparameter tuning (size of the ensemble, size of the sketches), this paper studies the statistical properties of Generalized-Cross-Validation applied on an ensemble of sketched ridge regressors with skecthing applied to the feature space. First, the authors develop squared risk asymptotics and provide consistency results (Thoerem 2) and then, extend these results to subquadratic error functionals (Theorem 3). These findings hold for the general class of asymptotically free sketching matrices. At the origin of the study, is a key theorem (LeJeune et al. 2022), that states that the sketched inversion of a sketched regularized matrix corresponds to the inversion of the initial regularized matrix with another hyperparameter. This gives rise to Theorem 2 that nicely relates the quadratic risk of the ensemble-based estimator to the risk of the rigde estimator plus a randomness term depending on the sketch via Theorem 1 and the so-cllaed S-transform. Moreover, functional and distribution consistency for general error functional are also proved. Simulations on toy and real data shown in the main paper and appendix confirm the interest of the approach for tuning both the regularization level (or sketch size in fine) and the size of the ensemble.

**Strengths:**

First of all, I would like to say that this is a very nice paper, very well written, solid and with a strong and insightful content. I have learned a lot when reading it.
The main strengths of the paper is its depth of view not only about GCV (which has a simple form) but also about the link between sketched ridge regression and regualrization in ridge regression, and the role of the ensemble trick.
I appreciated the richness of the discussion and the comments all along the paper.
Originality is also present here, with the exploitation of very recent results (LeJeune et al. 2022) but with a special angle here (GCV).

**Weaknesses:**

* Improvment of clarity in Assumption 2 statement and explanation.
The paper has the merit to introduce elements of free probability theory useful in Assumption 2. I regret not to have more intuition here: I can easily imagine that a form of independence (I've read the appendix) between $X^TX$ and $SS^T$ is useful but the notion of limiting S-transform is not at all discussed at this stage (page 4) and Assumption 2 remains not clear at all when beginning to read what follows. Same thing for Theorem 1, describe the $|ambda^+$ function.
* A simple analysis of the complexity in time and memory for the full aproach in constrat to other estimators (CV) would be welcome.
* Bonus: Is it interesting to come back on other risk estimators (Bootstrap) and clearly identify what could be done with this estimator or not with ensemble of sketched ridge regression.

**Questions:**

See my previous comments.

---

> ### Author Response · Authors · 2023-11-18
> **Specific response to the Reviewer qfu6**
>
> Thank you for your comprehensive review and the positive comments on our manuscript.
> We are delighted to hear that you found the paper to be informative and original.
>
> Your insightful review has prompted us to refine our explanations and further clarify the assumptions and theorems within our paper.
> We appreciate your suggestions and hope that these revisions will make our paper even more clear and impactful.
>
> > **The notion of limiting S-transform is not at all discussed at this stage (page 4) and Assumption 2 remains not clear at all when beginning to read what follows.**
>
> We acknowledge that Assumption 2 (now Assumption A after our revision) and Theorem 1 could benefit from additional intuition, especially concerning the S-transform and its role in our results.
> We have added a new footnote on page 4 to ground the assumption concretely with Gaussian and orthogonal sketches, which are commonly studied and proven to be asymptotically free.
> To clarify regarding freeness, the independence between $\mathbf{X}$ and $\mathbf{S}$ is not the only requirement; what is crucial is a form of incoherence, often achieved through rotational invariance of the sketching matrix.
> This incoherence is captured by the concept of asymptotic freeness in free probability theory.
>
> The S-transform, another key concept from free probability theory, is indeed opaque and difficult to give intuition for. Despite this, it is a standard notion in free probability, and the main thing that matters for our results is that is a function only of the eigenvalue distribution of the sketching matrix $\mathbf{S} \mathbf{S}^\top$.
> We assume analyticity of the S-transform, which holds for both Gaussian and orthogonal sketches, but we do not know in general what requirements this places on sketching matrices.
>
> > **Same thing for Theorem 1, describe the $\lambda^+$**
>
> As we define immediately before Theorem 1, $\lambda_{\min}^+$ is the minimum non-zero eigenvalue of the matrix.
> This arises because the multiplication by $\mathbf{X}$ effectively nullifies the zero eigenvalues, leaving only the non-zero spectrum relevant for determining singular inversion, and enabling us to consider zero and negative regularization.
>
> > **A simple analysis of the complexity in time and memory for the full approach in contrast to other estimators (CV) would be welcome.**
>
> We have now included a new Appendix H performing such a comparison. In particular, we demonstrate that the ensemble trick is always competitive with unsketched GCV (which is in turn generally more efficient than $k$-fold CV) as long as $q < p / 2$, and can be competitive even more generally in the right circumstances.
>
> > **Bonus: Is it interesting to come back on other risk estimators (Bootstrap) and clearly identify what could be done with this estimator or not with ensemble of sketched ridge regression.**
>
> We agree that contrasting our approach with other risk estimators, such as the bootstrap, could be valuable.
> The bootstrap, however, does not provide an estimator for prediction risk
> directly and has its computational challenges when dealing with large datasets.
> Our approach offers both computational efficiency and statistical consistency, which are crucial for practical applications.

---

> > ### Comment · Reviewer_qfu6 · 2023-11-23
> > **Feedback on rebuttal**
> >
> > I've read the rebuttal and appreciate the efforts of the authors to clarify some points.

---

### Official Review · Reviewer_t4Ef · 2023-11-05

**Soundness:** 3 good
**Presentation:** 3 good
**Contribution:** 3 good
**Rating:** 8
**Confidence:** 3

**Summary:**

This paper is about the (asymptotic) consistency of generalized cross validation (GCV) for estimating prediction risks of sketched ridge regression ensembles using tools from random matrix theory.

For general subquadratic prediction risk functionals, they extend GCV to construct consistent risk estimators, and obtain distributional convergence of the GCV-corrected predictions in Wasserstein-2 metric.

Although the consistency result seems intuitive and natural, they point out that GCV of the observation sketched ridge regression, is inconsistent, highlighting the subtlety of this subject.

**Strengths:**

This paper presents the theorems in a clear and rigorous way. All notations are presented again on a table in appendix. Theoretical result is supported by experimental result.

**Weaknesses:**

No major weakness is spotted.

**Questions:**

Asymtpotic freeness seems to be an essential assumption in the paper. Although this assumption is experimentally supported by artifical datasets, I wonder if one can observe similar matching on real-world dataset.

---

> ### Author Response · Authors · 2023-11-18
> **Specific response to the Reviewer t4Ef**
>
> Thank you for the positive feedback. We appreciate it.
> We are also happy that you enjoyed the clarity and rigor in our presentation.
>
> > **Although [asymptotic freeness] is experimentally supported by artificial datasets, I wonder if one can observe similar matching on real-world dataset.**
>
> Given the importance of the asymptotic freeness assumption, we agree that it is crucial to verify this assumption in realistic settings, such as with real-world data and practical sketches.
>
> Recall that for Gaussian sketches and uniformly random orthogonal sketches, asymptotic freeness is theoretically guaranteed for any data matrix $X$ independent of $S$. For other types of sketches, such as CountSketch or SRDCT, the validation of the asymptotic freeness assumption is indeed more complex.
>
> However, as you suggest, we can employ further empirical tests to support our claims on real-world data. We have added a new experiment in Figure A.2 in Section A.3.2, where we showcase empirical subordination relations on the real-world datasets we use in Figure 2 (RCV1 and RNA-Seq), similar to the previous experiment in Figure A.1 on artificial data. These results show that the resolvent relation given in Theorem 1, which underpins our theoretical results, holds with good accuracy on real datasets as well.
>
> We hope that these empirical validations, along with the theoretical underpinnings for more readily analyzable Gaussian and orthogonal sketches, offer a compelling argument for the practical relevance of our theoretical contributions.

---

### Author Response · Authors · 2023-11-18
**General response**

We are very grateful to all of the reviewers for their valuable time and feedback on our manuscript. We are happy to hear that the reviewers found our work original, well written, and insightful.

The reviewers have pointed out some weaknesses in the first submission of our work:

- We did not provide any computational complexity analysis (raised by Reviewers: `qfu6`, `WHDZ`, and `GR2e`).
- We only showed empirical support for freeness with artificial diagonal matrices rather than real data (raised by Reviewer: `t4Ef`).

In response to this valuable insightful feedback, we have revised our manuscript with the following significant changes:

- We have added a new Appendix H dedicated to comparing the computational complexity of evaluating GCV and $k$-fold CV on the unsketched data compared using the ensemble trick with sketched data. This comparison provides tangible insights into the practicality and efficiency of our approach in various computational scenarios.
- We have added a new Appendix A.3.2 and experiment in Figure A.2 which presents empirical evidence supporting the assumption of asymptotic freeness for real-world data with CountSketch and SRDCT. This section strengthens the practical relevance of our theoretical result. We have updated the supplemental material to include the code for the new experiment.

In addition, we have made a few independent changes for additional clarity.

- **Reorganization of technical assumptions.** We have reordered Assumptions A and B, moving the data structure assumption (formerly Assumption A, now Assumption B) from before Theorem 1 to just before Theorem 3. This separation clarifies that Theorem 2 is predicated solely on Assumption A (sketch structure), along with minimal data model norm boundedness. On the other hand, Theorem 3, which addresses the consistency of GCV, incorporates the additional Assumption B for a comprehensive analysis. This reorganization of assumptions allows for a more logical flow and a clearer understanding of the theoretical underpinnings of our work.
- **More details for observation sketching.** Proposition 7 has been expanded to include explicit formulae for $\widetilde{\lambda}_0$ and $\nu$, providing a deeper and more transparent exposition of the concepts that were initially in the supplement. The notation throughout the manuscript has been also standardized for consistency, with specific attention to the symbol $\nu$ which is now exclusively used in the context of observation sketching instead of $\widetilde{\mu}$.
- **Additional references.** We have added some additional references both in the main paper and the supplement. Among other things, these provide more review and point to known instances for asymptotic freeness.

All changes made to the manuscript have been indicated in **blue** within the text for easy reference, with the exception of the reorganization of assumptions and for minor typographic corrections, new references, and adjustments to mathematical notations.

We believe these amendments comprehensively address the concerns raised by the referees and significantly augment the manuscript. We are open to any further feedback and remain committed to enhancing the quality of our work. We eagerly await the subsequent comments from the reviewers.

---

### Author Response · Authors · 2023-11-22
**Additional minor revisions to the main paper**

Taking into account follow-up suggestions from Reviewers `GR2e` and `WHDZ`, we have now further squeezed additional comments after Equation 2 and Proposition 6. These are inked **blue** in the revision, as are all other changes indicated in the general response below.

---

### Meta-Review · Area_Chair_YXpa · 2023-12-10

**Metareview:**

This work provides theoretical foundations for generalized cross validation of an emsemble of sketched ridge regression. It is first studied for the squared risk, and then extended to subquadratic risks. The theoretical results are supported by experiments on synthetic and real data.

The reviewers all highly appreciate the theoretical contribution of this work and their insights for practical use.

**Justification For Why Not Higher Score:**

A more explicit discussion about how the theoretical foundations established in this paper could guide better sketching in practice may potentially increase the impact of this work.

**Justification For Why Not Lower Score:**

The paper is overall well-written with solid results.

---

### Decision · Program_Chairs · 2024-01-16

Accept (spotlight)